# Convergence of resistance and evolutionary responses in *Escherichia coli* and *Salmonella enterica* co-inhabiting chicken farms in China

Michelle Baker[1,13], Xibin Zhang[2,13], Alexandre Maciel-Guerra[1,13], Kubra Babaarslan[1], Yinping Dong[3], Wei Wang[3], Yujie Hu [3], David Renney[4], Longhai Liu[5], Hui Li[6], Maqsud Hossain[1], Stephan Heeb [7], Zhiqin Tong[6], Nicole Pearcy[1,7], Meimei Zhang[8], Yingzhi Geng[8], Li Zhao[9], Zhihui Hao[10], Nicola Senin [11], Junshi Chen[3], Zixin Peng [3] ✉, Fengqin Li[3] ✉ & Tania Dottorini [1,12] ✉

Sharing of genetic elements among different pathogens and commensals inhabiting same hosts and environments has significant implications for antimicrobial resistance (AMR), especially in settings with high antimicrobial exposure. We analysed 661 *Escherichia coli* and *Salmonella enterica* isolates collected within and across hosts and environments, in 10 Chinese chicken farms over 2.5 years using data-mining methods. Most isolates within same hosts possessed the same clinically relevant AMR-carrying mobile genetic elements (plasmids: 70.6%, transposons: 78%), which also showed recent common evolution. Supervised machine learning classifiers revealed known and novel AMR-associated mutations and genes underlying resistance to 28 antimicrobials, primarily associated with resistance in *E. coli* and susceptibility in *S. enterica*. Many were essential and affected same metabolic processes in both species, albeit with varying degrees of phylogenetic penetration. Multimodal strategies are crucial to investigate the interplay of mobilome, resistance and metabolism in cohabiting bacteria, especially in ecological settings where community-driven resistance selection occurs.

AMR is a major global health problem and livestock farms and their surrounding environment have been highlighted as a potential source of AMR infections[1]. The development of AMR in individual bacterial species is dependent not only on antibiotic exposure but also on the presence of other bacteria within their environment, with which they interact[2]. Hence, acknowledging the extent to which bacteria within the same environment are able to co-evolve and share their genome could help the development of more efficient treatments to fight AMR[3,4]. In this study, we have focused on two important opportunistic pathogens found in livestock *Escherichia coli* and *Salmonella enterica*, which both display high levels of drug resistance, and have zoonotic potential[5,6]. These species can share genetic

material both within and potentially between species, a mechanism by which AMR is spread[7].

A recent study found that in model experimental systems *E. coli* and *S. enterica* co-cultures evolved different antimicrobial resistance mechanisms compared to monocultures of either species subject to the same experimental conditions[8]. It was also shown that within the same environment, *E. coli* and *Salmonella spp.* are able to directly communicate using bacterial signalling, increasing antibiotic tolerance[9]. A previous study showed that plasmids carrying antimicrobial resistance genes (ARGs), can be transferred from *S. enterica* to *E. coli*[10] and proposed that the similarity of the plasmids in *S. enterica* isolated from chicken gut, to plasmids found in pathogenic *E. coli*,

another gut resident, could indicate transmission between the species[10]. Hence, both bacterial signalling and transfer of genomic material could impact AMR for species within the same environments. However, AMR phenotypes in *E. coli* are not a good indicator of AMR phenotypes in *S. enterica* even when taken from the same sample, suggesting that complex AMR dynamics need to be further studied on a genomic level to understand this disparity[11].

In this study, we collected isolates from the same biological samples, from animals and surrounding environments, on ten commercial poultry farms and four connected slaughterhouses in three provinces of China over two-and-a-half years. We focused on two important opportunistic pathogens *E. coli* and *S. enterica*, common in agricultural settings and representing a significant cause of diarrheal disease-associated mortality in humans, particularly, in low-to-middle income countries[12]. Isolates of both species were collected from within the same community and across interconnected ones including chicken faeces, chicken carcasses, chicken feather, chicken caecal droppings, chicken feed, external soil, barn environment, wastewater, anal swabs, abattoir environment and drinking water. We first characterised the population structure, evolution and AMR phenotypes of both *E. coli* and *S. enterica* circulating strains, highlighting differences across environments and hosts. Next, we used a novel data-mining approach that merges Bayesian divergence analysis, genome-scale metabolic (GSM) models, culture-based techniques, and machine learning (ML). The new data analysis pipeline was designed to provide a wider perspective on the relationships between genetic elements of *E. coli* and *S. enterica* isolates and AMR, by data mining all the possible correlations between SNPs in the coding and non-coding regions of the core genome as well as all the accessory genes, and AMR resistance/ susceptibility to multiple antibiotics. We found that most isolates of *E. coli* and *S. enterica*, within the same host and environment, possessed the same AMR-carrying MGEs, which also appear to have co-evolved, which in real-world settings pose a high risk of AMR transfer to humans and the environment[13]. Moreover, these AMR-carrying MGEs could also potentially be a pre-requisite for the bacteria to occupy the same host and environment, as the horizontal gene transfer process may drive the development of host adaptation[14,15]. Notably, these MGEs encoded clinically relevant ARGs (bla$_{CTX-M}$, *APH(3)*, *floR*, *mphA*, and *qnrS1*). By utilizing a machine learning pipeline that incorporates a comprehensive set of genetic features encompassing the entire genome, including SNPs within core genes, intergenic regions, and accessory gene content, we were able to pinpoint the genes, mutations, and regulatory elements that displayed a strong correlation with the antimicrobial susceptibility profiles against up to 28 different antimicrobials for each isolate of every species. The ML results revealed that both species had a common subset of features strongly linked to AMR (including both known and novel mutations and genes). When analysing the AMR-associated features using GSM models and protein-protein interaction networks, we found that they were linked to the same functional pathways, which were essential for growth, and which were affecting biochemical fluxes within both bacteria indicating potential common metabolic adaptations.

## Results

### Differences in antimicrobial-resistant phenotypes, phylogeny, SNPs, and evolution are observed between the *E. coli* and *S. enterica* isolates across farms, sources, and time

Altogether, we collected 518 *E. coli* isolates and 143 *S. enterica* isolates from a total of 692 animal and 285 environmental samples (see Methods) taken from both farms and abattoirs; with an overlap of 113 samples where both species were cultured from the same sample. Samples were taken from ten farms and four connected abattoirs in three provinces of China: Henan (three farms and two abattoirs), Liaoning (three farms and one abattoir) and Shandong (four farms and one abattoir) over a 25-month period, between March 2019 and April

2021, Supplementary Data 1. In each location samples were taken at three points over the six-week broiler production cycle: $t_1$ - mid-life ~3 weeks old, $t_2$ - full grown ~6 weeks old and $t_3$ - end-of-life ~6 weeks and 1–5 days, post-slaughter sampling. The most represented province by isolate count for *E. coli* was Henan ($n = 219$) followed by Shandong ($n = 209$) then Liaoning ($n = 90$). Analogously, for *S. enterica* the most represented province was Henan ($n = 111$), followed by Liaoning ($n = 25$) then Shandong ($n = 7$). The 518 *E. coli* and 143 *S. enterica* isolates were laboratory tested for resistance/susceptibility to up to 28 antimicrobials, Fig. S1. In general, most isolates in both species were resistant to penicillins, monobactams, cephalosporins and aminoglycosides, whilst resistance to carbapenems and polymyxins was much less frequent, Fig S1. To understand the genetic relatedness of the isolates and its relationship with AMR, maximum likelihood phylogenetic trees were constructed for each species, using the core genome (based on genes present in >99% of isolates) from whole-genome sequencing data, Fig. 1 and Fig. S2 (further information in Supplementary Note 1 in the Supplementary Information). In *E. coli*, resistance phenotypes were not related to phylogroup, however in *S. enterica* the serotype Enteritidis was more susceptible than other serotypes to all antibiotics except for polymyxins for which it was more resistant, Fig. S2. When considering individual antibiotics, the three tetracycline antibiotics (tetracycline, doxycycline and minocycline) were significantly different between species (adjusted $p$-value < 0.001, chi-squared test with Bonferroni correction), for doxycycline (DOX) and minocycline (MIN) the resistance was more frequently observed in *S. enterica* and for tetracycline (TET) it was more frequently observed in *E. coli*. Similarly, both polymyxins (polymyxin B and colistin) had resistance frequency significantly higher in *E. coli* (corrected $p$-value < 0.001, chi-squared test with Bonferroni correction). Two 3rd generation cephalosporins (cefepime and ceftazidime) were proportionally more resistant in *S. enterica* (corrected $p$-value < 0.001, chi-squared test with Bonferroni correction). Finally, the aminoglycoside streptomycin was found to have a similar proportion of resistant cases, but more intermediate and less susceptible isolates in *S. enterica* compared to *E. coli* (corrected $p$-value < 0.001, chi-squared test with Bonferroni correction).

The phylogeny results showed that *E. coli* isolates (Fig. 1 and Fig. S2a) were highly diverse with no clustering observed across isolates on sample source type or farm. Sample source type, farms, provinces, and collection dates were not associated with phylogroups ($p$-values > 0.05, Fishers exact test for count data with simulated $p$-value). Using the standard association index ($I^s_A$) to measure for clonality in the population[16,17], for the *E. coli* cohort we found the $I^s_A$ to be 0.2126 ($p$-value < 0.0001) at whole cohort level and 0.1313 ($p$-value < 0.0001) at ST type level, indicating the presence of clonality. Conversely, for *S. enterica* (Fig. 1 and Fig. S2b) statistically significant differences were observed among isolates in relation to serotypes per farm, source type and collection date ($p$-values < 0.0001, Fishers exact test for count data with simulated $p$-value), with Havana enriched in ceca samples and Enteritidis enriched in waste water (adjusted $p$-values < 0.05, chi-square), Kentucky enriched in Liaoning 1 and Henan 3 (adjusted $p$-values < 0.05, chi-square) and Enteritidis enriched in Henan 2 and Shandong 2 (adjusted $p$-values < 0.05, chi-square). June 2020 was associated with an enrichment of Havana serotype samples; this collection month was associated with $t_1$ collections from Henan 1, however Henan 1 collections across all timepoints were not enriched for Havana serotype. For *S. enterica*, the $I^s_A$ of 0.9077 ($p$-value < 0.0001) at whole cohort level and 0.3883 ($p$-value < 0.0001) at ST type level indicated stronger clonality compared to *E. coli*.

To further assess the genomic relatedness of both the *E. coli* and *S. enterica* isolates in our cohort we measured the number of different core genome SNPs, in a pairwise manner across all isolates and performed network analysis (see Supplementary Note 2). The results

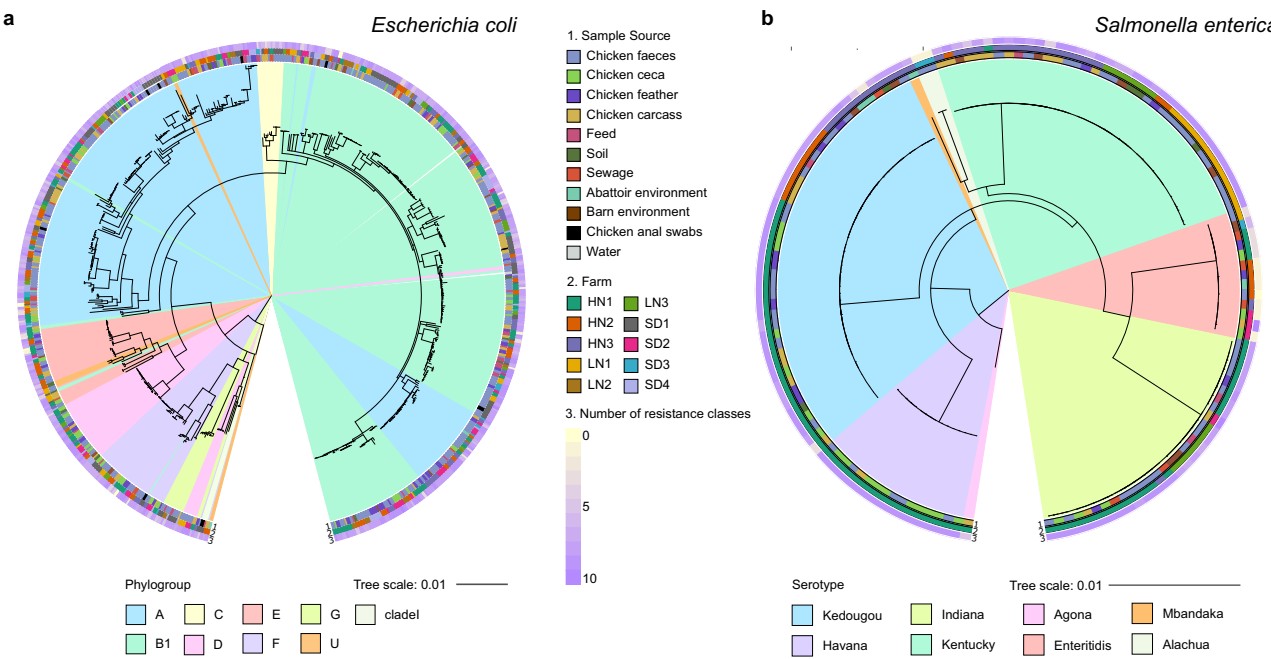

**Fig. 1 | Phylogeny of *E. coli* and *S. enterica* isolates collected from 10 commercial broiler farms in China.** (**a**) Maximum likelihood phylogenetic tree of the whole cohort of *E. coli* isolates based on core genome of the 518 isolates, cultured from the animal and environmental samples collected from the 10 farms and 4 abattoirs. Phylogroups are shown as coloured sections. Sample source, region and number of resistance classes are shown as rings around the tree. **b** Maximum likelihood phylogenetic tree of the whole cohort of *S. enterica* isolates based on core genome of the 143 isolates, cultured from the animal and environmental samples collected from the 10 farms and 4 abattoirs. Serotypes are shown as coloured sections. Sample source, region and number of resistance classes are shown as rings around the tree. Farm names are abbreviated: Henan 1 (HN1), Henan 2 (HN2) Henan 3 (HN3), Liaoning 1 (LN1), Liaoning 2 (LN2), Liaoning 3 (LN3), Shandong 1 (SD1), Shandong 2 (SD2), Shandong 3 (SD3), Shandong 4 (SD4).

indicate a geographic diversity and evolution in both species. Finally, to better understand the phylogenetic and SNP distance differences observed between *E. coli* and *S. enterica* and to explain the evolution of these two species, we performed a Bayesian inference of phylogeny for the major phylogroups and serotypes in our cohort. Bayesian Evaluation of Temporal Signal (BETS) analysis showed that eight of the major phylogroups and serotypes contained a temporal signal suitable for analysis (*S. enterica*: Enteriditis, Indiana, Kentucky, Kedougou, Havana; *E. coli*: A, E and F), Supplementary Table 1. The vast phylogenetic and genomic (SNPs) diversity of *E. coli* and *S. enterica* was supported by the Bayesian divergence analysis results showing that *S. enterica* strains were more recently evolved, with MRCAs of *S. enterica* being around 5 years (Fig. S3), compared to MRCAs for *E. coli* phylogroups (Fig. S4) which was in the range of hundreds of years. Root heights varied also with *S. enterica* serotype averages between 5–29 years and *E. coli* phylogroup root hesight averages between 151 and 400 years, Supplementary Data 2. Predicted nucleotide substitution rates for both species were relatively consistent across phylogroups and serotypes and were higher (two orders of magnitude) than though typically found in literature[18,19]. These higher than expected rates may have been caused bacterial stress induced by the usage of antibiotics on the farms in our cohort, which could be leading to increased mutation rates[20–23]. Of note, when considering genome-wide substitution rates in *E. coli*, clade E is predicted to mutate more slowly than clades A and F. Others have observed that phylogroup E tends to have a larger genome (as also seen in our data) resulting in lower replication rates[24] and one could speculate that this lower replication rate could result in the lower mutation rate we predict. This is a pattern we also see in our *S. enterica* isolates, with the largest genomes predicted to have the slowest mutation rates. For the two largest *E. coli* serotypes, for which a temporal signal was present (O83:H42 and O8:H16), the MRCA being also longer, compared to *S. enterica* (~20 years) with some clustering by farm observed (Fig. S5).

Together, these results show a diverse, non-clonal and evolutionarily distantly related cohort of *E. coli* isolates potentially indicative of large circulation populations of commensal *E. coli* in these farms, with many samples carrying multidrug resistance. In contrast *S. enterica* shows highly clonal, region specific, and evolutionarily closely related isolates, more indicative of outbreaks of bacterial growth, and as with *E. coli* many isolates are multidrug resistant.

### Within-host favours sharing of plasmids and mobile genetic elements carrying AMR genes between *E. coli* and *S. enterica*

Genomic mobility through plasmids and mobile genetic elements is correlated to antimicrobial resistance, giving the opportunity for genomic content to be shared between bacteria[25]. In our cohort 99.4% of *E. coli* isolates and 88.1% of *S. enterica* isolates carried plasmids (Supplementary Data 3). Using MOBsuite[26] plasmids were reconstructed and typed from both species. The total number of plasmids reconstructed was 3169 in *E. coli* (mean of 6.1 per isolate) and 572 in *S. enterica* (mean of 4 per isolate). Plasmids found in *S. enterica* were also significantly smaller (*p*-value < 0.0001) with a size range of 977bp-261kbp in *S. enterica* compared to 983bp-448kbp in *E. coli*. Gene content was found to significantly differ between plasmids from each species (*p*-value < 0.0001) with *E. coli* plasmids carrying on average 0.82% (±0.33%) of the genome and *S. enterica* plasmids carrying 0.45% (±0.22%). Similar proportions of plasmids with no identifiable replicon and multiple replicons were found in both species. *S. enterica* plasmids with no replicon were enriched in Liaoning province (LN1 and LN3), and Havana and Kentucky serotypes. *E. coli* no-replicon plasmids were enriched in farms from each of the provinces (HN1, LN3, and SD3) and phylogroups B1 and G. However, *E. coli* carried significantly more novel plasmids (*n* = 59) compared to *S. enterica* (*n* = 4), *p*-value = 0.047. In *E. coli* the most prevalent plasmid replicon was IncFIB, found in 402 isolates, followed by IncHI2 found in 239 isolates. In *S. enterica*, Col(-pHAD28) was most prevalent, found in 79 isolates followed by IncHI2

found in 58. All these plasmids have previously been associated with AMR gene carriage[17,27–29]. When comparing plasmid types and content in the 113 *E. coli* and *S. enterica* strains isolated from the same samples we found that 70.6% of isolate pairs cultures from the same sample carried the same plasmid types. In the plasmid types that were shared in isolate pairs, we observed more sharing than would be expected by chance compared to typical plasmid prevalence in isolates collected from chickens in China ($p < 0.0001$, chi-squared test). Among these the most common shared replicon types were Col(pHAD28) and IncHI2, Fig. S6. In particular, Col(pHAD28) and IncHI2 were found in 29 and 27 isolate pairs respectively, Fig. S6. For IncHI2, a major AMR carrier[30], a Bayesian phylogenetic analysis of reconstructed IncHI2 plasmid sequences within our samples, showed that generally *E. coli*-sourced plasmids and *S. enterica*-sourced plasmids fell in different clades with a MRCA of 16 years, Fig. 2. However, there were two regions of more recent evolution (red rectangles on Fig. 2), with the MRCA dating back to only 2019, and within these we have pairs of *E. coli* and *S. enterica* strains isolated from the same sample, potentially indicative of recent transmission of this plasmid between species.

Next, we assessed whether the same mobile ARGs (AMR genes in the 5 kb vicinity of an MGE)[31–35] were found in pairs of *E. coli* and *S. enterica* isolates collected from the same sample. Considering known ARGs according to the CARD database[36], we found 88 *E. coli*-*S. enterica* isolate pairs (78%) carrying the same mobile ARG (Supplementary

Data 4). These mobile ARGs included clinically relevant $bla_{CTX-M}$, *APH(3)*, *floR*, *mphA*, and *qnrS1* genes, all known important to AMR in human health[37] (Fig. S7). Overall, 14 different mobile ARG patterns were found (Supplementary Data 4). The spread of mobile ARGs across farms was broadly reflective of the distribution of isolate pairs ($p$-value = 0.064, chi-squared test), Fig. S7a. The spread of mobile ARGs across source types significantly differed from the expected distribution ($p$-value < 0.001, chi squared test) driven by a higher-than-expected number of mobile ARGs in feed samples (standardized Pearson residual = 4.468), Fig. S7b.

Gene structures of the ARG containing contigs for mobile ARGs showed highly conserved structures in both *E. coli* and *S. enterica* isolates, from the same sample and those from different samples, as seen for example in the gene structures of the *QnrS1* gene (Fig. 3).

Overall, the patterns of plasmid and mobile ARG presence suggest there is high potential for transfer between *E. coli* and *S. enterica* in the farm environments that we have studied which could lead to the spread of clinically important ARGs, and suggestions that, historically, this is likely to have occurred. However, we have not seen any evidence of this happening on a large scale in the timeframes we have studied.

To investigate the influence of co-inhabitation of *E. coli* and *S. enterica* as well as the influence of country of collection (antimicrobial usage and microbial ecology) on the observed results (proportion of distinct plasmid types and mobile ARGs, as well as amount of sharing between the two bacteria species), we considered three control sets. The first was formed of Chinese bird samples with apparent absence of co-inhabitation (the challenges related to demonstrating absence of co-inhabitation are discussed in Supplementary Note 3). The second and third were European, retrieved as publicly available data from two previous research projects. Namely: "EFFORT against AMR[38]" (206 *E. coli* isolates from chicken faeces with AMR phenotypes, collected in five different European countries - Denmark, Germany, Switzerland, Poland, and Spain - where the birds are subjected to strict control measures against *Salmonella*, thus co-presence of *S. enterica* is unlikely) and "ENGAGE[39]" (92 *S. enterica* chicken cecum and faeces isolates from Italy, where samples may or may not contain co-inhabiting *E. coli*). We then performed multiple statistical comparison tests to search for differences in proportions of plasmids and mobile ARGs (including shared ones) when considering our Chinese isolates (faeces and caecal swabs) and the European ones (see Supplementary Note 3 for further details). In general, our results show that:

a. both the number of distinct plasmid types and the number of mobile ARGs is higher in the Chinese cohort, indicating a likely influence of the country of collection in the observed results;

b. there is a positive correlation between co-inhabitation of *E. coli* and *S. enterica*, and the proportions of mobile genetic elements (plasmids and mobile ARGs) observed in their isolates, as indicated in particular by the comparison of the Chinese co-inhabiting vs not co-inhabiting cohorts, and by the comparison of Chinese *E. coli* isolates (with confirmed co-inhabitation) with the EFFORT *E. coli* isolates (with likely absence of co-inhabitation, due to the very low *Salmonella spp.* prevalence, owing to the EU *Salmonella* control measures Regulation (EC) No. 200/2012[40]– resulting in strong *Salmonella* control/vaccination programmes and less than 0.5% *Salmonella* positive flocks in commercial broiler chickens).

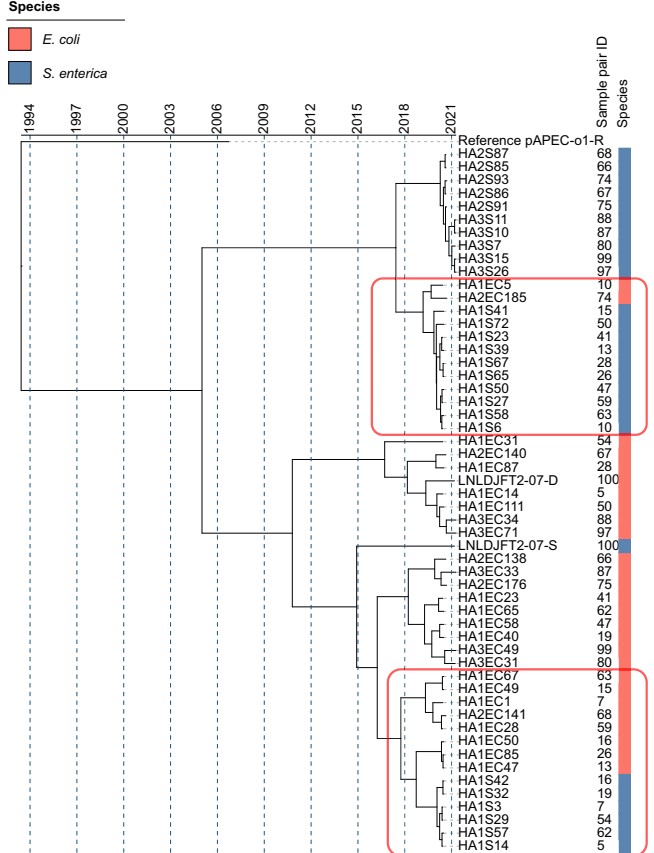

**Fig. 2 | Bayesian evolutionary analysis shows recent common evolution of IncHI2 plasmids from *E. coli* and *S. enterica* isolates collected from the same samples and environments.** Bayesian estimation of IncHI2 plasmid divergence times using sequencing data from 27 *E. coli* and 27 *S. enterica* strain pairs collected from the same samples, with the reference pAPEC-o1-R plasmid used as an outgroup (GenBank DQ517526.1). Sample pair numbers are given with each number corresponding to an *E. coli* and *S. enterica* isolate recovered from the same sample. Red rectangles indicate two areas of recent evolution containing IncHI2 plasmids from both species.

## Machine learning unravels known and novel AMR-associated core genome SNPs and accessory genes correlated with resistance/susceptibility profiles to multiple antimicrobials in both species

Given the similar proportions of resistance, against 28 antimicrobials, across the 518 *E. coli* and against 26 antimicrobials across 143 *S. enterica* isolates, despite phylogenetic differences, we further investigated which genetic determinants (features) were underlying the

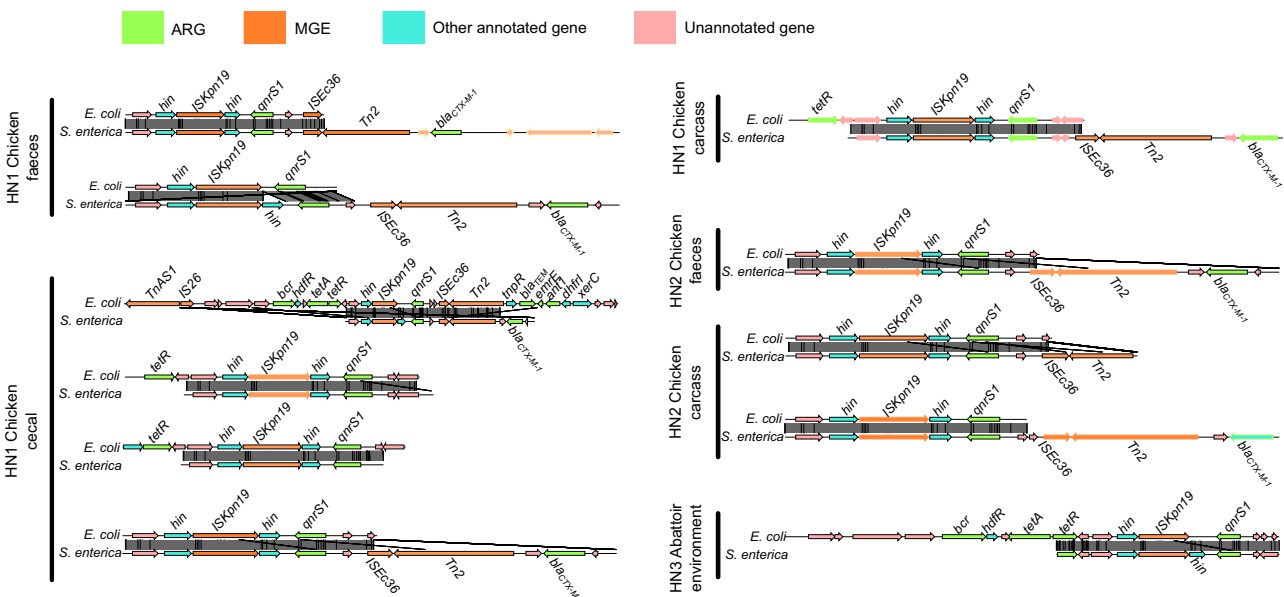

**Fig. 3 | Genomic structure of *qnrS1* mobile ARG patterns in *E. coli* and *S. enterica* isolates collected from the same samples.** Each pair of samples was aligned by *qnrS1*. Coding DNA sequences are coloured according to the function of their encoded protein, as follows: ARGs were coloured in green; MGEs were coloured in orange, other annotated genes were coloured in cyan and other open reading frames (ORF) were coloured in pink.

experimentally determined resistance/susceptibility profiles and if they were in common between the two species. We assessed whether the resistance/susceptibility profiles of either *E. coli* or *S. enterica* could be explained by the presence of known AMR genes (as found in CARD[36]). As previously done by us and others[17,41,42], we used the Jaccard/Tanimoto similarity coefficients[43,44] to identify the intersection between resistance phenotypes and the known AMR genes in a pairwise manner. We found, for *E. coli*, that the maximum statistically significant Jaccard value was 0.62 with many antibiotics achieving a maximum Jaccard association of much less than this (range of maximum Jaccard coefficient per antibiotic 0.006–0.62, mean 0.32), Supplementary Data 5. Similarly for *S. enterica*, the maximum statistically significant Jaccard similarity coefficient was 0.71 (range of maximum Jaccard coefficient per antibiotic 0.01–0.71, mean 0.51), Supplementary Data 5. Given the low values of the Jaccard coefficients, we conclude that the presence of known AMR genes (as found in the CARD database) alone was not able to adequately explain AMR phenotypes in either the *E. coli* or *S. enterica* isolates in our study, necessitating an alternative approach. To do this we employed a supervised machine learning pipeline (Figs. S8a and S9a). The pipeline is aimed at mining sequencing data to identify the genetic elements that more strongly correlate with observed phenotypic differences, which in this case are related to resistance/susceptibility to antibiotics. Note that whilst the previously described analysis specifically focused on plasmids and mobile ARGs, the machine learning pipeline was designed to provide a wider perspective by investigating all the SNPs in the coding and non-coding regions of the core genome as well as all the accessory genes in search for correlation to AMR traits. Firstly, a test for the presence of multicollinearity was carried out on the ML features. The variance inflation factors (VIFs) were computed using the StatsModels package in Python. Multicollinearity was found in all antibiotic models: in *E. coli* models VIFs ranged from 2 to 265 (mean 109), with many at infinity; in *S. enterica* models VIFs ranged from 7 to 50 (mean 20), with many at infinity. For this analysis we improved our original ML-based core methods[17,45,46] in a notable way (see Fig. S8a): information about different genetic features (SNPs from coding and non-coding regions and presence/absence of accessory genes) were fed all at the same time as input into the model. This would capture the co-occurrence of multiple mechanisms (mutations, horizontal gene transfer−HGT) as

well as their additive effect on resistance. To ensure the best performance and avoid any bias both species underwent population structure correction and were tested against a panel of seven ML methods, five classifiers and two meta-methods (Linear SVM, RBF SVM, Random Forest, Extra tree classifier, Logistic regression, Adaboost and XGBoost), with the best classifier performance assessed using the Friedman and Nemenyi tests (see Methods and Fig. S10 for details). To correct for unbalanced classes SMOTE[47] or SMOTEENN[48] was used to synthetically oversample the minority class for both and under sample the majority class for SMOTEENN, and nested cross validation was employed. This procedure necessitated a minimum of 12 samples in the minority class for each predictive model, hence some antimicrobial datasets were insufficient for ML. For *E. coli*, ML was able to be carried out on 21 antimicrobials. Of these 21 models, 17 achieved high performance with an AUC greater than 0.9, and 13 of those achieved an AUC greater than 0.95, with the doxycycline performing best (AUC = 0.985), Fig. S8b. Similarly, all other performance metrics were good for these 17 models, Fig. S8c−g. All four poorly performing predictive models (amoxicillin-clavulanic acid, ceftazidime, cefotaxime-clavulanic acid and cefepime) were beta-lactam class antimicrobials, and these models also had a small number of features selected as predictive by the data pre-processing, Fig. S11a. The performance metrics for all classifiers are in Supplementary Data 6, while the selected features for each antibiotic model are in Supplementary Data 7. For *S. enterica* a different ML pipeline was used as shown in Fig. S9a with pre-processing of all three feature types separately using an extra tree classifier and resampling of classes using SMOTEENN[48], see "Methods" for details of ML pipeline choice. Due to the relatively low sample number and unbalanced nature of the resistance-susceptibility profile of the isolates predictive models were built for only the thirteen antimicrobials with enough samples in the minority class. Of these thirteen models, all had good AUC performance, >0.9, with ten achieving an AUC >0.95, Fig. S9b. Performance across all metrics was high, Fig. S9b−g, and gentamicin (GEN) performed best across all metrics. Unlike for *E. coli*, the poorer performing antibiotic models came from three different antibiotic classes: beta lactams (AMC), diaminopyrimidines (SXT) and tetracycline (TET) and poorer performing models did not necessarily have a low number of features, Fig. S11b. The performance metrics for all classifiers are in Supplementary

Data 6, while the selected features for each antibiotic model are in Supplementary Data 7.

The AMR-related features selected by the machine learning showed differences according to the type and genomic localization (i.e. SNPs/accessory genes) correlated to the resistance against a specific antibiotic. Both synonymous and nonsynonymous core genome SNPs were selected by the ML pipeline, Supplementary Data 8. In both species, accessory gene presence-absence tended to be selected in multiple AMR models, whilst SNP features originating from the core genome tended to be more specific to a single antibiotic, Fig. S12. For each species, and each antibiotic model, SHAP (SHapley Additive exPlanations) values were calculated. SHAP values disclose the individual contribution of each gene mutation/accessory gene on the output of the model, for each isolate. Figures 4 and 5 use a bee swarm plot to show the correlation of each of the top ten most important genes for each model for *E. coli* and *S. enterica* respectively, to predict the resistance phenotype. For *E. coli*, in most cases the presence of the SNP in genes/accessory gene was positively correlated to the resistance phenotype, though there were some genes for which the absence of the SNPs/accessory gene was positively correlated to the resistance phenotype, most notably in the prediction of CAZ-C and AMI resistance, Fig. 4. For *S. enterica*, a more mixed pattern emerged, with many gene SNPs/accessory genes negatively correlated to the resistance phenotype, i.e., correlated to susceptibility, Fig. 5. Most notably, for gentamicin, the presence of 9 of the top 10 genes were positively correlated to the susceptible phenotype.

To understand the relationship between AMR phenotype and genotype, we cross-referenced the SNPs that acted as predictors for AMR for each antibiotic to the pangenome for each model data set and identified the corresponding genes. In total the *E. coli* features (SNPs and accessory genes) that correlated to AMR profiles across all antibiotic ML models, mapped back to 4419 genes, 20.3% of the pangenome. Genes correlated to *E. coli* resistance were enriched for plasmid located genes with 256 (5.6% of the 4419 genes) found in plasmids, compared to 2.4% of the pangenome. In contrast, genes correlated to *S. enterica* were less enriched for plasmid localisation, with 89 genes plasmid-located (2.7%) compared to 1.9% in the pangenome. Of these just 1% (44 genes) were known AMR genes (defined as being found in public AMR databases, see Methods). For *S. enterica* ML selected features (SNPs and accessory genes from all ML models) mapped back to 3501 genes, 44.2% of the pangenome and 1.14% of these (40 genes) were known AMR genes. Considering these known AMR genes that were selected from the ML pipeline as being predictive of resistance/susceptibility (SNPs or accessory genes), many of these were found in multiple ML models (82% for *E. coli* and 78% for *S. enterica*), Fig. 6. For *E. coli*, of the 44 known AMR genes selected, those relating to multidrug resistance (MDR) were most frequent (9 of 44 genes), followed by aminoglycoside genes (8 of 44). For *S. enterica* of the 40 known AMR genes, aminoglycoside genes (11 of 40) were most frequent, followed by beta lactam genes (8 of 40).

As done previously[45], we selected only the top-ranked AMR-related genetic determinants that most strongly contribute to the performance of the ML classifier. This was done by limiting our analysis to genes corresponding to the top 10% of ranked features recognized as discriminant by the AMR classifiers. This led to a total of 1089 genes in *E. coli* and 688 in *S. enterica*, that were correlated to at least one antibiotic. Of these 88 were correlated to at least one antibiotic in both *E. coli* and *S. enterica*. Protein-protein interaction (PPI) networks of these 88 genes, Fig. S13, showed that in both species these genes were interacting with each other significantly more frequently than would be expected by chance (*E. coli* adjusted *p*-value $1.17 \times 10^{-11}$; *S. enterica* adjusted *p*-value $1.93 \times 10^{-8}$; hypergeometric test). These genes fell into multiple functional groups according to KEGG ontology, Supplementary Data 9, including various metabolic processes, DNA repair, cellular transport, cellular community and drug resistance all of which have

been linked to AMR[45,49,50]. Of particular interest, the genes *ompC* and *oppC* which were selected in both species are involved in beta-lactam resistance[51]. The *opp* gene operon is also known to be involved in aminoglycoside resistance[52] and quorum sensing[51], whilst *ompC* is known involved also in signal transduction[51]. The *cys* gene pathway had many genes selected by machine learning, these genes were highly connected with other genes in the PPI networks and in the KEGG ontology were involved in purine and pyrimidine metabolism, previously suggested to be important in resistance[45].

## Machine learning reveals differences in AMR-associated mutations and accessory genes between co-inhabiting *E. coli* and *S. enterica* in chickens and those of *E. coli* and *S. enterica* that do not necessarily co-inhabit

Similar to what done when searching for plasmids and mobile genetic elements carrying AMR genes and present in co-inhabiting *E. coli* and *S. enterica*, we wanted to investigate whether the results (identified genetic elements associated with AMR) predicted by the machine learning pipeline trained with the Chinese *E. coli* and *S. enterica* cohorts (with confirmed co-inhabitation of the bacterial species) would change if the same pipeline was trained with different cohorts (e.g., from another country, and/or with no confirmed co-inhabitation). To this purpose, we resorted to our Chinese *E. coli* not co-inhabiting chicken isolates and again to the previously described EFFORT (*E. coli*) and ENGAGE (*S. enterica*) European datasets, and investigated how the predictions would change once the machine learning pipelines were trained using those sets. The full results are given in Supplementary Note 4, Supplementary Tables 3–6, and Supplementary Data 12–14.

In summary, when looking at the European cohorts and at the Chinese not co-inhabiting isolates, the comparative analyses were only possible for subsets of antibiotics with sufficient data for ML training. The results indicate a larger overlap of results (genetic elements shared by *E. coli* and *S. enterica*, identified by ML) when comparing co-inhabiting and not co-inhabiting isolates collected in China, with respect to overlaps observed when comparing results from co-inhabiting Chinese isolates with European isolates. In more detail, a 39.67% prediction overlap was observed when comparing Chinese co-inhabiting and not co-inhabiting datasets, whilst a 14.19% prediction overlap was observed when comparing Chinese co-inhabiting *E. coli* vs European not co-inhabiting *E. coli* (EFFORT dataset), and 3% prediction overlap was observed when comparing Chinese co-inhabiting *S. enterica* vs European not-necessarily co-inhabiting *S. enterica* (ENGAGE). As stated earlier, the absence of co-inhabitation in the European samples cannot be fully demonstrated and may be unlikely at least for the ENGAGE set, hinting at the observed differences being likely influenced in no small amount by country of collection, although more investigation should be needed to fully assess the influence of both co-inhabitation and country of collection in the machine learning predictions.

## Integration with GSM models reveals that many top-ranked AMR-related genes are essential genes for growth and affect common metabolic pathways correlated to AMR in both species

To further investigate the systemic relationships connecting the identified AMR genetic signatures on a mechanistic level, and to elucidate their mechanistic effects beyond genes encoding proteins targeted by drugs (i.e., positive selection in basal biosynthetic, regulation, and repair pathways), we integrated the top 10% ranked genetic determinants identified by ML with the GSM models of *E. coli* (K-12 MG1655, iML1515[53]) and *S. enterica* (STM v1.0[54]), see Supplementary Data 10. Many accessory genes, as missing from these reference genomes, were not included in the analysis. For *E. coli*, for each antibiotic model on average 10% (n = 30, mean across 19 antibiotic models) of metabolic genes accounted for in iML1515 from the top-ranked 10% features (n = 135, mean across 19 antibiotic models),

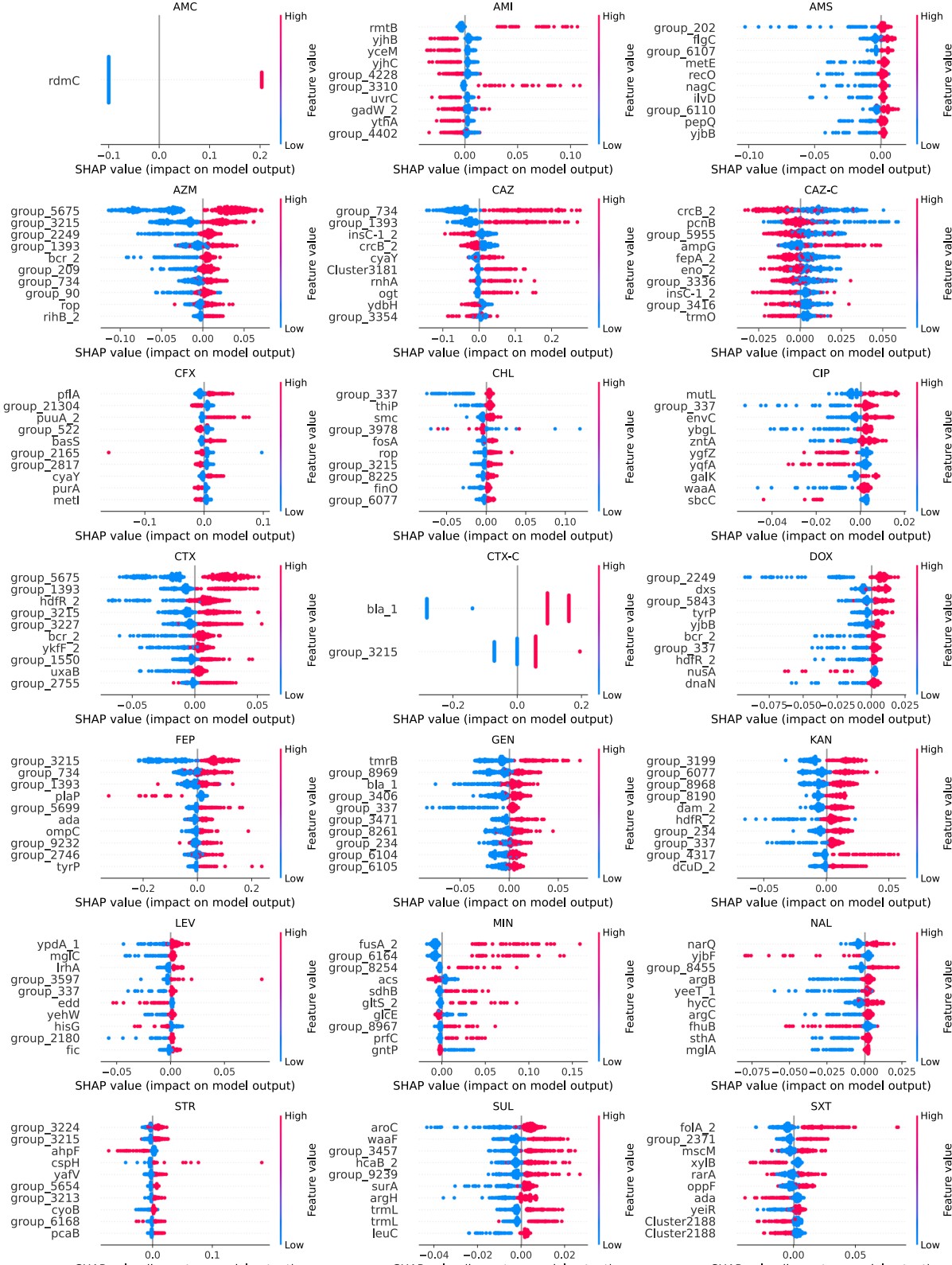

**Fig. 4 | Bee swarm plot of SHAP-calculation for the ten highest ranking genes for each of the *E. coli* antibiotic ML models.** Genes are sorted by their mean absolute SHAP value in descending order with genes carrying most important features (SNP or accessory gene) at the top. Genes labels as 'group_' are unannotated genes. Each dot corresponds to one isolate in the study. The colour red indicates the presence of the feature while the colour blue indicates its absence.

The bee swarm plot shows how the different feature in each isolate affects the prediction of the ML model towards resistance to the respective antibiotic. Positive SHAP values indicate a change in the expected model prediction towards resistance, while negative SHAP values indicate a change towards susceptibility. The plot is based on the ML model with all selected features included.

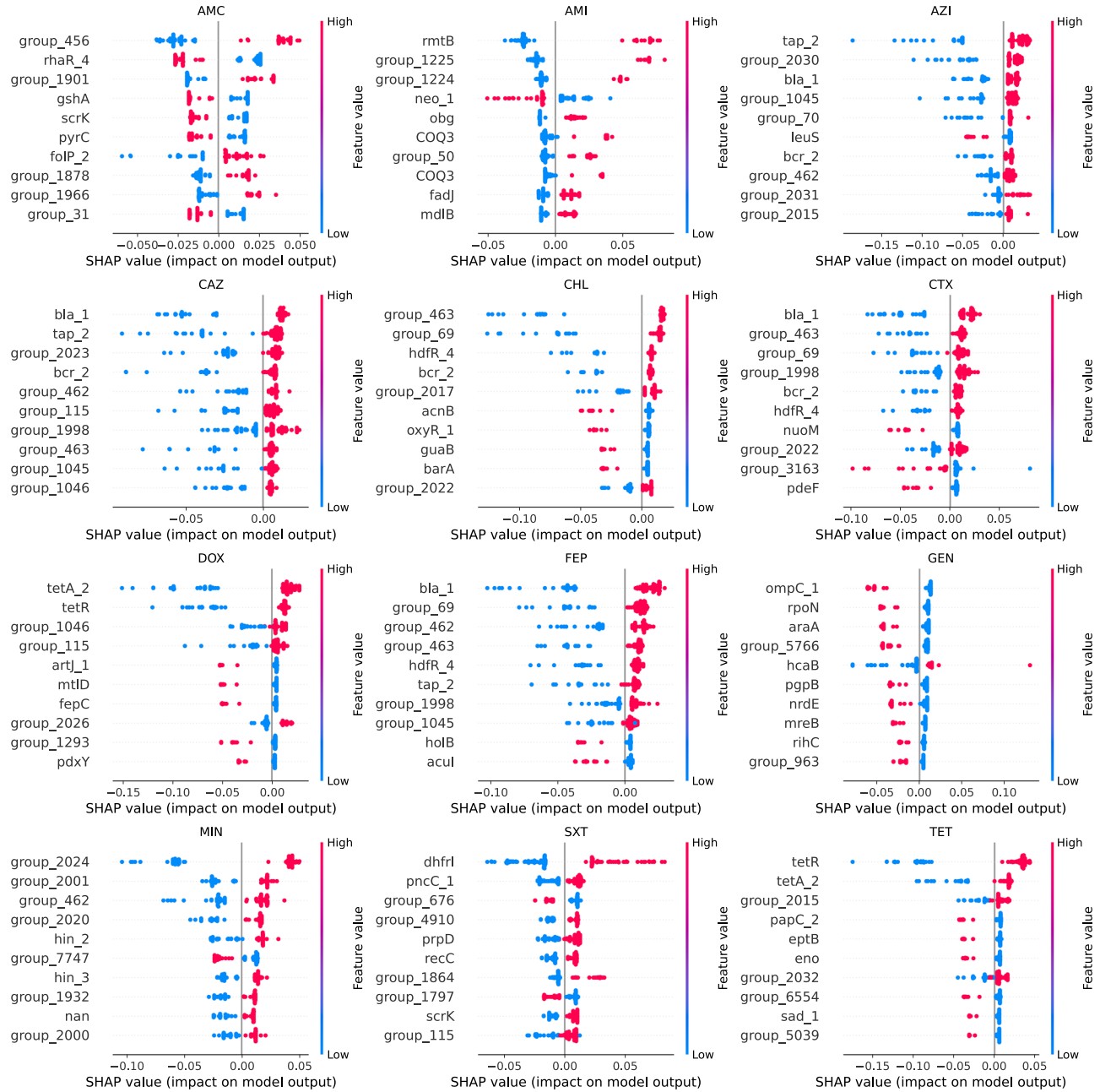

**Fig. 5 | Bee swarm plot of SHAP-calculation for the ten highest ranking genes for each of the *S. enterica* antibiotic ML models.** Genes are sorted by their mean absolute SHAP value in descending order with genes carrying most important features (SNP or accessory gene) at the top. Genes labels as 'group_' are unannotated genes. Each dot corresponds to one isolate in the study. The colour red indicates the presence of the feature while the colour blue indicates its absence. The bee swarm plot shows how the feature in each isolate affects the prediction of the ML model towards resistance to the respective antibiotic. Positive SHAP values indicate a change in the expected model prediction towards resistance, while negative SHAP values indicate a change towards susceptibility. The plot is based on the ML model with all selected features included.

whilst for *S. enterica* proportionally more metabolic genes were found, 28% (*n* = 24, mean across 12 antibiotic models). To investigate the importance of these metabolic genes to the bacteria we simulated 'loss of function' mutations for each gene using the GSM models. For *E. coli* 22 knockouts of 22 genes were predicted to be lethal in rich environmental conditions (*luxS, accD, hemE, dxs, ubiD, coaE, ispB, aroC, ispG, lptG, ribC, ispA, lpxD, waaA, murA, kdsC, cysG, folC, psd, hemD, mraY, ftsI. yrbG, hemL, ubiA*). The gene *lptG* is an ABC transporter and was selected in 15 of 19 antibiotic ML models, so it may be an important drug target for multidrug resistance. Similarly, the genes *dxs* and *ubiD* were selected in ten ML models. For *S. enterica* only ten genes were essential in rich media (*dfp, fepD, ribF,*

*fepB, luxS, hemB, aroA, fepC, entF, mrsA*) with three of those from the *fep* gene operon, the primary iron import transporter in *S. enterica*, important for virulence[55].

To investigate the system level effect of each gene on metabolism, beyond essentiality, we performed a flux balance analysis. In particular, we knocked out individual genes, blocking the flux through reactions associated with that specific gene and evaluated the change in flux span as a result of the gene knockout. In doing so, we can infer potential metabolic adaptation mechanisms that can be linked to a change in gene function (i.e., downregulation, overexpression, or deletion) and find clusters of genes affecting the same reaction pathways, which may indicate metabolic adaptions to antibiotic stress.

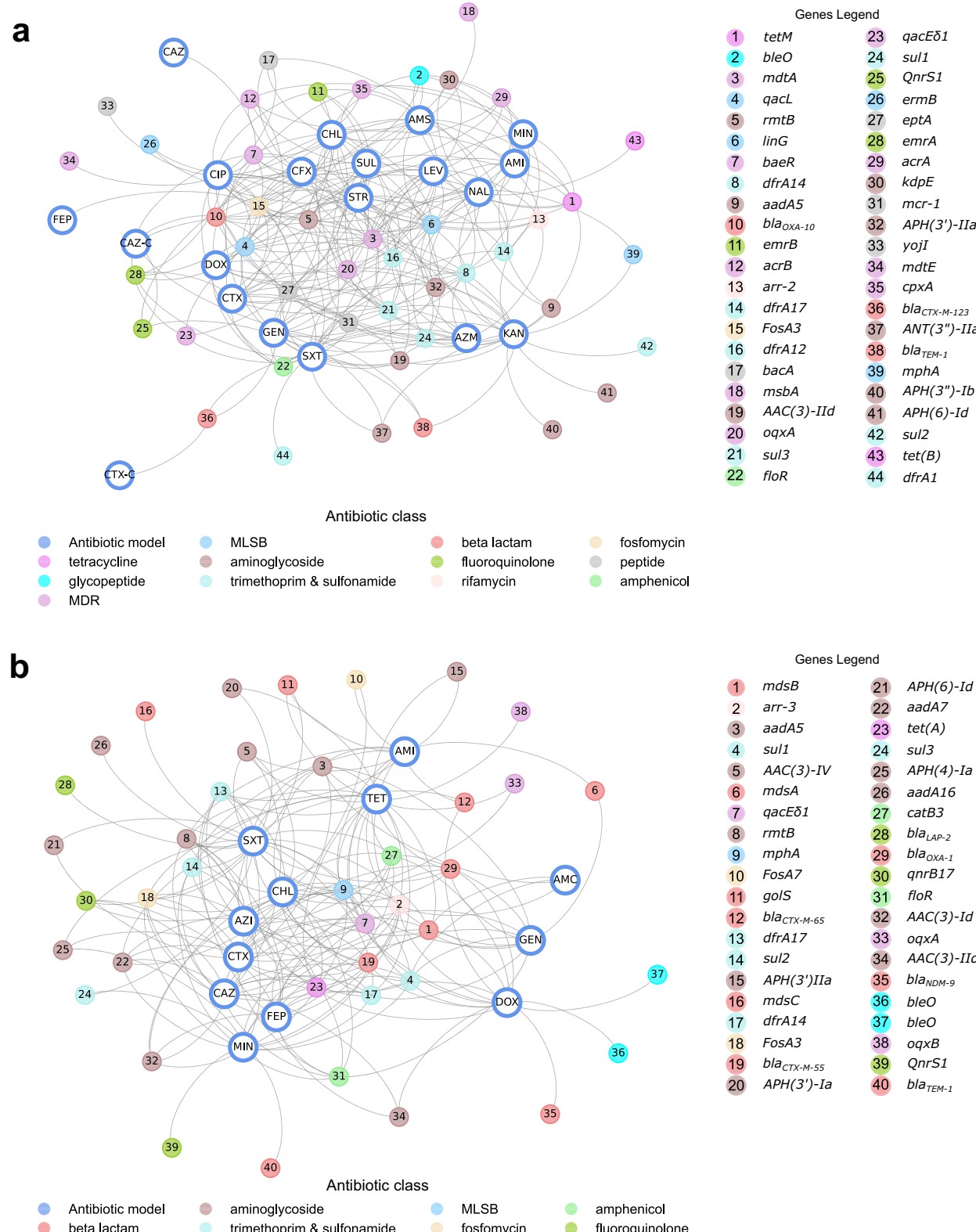

**Fig. 6 | Known AMR genes predictive of resistance in antibiotic models.**
**a** Undirected graph of genes found by ML as predictive of resistance-susceptibility profiles to a panel of 21 antimicrobials in *E. coli* and present in public AMR gene databases connected to the antibiotic model in which they were selected. Nodes of the graph represent either the antibiotic model (indicated by three letter antibiotic abbreviation) or the known AMR gene. Gene nodes are colour-coded by antibiotic class. Edges of graphs connect antibiotic model nodes to gene nodes and are unweighted. **b** Undirected graph of genes found by ML as predictive of resistance-susceptibility profiles to a panel of 12 antimicrobials in *S. enterica* and present in public AMR gene databases connected to the antibiotic model in which they were selected. Nodes of the graph represent either the antibiotic model (indicated by three letter antibiotic abbreviation) or the known AMR gene. Gene nodes are colour-coded by antibiotic class. Edges of graphs connect antibiotic model nodes to gene nodes and are unweighted.

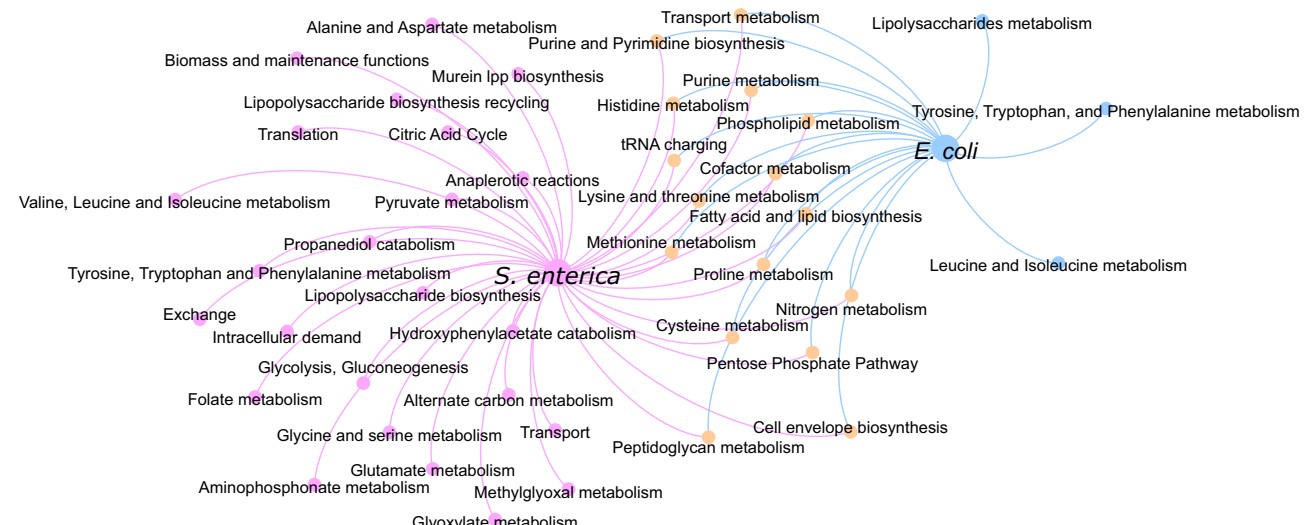

**Fig. 7 | Flux balance analysis reveals the effects of AMR-conferring genes on metabolite yields and reaction fluxes and shows a great overlap of *E. coli* and *S. enterica* pathways enriched with genes correlated to AMR.** Undirected graph network showing, for both the *S. enterica* GSM model (STM v1.0) and the *E. coli* GSM model iML1515, pathways enriched with the AMR-associated genes that, when knocked out, significantly affected the reaction fluxes throughout the pathway. Significance was tested using hypergeometric enrichment tests (two-tailed) with the false discovery rate (FDR) threshold less than 1%. The pathway nodes (small circles) from each model are connected to the GSM model by unweighted edges. Pink nodes were significantly affected in the *S. enterica* model only, blue nodes in the *E. coli* model only and orange nodes in both models.

Pathway enrichment on those genes significantly affecting the reaction fluxes revealed a number of significant pathways in each species and a core set of pathways significantly effected in both species (Fig. 7). The latter involving pentose phosphate pathway, cofactor metabolism, cysteine metabolism, histidine metabolism, lysine and threonine metabolism, tyrosine, tryptophan and phenylalanine metabolism, cell envelope biosynthesis, phospholipid metabolism, purine and pyrimidine biosynthesis, purine metabolism, nitrogen metabolism, methionine metabolism, proline metabolism, peptidoglycan metabolism, transport metabolism, and tRNA charging (Fig. 7).

Next, we considered which genes were responsible for all significantly affected pathways in the flux balance analysis, Fig. 7. In *E. coli* many genes were found to be reducing the flux span in each pathway, Fig. 8a, including a large number of genes in the purine and histidine pathways, highlighting the potential importance of these as potential gene targets. To understand whether these pathways are consistently important across the phylogenetic distribution of samples we looked at whether the mutations identified with the ML in the AMR-associated genes and correlated to each pathway were present in every *E. coli* isolate, Fig. 8b. Interestingly, with the exception of a small group of isolates within phylogroup E, all *E. coli* isolates had accessory genes and/or mutations in core genes present in every significant pathway indicating that these AMR-associated genetic determinants are broadly present across the *E. coli* phylogeny. However, the lack of mutated genes/accessory genes in phylogroup E is relevant as this phylogroup is associated with human pathogenic *E. coli* including the highly virulent O157:H4[56].

For *S. enterica*, many of the same pathways as for *E. coli* were significantly affected, with 16 genes on which AMR-associated mutations were selected from both *E. coli* and *S. enterica* ML predictive models Fig, 9a. However, the genes within these pathways carrying the AMR-associated mutations showed different phylogenetic patterns to those selected by ML in *E. coli*, Fig. 9b. When examining individual isolates, Fig. 9b reveals a distinct pattern of AMR feature presence and absence in *S. enterica* phylogeny. Among the various serotypes, only the Enteritidis serotype isolates exhibit the ML-selected AMR features across all pathways, whereas the remaining serotypes display features associated to a considerably smaller number of metabolic pathways.

This indicates that the interplay between metabolism and AMR in *S. enterica* is influenced by serotype specificity. In summary, a high number of the same metabolic pathways are correlated to AMR in both *S. enterica* and *E. coli*, with genes correlated to AMR and significantly underlying metabolic function overlapping in both species. However, these AMR-associated determinants show a different phylogenetic pattern between the two species: prevalent across all metabolic pathways throughout the phylogeny of *E. coli*, whilst, in *S. enterica*, these determinants are prevalent to certain serotypes across all pathways.

To perform a preliminary validation of the AMR-related genetic elements (SNPs and accessory genes) identified by the ML pipeline, we selected three of the top-ranked SNP candidates, prioritizing the following aspects in relation to the associated genes: (i) presence in the *E. coli* and *S. enterica* co-inhabiting set; (ii) harbouring non-synonymous SNP; (c) significantly impacting reaction flux when knocked out, as highlighted by the genome-scale metabolic model; (d) availability of the knockout strain. This resulted in three genes, top-ranked by ML for the antibiotics ampicillin and doxycycline, namely: *hisA* (obtained by mapping the SNP: P109Q back onto the genome), *argI* (SNP: A153T) and *fhuB* (SNP: N47D). To our knowledge, neither of these three genes is currently present in any AMR databases.

Although knocking out the entire gene cannot replace the in vitro study of the effects of the individual mutations onto the AMR phenotype, our preliminary results showed that all three gene mutants exhibited increased antimicrobial susceptibility compared to the parental wild type (Supplementary Note 5).

## Discussion

Considerable attention has been recently given to achieving a better understanding of the AMR evolutionary dynamics in microbial communities[57] and their impact on opportunistic pathogens such as *E. coli* and *S. enterica* with high zoonotic potential and inhabiting ecological settings, where communities are interconnected and can impact resistance selection. However, to which extent these key pathogens rely on the MGEs and mutations in the core genome for the acquisition and sharing of AMR is unclear. Analogously, despite, a recognition of the role metabolism-resistance trade-offs in the evolution of AMR[45,58],

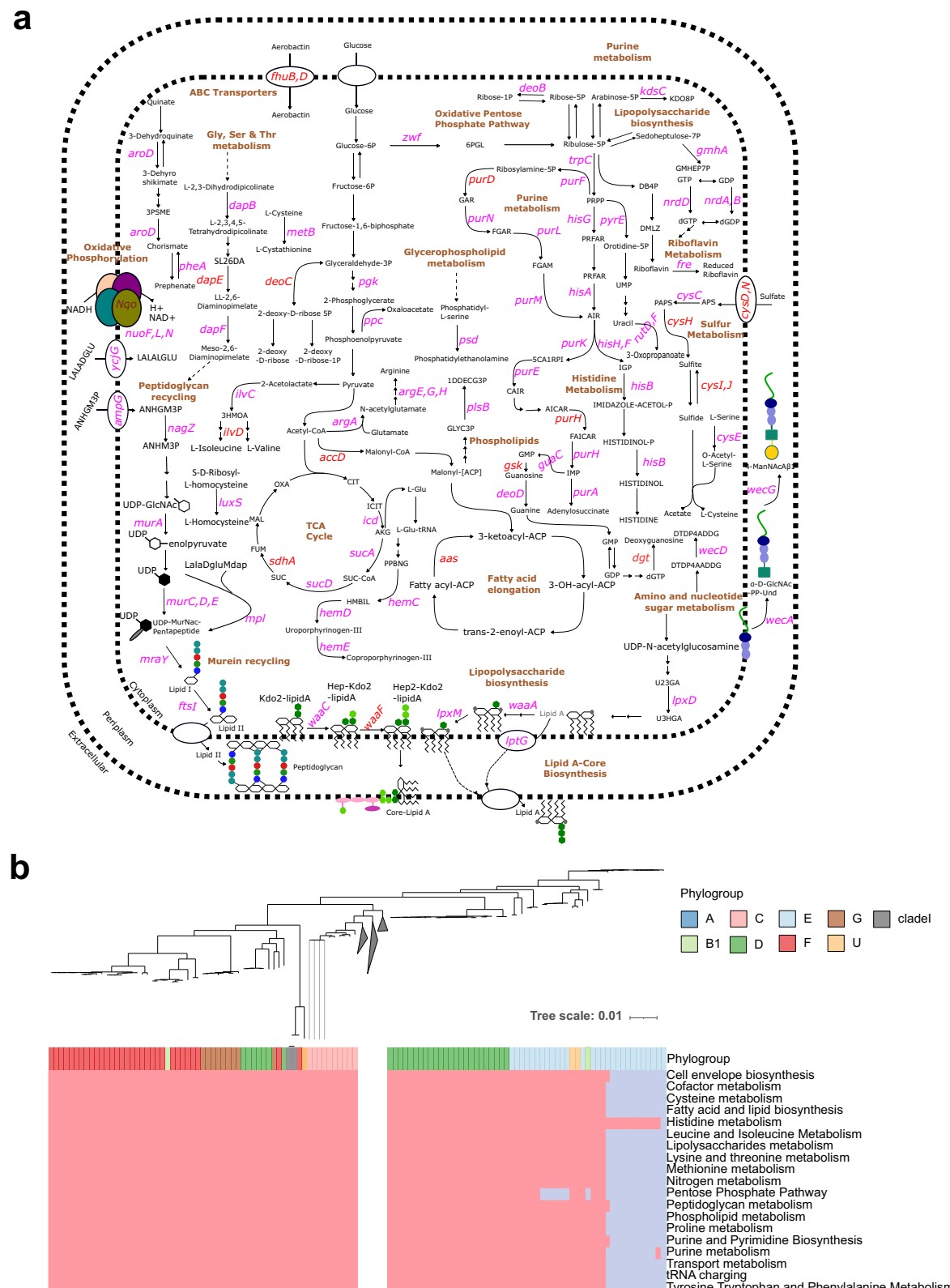

the impact such trade-off has on the co-adaptive trajectories that are most likely to be followed by species resident within same communities remains a challenge.

The gut microbiota of chickens raised in intensive farms with high antimicrobial exposure is a significant reservoir for the transmission of AMR among both resident pathogens and commensals in the animals, as well as across interconnected environmental communities

surrounding the farms. Studying antimicrobial resistance and metabolism in this ecological context presents a unique opportunity to investigate the dissemination and evolution of AMR, as well as the trade-offs between metabolism and AMR in vivo.

We anticipate that the collective evolutionary trajectory of resistance and metabolism of coexisting bacteria subject to the same antibiotic pressure, described in this work may affect resistance and

**Fig. 8 | AMR-associated genes have been found to impact numerous metabolic pathways in *E. coli*, many of them are also considered essential metabolic genes, and are prevalent across all metabolic pathways throughout the phylogeny of *E. coli*. a** An overview of the metabolic pathways affected by those AMR-associated genes significantly influencing reaction fluxes in *E. coli*. All genes annotated were found to have reduced the flux span through the metabolic system when knocked out, 11 of which were essential genes (i.e. gene knockouts were lethal in rich media). Significance was tested using hypergeometric enrichment tests (two-tailed) with the false discovery rate (FDR) threshold <1%. Genes highlighted in pink carried mutations that were selected in the *E. coli* ML models only, genes in red carried mutations that were selected in both the *E. coli* and *S. enterica* models, metabolic pathways are labelled in brown. **b** Heatmap showing, for each *E. coli* isolate (columns of the heatmap), whether the isolate contained an AMR-associated gene that if knocked out also significantly affected one or more metabolic pathways. The top part of the heatmap shows the phylogenetic relationship between each isolate. The first row of the heatmap indicates the phylogroup to which each isolate belongs. The subsequent rows indicate the metabolic pathways into which the genes were grouped, with only pathways significantly enriched with ML selected genes shown. Pink represents the presence of a mutated gene/accessory gene, blue represent the absence. For brevity, A and B1 clades, which showed presence of genes in every pathway, were collapsed.

evolutionary responses to antibiotic treatments of other bacteria inhabiting the same microbiota more broadly.

In this work, by using a data-mining approach powered by ML, Bayesian divergence analysis and GSM modelling, we showed that at a larger scale, differences in the phylogeny and evolution of *S. enterica* and *E. coli* were observed, but, at a finer scale, most isolates of each species inhabiting the same host and environment, possessed the same plasmids and MGEs carrying clinically relevant ARGs.

In particular we found IncHI2, a plasmid known to be a very important carrier of AMR genes[59] in a number of *S. enterica* and *E. coli* strains isolated from the same samples. Bayesian divergence analysis highlighted two recently co-evolved clusters of IncHI2 plasmids isolated from both species, indicative of relatively recent HGT between species which may also indicate a host adaptation process for these species. This plasmid has been previously associated with cross species horizontal transfer[60], however, to our knowledge direct transfer of this plasmid (IncHI2) between strains isolated from these two species has not previously been observed, although it has been observed in a small number of other plasmid types, e.g., IncY[61]. Analogously, we found same mobile ARG sequences (ARGs-transposons) in pairs of *E. coli* and *S. enterica* isolates collected from the same sample. Whilst the same mobile ARGs have been found in different bacterial species[62] and different sample sources[31,32], this is the first time that almost all of *E. coli* and *S. enterica* isolates collected from the same sample, showed mobile ARGs including clinically relevant ARGs (*bla*CTX-M, *APH(3)*, *floR*, *mphA*, and *qnrS1*) with a highly conserved mobile ARG structure. These resistances (beta-lactams, *bla*CTX-M, aminoglycosides, *APH(3)*, florfenicol and chloramphenicol, *floR*, macrolide, *MphA* and quinolone, *qnrS1*) are a growing problem worldwide. The fact that these antimicrobial resistance genes are found in conserved mobile genetic elements between *S. enterica* and *E. coli*, is particularly relevant because it means that they can easily be transferred between different bacterial species, accelerating the spread of resistance. A consideration arising from this result is that, probably many other species present in the same samples and interconnected communities have the same resistances. Our results indicate that sharing is influenced by many factors, and we have reported evidence on the effects of co-inhabitation, as well as country of collection, possibly reflecting differences in regulations, interventions and overall microbial ecology. It is probable that conducting large-scale analyses will be essential to consistently detect these occurrences of AMR gene transfer among strains of different species.

Antibiotic susceptibility testing done against 28 antimicrobials showed similar proportions of resistant isolates in both species for many antibiotics, in line with previous studies[11]. However, when the resistance-susceptibility profiles were analysed against the genomes of each isolate from each species through our machine learning pipeline, both known and novel genomic features (SNPs in the core genome and accessory genes) were found predictive of the AMR profiles. Comparison between the two species highlighted a common core set of genes that were linked to AMR (604 annotated genes, of which 88 were in the top 10% of most important genes correlated to AMR). In both species, accessory gene presence-absence tended to be selected in multiple AMR models, whilst SNP features originating from the core genome tended to be more specific to a single antibiotic. Overall, we observed that whilst for *E. coli* the genes mainly were positively correlated with resistance, for *S. enterica* the genes mainly were correlated with susceptibility. The genes from this core set were found to have a significant level of interaction with each other and with some known AMR genes (e.g., *gyrA*, *gyrB*, *bcr*, *oppC*)[36,51]. *OmpC* variants have been linked to multidrug resistance in *E. coli*[63] and very recently the mutations in the gene *ompC* have been suggested as being correlated to extensively drug resistant *E. coli* with the appearance of mutations preceding the acquisition of resistance[64]. Our results suggest that this gene could have similar important functions also in *S. enterica*. These findings demonstrate the efficacy of integrating whole-genome sequencing and ML in creating a streamlined approach for identifying both known and new AMR genes. Given the constantly evolving nature of AMR, it is crucial to advance techniques capable of detecting novel and emerging genetic determinants that underlie complex resistant phenotypes. By outperforming traditional methods, ML-enhanced techniques offer increased accuracy and potency.

GSM modelling further indicated a core set of same metabolic pathways which were affected by the AMR-associated genes found in both *E. coli* and *S. enterica* by the ML pipeline. Whilst several of these genes have been previously correlated to AMR, this is the first time they have been found to be linked to metabolic processes and to functionally overlap in the two species. One such pathway, tRNA charging, is essential to protein translation and recent papers have suggested that bacteria may use mistranslation to control exposure to stress caused by antibiotics[65]. This response involves mistranslation of methionine[66] and so could be linked to methionine metabolism, another pathway significantly linked to AMR in both species. In addition, cell envelope biosynthesis, pentose phosphate pathway, histidine metabolism and phospholipid metabolism have all previously been linked to potential antibiotic resistance mechanisms[45]. Of the AMR-related genes found to affect the metabolic pathways in both species there are several genes from the *cys* operon, as well as several *pur* genes. Both of these gene clusters have been previously highlighted in connection to AMR in *E. coli*[45]. In addition, genes involved in the TCA cycle were selected in each species with *sdhA* selected in both; the TCA cycle has recently been highlighted as a promising target for new antimicrobial therapies[67].

In this study, we developed a custom ML pipeline to incorporate a large number of genetic elements (SNPs in coding and non-coding regions of the core genome, as well as accessory genes) and perform a broad-spectrum search for correlations to AMR. It would be interesting to study how our results compare to applying GWAS methods instead[68–73] (see also Supplementary Note 6).

A limitation of this study is that the analysis of association of AMR resistance to genetic features was primarily in silico, with experimental gene validation restricted to a small number of genes and a knock out approach (Supplementary Note 5). To fully validate the predictions made in this study experimental validation of the specific genetic mutations would be required. It is important to note that the in silico mutations made for metabolic genes should be tested in an in vitro

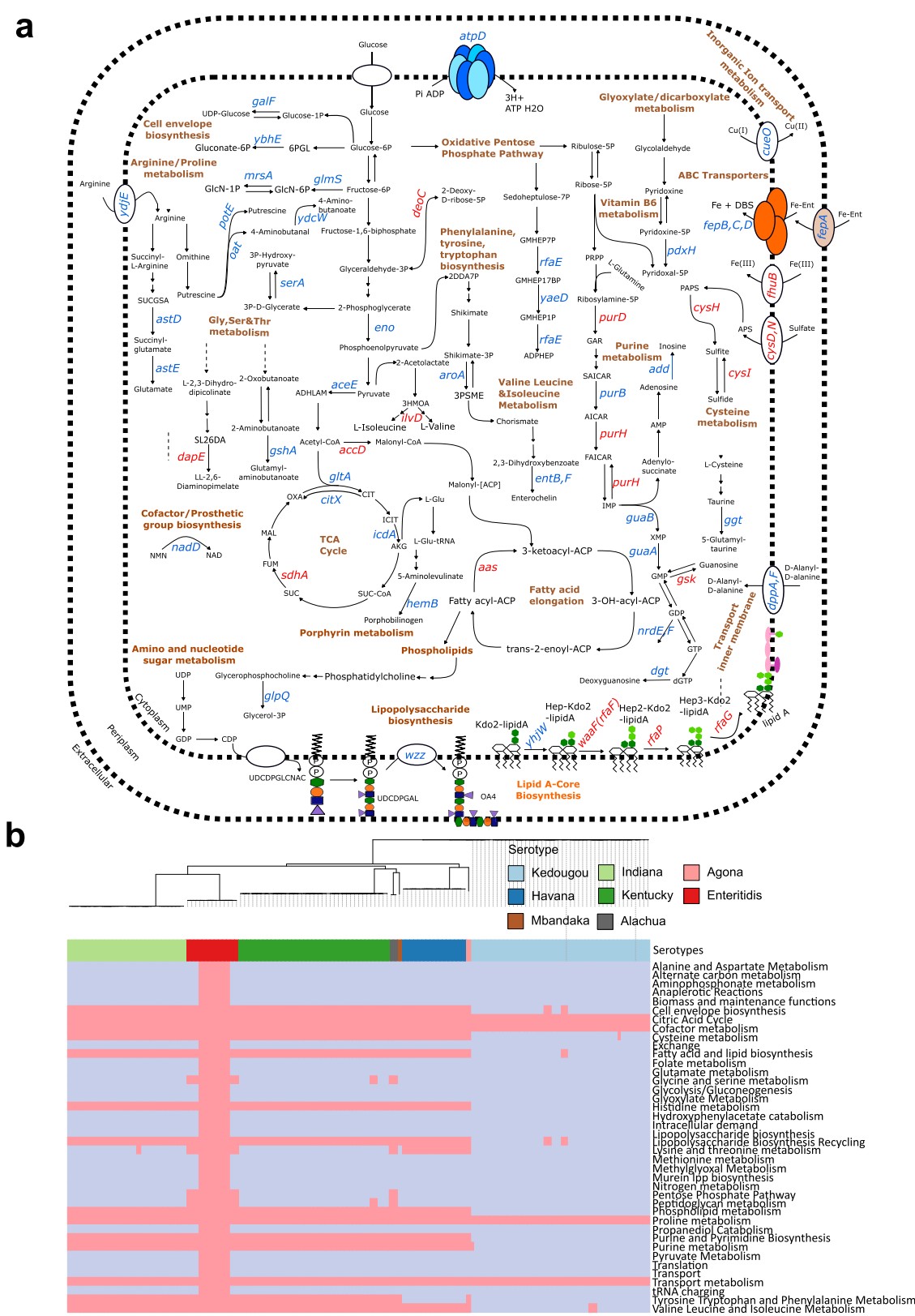

model (e.g. natural host derived and human macrophages and/or epithelial cells) in order to validate the physiological relevance of the metabolic pathway findings. In relation to the analyses done to understand the influence of co-inhabitance as well as country of collection in the observed results (genetic elements tied to AMR), another limitation of our study is intrinsic to the challenge of accounting for the many further confounding factors which may have influenced our

results. Further insight is given in Supplementary Note 6. One further limitation of the results is the close genomic homology of *E. coli* and *S. enterica*, which may have contributed to some of the overlapping AMR-associated genes selected by the machine learning and metabolic modelling. Future studies should consider comparisons between co-inhabiting gut bacteria species less closely related, for example *Enterococcus spp.* or *Campylobacter spp.*

**Fig. 9 | AMR-associated genes have been found to impact numerous metabolic pathways in *S. enterica*, many of them are also considered essential metabolic genes, however, are prevalent across all pathways only in certain serotypes in the *S. enterica* phylogeny. a** An overview of the metabolic pathways affected by those AMR-associated genes significantly influencing reaction fluxes in *S. enterica*. All genes annotated were found to have reduced the flux span through the metabolic system when knocked out, six of which were essential genes (i.e. gene knockouts were lethal in rich media). Significance was tested using hypergeometric enrichment tests (two-tailed) with the false discovery rate (FDR) threshold <1%. Genes highlighted in blue were selected in the *S. enterica* ML models only, genes in red were selected in both the *E. coli* and *S. enterica* models, metabolic pathways are labelled in brown. **b** Heatmap showing, for each *S. enterica* isolate (columns of the heatmap), whether the isolate contained an AMR-associated gene that if knocked out also significantly affected one or more metabolic pathways. The top part of the heatmap shows the phytogenic relationship between each isolate. The first row of the heatmap indicates the phylogroup to which each isolate belongs. The subsequent rows indicate the metabolic pathways into which the genes were grouped, with only pathways significantly enriched with ML selected genes shown. Pink represents the presence of a mutated gene/accessory gene, blue represent the absence.

Overall, this work has shown the importance and the power, as also suggested by others[57], of adopting bespoke analytical methods when studying how bacteria coexisting in multi-species communities respond to antibiotics. Furthermore, it also emphasizes the need to explore not only individual pathogens but also the intricate microbial ecosystems they inhabit and evolve in.

## Methods

### Ethics statement

This study was performed in accordance with protocols approved by the Ethics Committee of the State Key Laboratory of the China National Centre for Food Safety Risk Assessment (CFSA). The ethical approval number of CFSA is 2018018. Ethical approval has also been obtained from the Research Ethics Committee in the School of Veterinary Medicine and Science at the University of Nottingham, application ID: 2340 180613.

### Experimental design

For the study we selected ten large-scale commercial poultry farms belonging to three different provinces in China (Shandong, Henan and Liaoning), covering an area of 472,500 km², each farm feeding into one of four regional abattoirs (two in Henan, one in Liaoning and one in Shandong). Each farm features multiple barns, each containing between 12,000 and 32,800 birds, leading to a total production capacity of 110,730 to 380,000 birds per breeding cycle (depending on farm). Broiler production is based on self-breeding with broilers bred on the farm and moved to the barns in same-aged batches. Of the ten selected farms, four (three in Liaoning and one in Shandong) use net housing systems, whilst the other six use cage housing systems. During collection, the number of birds per barn did not significantly differ between the two housing systems ($t$-test, $p$-value = 0.07). *E. coli* isolates ($n = 96$) taken from one farm (Shandong 1) were part of a pilot experiment with the results previously published[17].

In total 977 samples were collected from: chicken faeces ($n = 372$), chicken cecal ($n = 52$), chicken anal swabs ($n = 10$), chicken feathers ($n = 65$), chicken carcasses ($n = 193$), feed ($n = 64$), barn environment swabs ($n = 32$), drinking water ($n = 58$), abattoir environment swabs ($n = 15$), wastewater ($n = 32$), external soil ($n = 84$). Details on the collection methods are as follows. For faecal and cecal samples, each sample consisted of ~10 g fresh sample of mixed chicken faeces (2–3 chickens) or an ~2 g mixed fresh sample of 2–3 chicken caecal droppings, collected from the bottom of the chicken cage/net using a sterilized spoon. Chicken anal swab samples were collected using cotton-tipped swabs. Feather samples were collected from the bottom of the chicken cage/net and swabbed using cotton tipped swabs. Barn floor samples were taken using a sterilized spoon. Not less than 20 mL each of chicken drinking water from farm and waste effluent water from the slaughterhouse were collected from the water pipe or by pipettes. About 10 g each feed sample was collected using a sterilized spoon. Pooled carcass samples were collected in the abattoirs using a sponge swab (SS100NB, Hygiena International, Watford, UK) on the surface of the carcass. Abattoir environment samples were taken from the processing and collected from multiple surfaces, e.g., the cutting table and transfer belt of the cutting and deboning house. Soil samples consisted of about 10 g soil, collected outdoors at depth of 1–3 cm, 5 m from the external barn walls, to ensure sufficient separation from areas of human use. All biological samples were collected using aseptic techniques, and then stored in secure containers at 4 °C during transportation to the laboratory and extracted within 24 h.

### Bacterial isolation and identification

A quantity (volume) of 1 g (mL) sample of faeces, soil, feed, water each was vortexed with 9 mL of sterile buffered peptone water tube (BPW; Luqiao Inc., Beijing, China) for 1 min; chicken carcass sponge samples were homogenised with 10 mL BPW for 1 min in a stomacher bag. For isolation of *E. coli*, ~1 mL dilution (of any of the above samples) was then added to 9 mL *E. coli* (EC) broth (Luqiao Inc.) and incubated at 37 °C for 16–20 h in order to enumerate presumptive *E. coli* populations. A loopful of these solutions was then streaked onto an eosin-methylene blue (EMB) agar and MacConkey (MAC) Agar (Luqiao Inc.) and incubated at 37 °C for 18–24 h. Typical *E. coli* colonies were screened and subsequently characterized by a Bruker MALDI Biotyper (Germany). The positive isolates identified were further confirmed by PCR using *E. coli*-specific primers ITS-F (5′-CAATTTTCGTGTCCCCTTCG-3′) and ITS-R (5′- GTTAATGATAGTGTGTCGAAAC-3′)[74]. Thermal amplification conditions were as follows: pre-incubation at 94 °C for 5 min, followed by 30 cycles of denaturation at 94 °C for 30 s, annealing at 55 °C for 30 s, elongation at 72 °C for 30 s, and a final extension of 72 °C for 5 min. Out of all isolates analysed, 518 were identified as *E. coli* positive and included: chicken faeces ($n = 261$), chicken carcasses ($n = 67$), chicken feathers ($n = 53$), chicken cecal droppings ($n = 45$), chicken feed ($n = 30$) external soil ($n = 16$), barn environment ($n = 16$), waste water ($n = 12$), anal swabs ($n = 9$), abattoir environment ($n = 7$) and drinking water ($n = 2$). Similarly, for isolation of *S. enterica*, ~1 mL enriched dilution was added to 9 mL RV broth and SC broth (Luqiao Inc.) and incubated at 42 °C and 37 °C for 16–20 h, respectively, in order to enumerate presumptive *Salmonella* populations. A loopful of these solutions was then streaked onto a Chromogenic Salmonella agar (Luqiao Inc.) and incubated at 37 °C for 24–48 h. Typical *Salmonella* colonies were screened and subsequently characterized by Bruker MALDI Biotyper (Germany). The positive isolates identified were further confirmed by PCR using *Salmonella* specific primers invA-F (5′-GTGAAATTATCGCCACGTTCGGGCAA-3′) and invA-R (5′-TCATCG-CACCGTCAAAGGAACC-3′)[75]. Thermal amplification conditions were as follows: pre-incubation at 94 °C for 5 min, followed by 30 cycles of denaturation at 94 °C for 30 s, annealing at 55 °C for 30 s, elongation at 72 °C for 30 s, and a final extension of 72 °C for 5 min. In total, 143 were identified as *S. enterica* positive and included: chicken faeces ($n = 59$), chicken cecal droppings ($n = 23$), chicken carcass ($n = 20$), chicken feather ($n = 13$), barn environment ($n = 8$), wastewater ($n = 6$), external soil ($n = 5$), chicken feed ($n = 4$), abattoir environment ($n = 4$), drinking water ($n = 1$). The identified *E. coli* and *S. enterica* isolates were kept in brain heart infusion broth (BHI) medium with 20% glycerol at −80 °C freezer for further characterization.

### Antimicrobial susceptibility testing

Antimicrobial susceptibility to a panel of agents was determined by broth microdilution and interpreted according to the criteria based on

the Clinical & Laboratory Standards Institute (CLSI) interpretive criteria (CLSI 2009). The resistance/susceptibility of 28 and 26 antimicrobial compounds were measured for each of the *E. coli* and *S. enterica* isolates, respectively, ampicillin (AMP), ampicillin/sulbactam (AMS), tetracycline (TET), chloramphenicol (CHL), trimethoprim/sulfamethoxazole (SXT), cefazolin (CFZ), cefotaxime (CTX), cefotaximecClavulanic acid (CTX-C – *E. coli* only), ceftazidime (CAZ), ceftazidime/clavulanic acid (CAZ-C – *E. coli* only), cefoxitin (CFX), gentamicin (GEN), imipenem (IMI), nalidixic acid (NAL – *E. coli* only), azithromycin (AZI – *S. enterica* only), sulfisoxazole (SUL), ciprofloxacin (CIP), amoxycillin/clavulanic acid (AMC), polymyxin E (CT), polymyxin B (PB), minocycline (MIN), amikacin (AMI), aztreonam (AZM), cefepime (FEP), meropenem (MEM), levofloxacin (LEV), doxycycline (DOX), kanamycin (KAN) and streptomycin (STR). *E. coli* ATCC 25922 was used as a control for the antimicrobial susceptibility testing.

### DNA purification and extraction
All the *E. coli* and *S. enterica* isolates were subjected to genomic DNA extraction in accordance with the manufacturer's protocol of E.Z.N.A. Bacterial DNA Kit (Omega Bio-Tek, Norcross, GA, USA) followed DNA sequencing on an Illumina Hiseq 2500 PE150 platform (Illumina, San Diego, CA, USA).

### Library construction and whole-genome sequencing
The template genomic DNA was fragmented by sonication to an insert size of 350 bp using NEBNext Ultra DNA Library Prep Kit for Illumina (NEB, USA) following the manufacturer's recommendations and index codes were added to attribute sequences to each sample and sequenced using an Illumina Hiseq 2500 PE150.

### Genome assembly and annotation
All sequences were pre-processed through readfq v10[76]. To clean the data, reads containing low-quality bases (mean quality value ≤ 20) over 40% were removed. Reads with greater than 10% unidentified bases (N) were removed as well as the adapters. The whole-genome shotgun sequencing produced high-quality reconstructed genomes with a N50 larger than 50,000 and less than 250 contigs. Cleaned data were processed for genome assembly with SPAdes v3.13, and QUAST v4.5 was used for assessing the assembly. The contigs with length shorter than 500 nucleotides were filtered out. The completeness and contamination of genomes were assessed through checkM with the lineage_wf pipeline. Genomes were annotated with Prokka v1.14.5[77] using default parameters with−addgenes−usegenus.

### Screening of annotated genes against ABR databases, plasmid databases and in silico subtyping
The whole-genome sequences were screened against the CARD[36] database with a minimum coverage of 70% and minimum identity of 90% to identify known AMR-associated genes in the isolate cohort. In addition the annotated genes obtained from Prokka v1.14.5[77] were screened against the CARD[36], ARG-Annot[78] and Resfinder[79] databases using Abricate[80] and the NCBI AMRfinder[81] database; all the comparative analyses have been done with a minimum coverage of 70% and identity of 80%. Plasmids screening was conducted using the PlasmidFinder[82] database in Abricate[80]. Sequence types for both species were identified through MLST[83] which mapped the sequences to the PubMLST[84] database. For *E. coli* Phylogroups were identified using in silico Clermont typing[85]. Serotypes were identified through the EcOH database[86] using Abricate[80]. For *S. enterica*, serotypes were identified with SeqSero2[87].

To compare plasmid presence with the level observed previously we downloaded from BV-BRC[88] (accessed 9th March 2023), all the good quality *E. coli* (n = 705) and *S. enterica* (n = 265) isolates collected from chicken in China (Supplementary Data 11) and conducted plasmid screening on these using the PlasmidFinder[82] database in Abricate[80].

Jaccard/Tanimoto similarity coefficients were calculated in a pairwise manner between the presence of resistant AST profiles for each antibiotic and the presence of known AMR genes as found in CARD. The Jaccard coefficient was calculated as

$$\frac{|A \cap B|}{(|A| + |B| - |A \cap B|)} \quad (1)$$

where $A$ is a resistant phenotype and $B$ is the presence of the AMR gene. A Jaccard value of 1 represents perfect intersection and 0 represents no intersection. The significance of the Jaccard coefficient was tested statistically using an MCA approach[89] and Jaccard coefficients with an FDR adjusted $p$ value < 0.05 were used[42].

### Population structure analysis
Linkage disequilibrium of the 518 *E. coli* and 143 *S. enterica* isolates in our cohort was evaluated using the standardized index of association ($I^S_A$)[16], which estimates the homologous recombination for the cohorts by assessing the linkage disequilibrium among the seven MLST loci. For each species separately the LIAN Ver. 3.7 program was used to calculate the $I^S_A$ for all the isolates and for a subset of them (one isolate for each ST type) from the ratio of the variance of observed mismatches in the test set ($V_D$) to the variance expected for a state of linkage equilibrium ($V_e$), scaled by the number of loci used in the analysis (L)[16],

$$I^S_A = \frac{1}{L-1}\left(\frac{V_D}{V_e} - 1\right) \quad (2)$$

The significance of $I^S_A$ was determined by a Monte Carlo simulation with $10^5$ resampling.

### Generation of genetic features input files
For each species, all annotated genomes were taken as input for pangenome analysis with core gene alignments through Roary v3.13[90]. For hypothetical proteins, assigned with group IDs by Roary, we identified putative gene names via a BLAST search of the GenBank and SwissProt databases at 80% identity.

The core genome alignment for each species was taken as input to produce a file of core gene SNPs present in the cohort using snp-sites[91]. A presence-absence table of accessory genes was generated from the gene presence-absence Rtab file produced in Roary[90], with genes considered to be accessory genes if present in less than 99% of isolates (and hence not included the core genome). To map intergenic SNPs, for each species an alignment of core intergenic regions was created using Piggy v1.5[92]. As input all the output files from Roary[90] and gff files for each isolate were used. An alignment of the intergenic clusters was generated by Piggy v1.5[92] and SNPs were called from this alignment using SNP-sites.

### Network analysis based on core genome SNPs
Networks of *E. coli* and *S. enterica* isolates collected from different samples sources in the farms and different regions of China were created using a pairwise hamming distance comparison based on core genome SNPs. Each node represents an isolate, while the edge represents the hamming distance between two isolates multiplied by the total number of SNPs found in our cohorts (215,224 SNPs in *E. coli*; 96,135 SNPs in *S. enterica*). A threshold of 15 or less SNPs difference was used to filter the edges in the network as suggested by Ludden et al. (2019)[93] and used by us previously[17]. To consider intra- and intercluster variability across geography and time a threshold of <100 SNPs was applied as done previously[94] and this subset of isolate pairs was plotted using Python (Matplotlib v3.6.2[95]). Statistical comparisons were made using the SciPy package[96] implementing ANOVA tests with post hoc Tukey comparisons.

## Whole-genome phylogenetic analysis and Bayesian evolutionary analysis

IQTree v2.1.4-beta[97] was used to construct the maximum-likelihood phylogenetic trees from the core genome alignment. The alignment length of the *E. coli* core genome was 2,214,946 nucleotide sides of which 198,217 were informative. For the *S. enterica* core genome the alignment length was 3,489,080 nucleotide sites with 87,084 informative sites. For both *E. coli* and *S. enterica* different nucleotide replacement models were tested automatically with the GTR (+F + R3) replacement model selected as the best model. The Ultrafast bootstrap algorithm was used with 10000 replicates to assess branch support. The phylogenetic trees were subsequently visualised through iTOLv5[98]. Subsets of sequences from this study from individual phylogroups (*E. coli*) or serotypes (*S. enterica*), were selected for Bayesian evolutionary analysis using BEAST v 1.10.4[99]. Analysis was conducted on a core genome alignment of each lineage by using Roary v3.13[90]. The alignment lengths varied between 510,551 and 4,261,565 and can be found in Supplementary Data 2. BETS analysis[100] was used to check for a temporal signal in each serotype/phylogroup and found temporal signal in five *S. enterica* serotypes (Enteriditis, Indiana, Kentucky, Kedougou, Havana) and three *E. coli* phylogroups (A, E and F), which were taken further for analysis in BEAST, Supplementary Table 1. The BETS analysis showed that there was insufficient temporal signal in *E. coli* Clades B1 and D, so these clades were not subjected to BEAST analysis. For each species, all combinations of three clock models (strict, uncorrelated log normal, and uncorrelated exponential) and four tree priors (constant coalescent, logistic growth, Bayesian skyline, and birth-death model) were tested using steppingstone sampling on a subset of the isolates to identify the best model. The GTR-gamma nucleotide substitution model was used, as selected for the maximum likelihood tree. The analysis was run for three independent chains until the effective sample size (ESS), that is, the effective number of independent draws from the posterior distribution, for all parameters was greater than 200 per chain. Convergence was assessed in Tracer v1.7.1[101], and chains were subsequently combined using LogCombiner v1.10.4[102]. The maximum clade credibility tree was selected using TreeAnnotator v1.10.4[102] and then visualized in iToL v5[98].

## Plasmid reconstruction and evolutionary phylogeny

Plasmids were reconstructed using the MOB-recon algorithm in the MOB-suite package v 3.1.2[26], using default settings. For a subset of IncH2 plasmids, found in both species in a single sample, variants were called against the plasmid reference R478 (GenBank accession no. BX664015) and a core SNP alignment was generated in snippy v.4.4.5[103]. An *E. coli* poultry-sourced IncHI2 plasmid sequence, pAPEC-o1-R (Genbank DQ517526.1) was also included as an outgroup for the phylogeny. The core SNP alignment was used as input for Bayesian evolutionary analysis using BEAST v 1.10.4[99].

To select the best BEAST model all combinations of three clock models (strict, uncorrelated log normal, and uncorrelated exponential) and three tree priors (constant coalescent, logistic growth, and Bayesian skyline) were tested using path sampling to identify the best model. Log marginal likelihood values were in the range of −411,744 to −403,646. The best model was a random uncorrelated exponential clock model, with a constant coalescent growth model. The GTR-gamma nucleotide substitution model was used, for nucleotide substitution. Using the chosen model, the analysis was run for three independent chains of 100 million steps. Convergence was assessed in Tracer v1.7.1[101], and chains were subsequently combined using LogCombiner v1.10.4[102]. The maximum clade credibility tree was selected using TreeAnnotator v1.10.4[102] and then visualized in iTOL v5[98]. In addition to the phylogroup analyses, the BEAST analysis was also conducted for two *E. coli* serotypes, O83:H42 and O8:H16, using the same models used for the phylogroup analyses.

## Mobile ARG analysis

To look for the presence of shared mobile ARG content across different sources, ARGs carried by both species were considered for samples where both species were found in a single sample. Filtered contigs (>500 bp) in each assembly were searched for ARGs and MGEs using a BLASTn[104] search against the CARD[36] and ISfinder[105] databases using a high identity (90%) and coverage (90%) to prevent false positives and variant uncertainty[106]. The distance between them ARG and MGE was calculated based on the position of the ARG and MGE in the contig[31]. ARG carrying contigs with a distance between ARG and MGE of >5 kb were discarded[31,33], with the remaining contigs classed as mobile ARGs. Contigs were annotated using Prokka 1.14.6[77]. ARGs were further classified as clinically important if the ARG was included in the Risk I (clinically important ARGs dataset) according to Zhang et al.[37]. The structure for the mobile ARG patterns (the MGE type, ARG carried, MGE carried, sample source, farm, and distance) was summarised. For the ARG *qnrS1*, the gene structure was visualised using EasyFigv2.2.5[107].

## Machine learning analysis

Machine learning methods were used to search for the features in the genome sequence of *S. enterica* and *E. coli* isolates which could strongly correlate to resistance to each one of the of the 26 and 28 selected antimicrobials, respectively. The AMR phenotype (resistant, susceptible) of each sample was used as the class label with intermediate phenotypes neglected. One ML pipeline was proposed for each one of the studied species. In both cases, to correct for the population structure a weighted pairwise chi-squared tests between each feature and the phenotype class was used, as suggested by Aun et al.[108] using each feature class (accessory genes, core genome SNPs and intergenic region SNPs) individually. The weights of each genome were calculated using the method of Gerstein, Sonnhammer, and Chothia[109]. As the classes were unbalanced, we oversampled the minority class as a pre-processing step for *E. coli* using a Synthetic Minority Over-sampling Technique approach (SMOTE)[47] and during the training phase for *S. enterica* using a SMOTE and Edited Nearest Neighbours (SMOTEENN) approach[110] to balance the proportion of classes in the data set. The Python package Scikit-learn version 1.2.1[111] was used to make the classification and to select the most important features.

The overall data analysis pipeline consisted of two phases:

- Phase I – WGS Whole-genome sequence features preselection: For each antibiotic, isolation of a first set of WGS features (i.e. presence/absence of accessory genes, core genome SNPs and intergenic region SNPs) was conducted. For the *E. coli* pipeline the features were selected based on the correlation with the resistance/susceptibility profiles of *E. coli* using a chi-square test ($p$-value $< 10^{-5}$) and followed by further selection based on the Gini feature importance of an ExtraTree Classifier with 50 estimators. For the *S. enterica* pipeline, the features were selected based on the Gini feature importance of an ExtraTree Classifier with 50 estimators. In both cases the features were selected if their Gini importance was higher than the overall mean value.

- Phase II - Assessment of feature predicting-power through the development of ML-powered predictive functions: a panel of machine learning methods (logistic regression (LR), linear support vector machine (L-SVM), radial basis function support vector machine (RBF-SVM), extra tree classifier, random forest, adaboost and xgboost) were then run using as input the pre-selected features uncovered on the first step and their performances were evaluated based on a nested cross validation.

Nested Cross-validation (NCV)[112] was employed to assess the performance and select the hyper-parameters of the proposed classifiers. NCV consists of an outer loop dedicated to randomly split the

data into new training and testing sets, and an inner loop where different configurations (sets of hyperparameters) for the predictive function are tested with the outer loop training and testing sets. In our analysis the inner loop of the NCV used a stratified threefold cross-validation; while the outer loop measured the ROC-AUC (receiver operating characteristic area under the curve) accuracy, sensitivity, specificity and Cohen's kappa of the test data set (unseen in the inner loop for the training) using fivefold stratified cross-validation. Thirty iterations were carried out, wherein each iteration an NCV was employed.

To compare the results obtained by the seven different classifiers used, a Friedman Statistical F-test ($F_F$) with Iman-Davenport correction was employed for statistical comparison of multiple[113]. The classifiers for each dataset were ranked separately based on their AUC performance, i.e., the classifier with the highest AUC gets ranking 1, the second highest AUC gets ranking 2, and so on. In case of ties, average ranks are assigned. The $F_F$ test was applied, and the null hypothesis was rejected. The post-hoc Nemenyi test[113] was used to find if there is a single classifier or a group of classifiers that performs statistically better in terms of their average rank after the $F_F$ test has rejected the null hypothesis that the performance of the comparisons on the individual classifiers over the different datasets is similar.

Undirected graphs were created using NetworkX[114] to visualize the interconnected whole-genome sequence features (accessory genes, core genome SNPs and intergenic region SNPs) with the antibiotic models where they were selected. In addition, undirected graphs were created to visualize the interconnected ARGs (based on the selected features) and the antibiotics where they were selected.

### Genome scale metabolic model

The cobra toolbox in python was used for all simulations. The models iML1515[53] of *E. coli* K-12 MG1655 strain and STM v1_0[54] of *S. enterica* subsp. enterica serovar Typhimurium str. LT2 were downloaded from the BiGG database[115] using the cameo python toolbox[116].

Flux variability analysis (FVA) was applied to the wild-type model and each knockout model using the cobra toolbox in python[117]. FVA calculates the minimum and maximum flux through each reaction in the model, given a set of constraints, resulting in the range of possible fluxes for each reaction (flux span). FVA was simulated using glucose as the only carbon source in aerobic minimal M9 medium conditions. Note that reaction loops in the solution were not allowed. Networkx's greedy modularity algorithm[114] was applied to assign genes and reactions to a cluster in order to identify groups of genes that have a similar impact on the metabolic fluxes. We identified metabolic pathways that were enriched in each cluster using hypergeometric enrichment tests using the scipy function hypergeom. We considered a pathway as significantly enriched in a cluster using hypergeometric enrichment tests if the false discovery rate (FDR) was <1% and used the Benjamini-Hochberg method for correction against multiple testing. We considered two sets of pathway lists for the enrichment. The first used the 40 subsystems as defined in the GSM models. A second list of pathways was downloaded from the BioCyc database using the SMART tables for *E. coli* and *S. enterica*[118], which provided a more extensive list of specific metabolic pathways. Significant pathways from each model were visualised in Gephi v0.9.7[119]. To create metabolic system diagrams the KEGG (Kyoto Encyclopaedia of Genes and Genomes) was used to create the pathway for each species, *Salmonella enterica subsp. enterica serovar Typhimurium LT2*: STM0194 (stm) for *S. enterica* and *Escherichia coli K-12 MG1655*: b0153 (eco) for *E. coli*. In addition, MetaCyc and BIGG models web services were used to check each reaction's metabolites and responsible genes, and genes were also manually curated using literature.

### Statistical analysis

Statistical comparisons were made using the SciPy package implementing: 1. ANOVA tests with post hoc Tukey comparisons to consider intra- and inter-cluster variability across geography and time for the network analysis ($p$-value < 0.001); 2. A two-sided chi-squared test with Bonferroni correction to evaluate the similarities between *E. coli* and *S. enterica* AMR patterns over different antibiotics ($p$-value < 0.001); and 3. A two-sided Fishers exact test with simulated p-value was used to test for consider variations in serotypes between farms and source type. 4. Two-sided proportion tests Proportion test (F tests) with Bonferroni correction were used to assess differences in the frequency of MGE types among different cohorts with $p$-values < 0.05 considered statistically significant. A two-sided Friedman Statistical F-test ($FF$) with Iman-Davenport correction for statistical comparison of multiple datasets over the seven different classifiers used ($p$-value < 0.05). With 7 classifiers and N antibiotic models, the Friedman test is distributed according to the F distribution with 7−1 = 6 and (7−1)×(N−1) degrees of freedom. The number of antibiotic models used for the *E. coli* datasets were: 21 (Chinese all isolates), 3 (Chinese co-inhabiting chicken isolates), 15 (Chinese not co-inhabiting chicken isolates), 7 (EFFORT not co-inhabiting chicken isolates) The critical values for the *E. coli* datasets were: $F(6,120)$ for $p$ value = 0.05 is 2.17500625 for the Chinese all isolates, $F(6,12)$ for $p$ value = 0.05 is 2.99612038 for the Chinese co-inhabiting chicken isolates, $F(6,84)$ for $p$ value = 0.05 is 2.20855381 for the Chinese not co-inhabiting chicken isolates and $F(6,36)$ for $p$ value = 0.05 is 2.36375096 for the EFFORT not-necessarily co-inhabiting chicken isolates. While the number of antibiotic models used for the *S. enterica* datasets were: 13 (Chinese all isolates), 5 (Chinese co-inhabiting chicken isolates), 5 (ENGAGE not co-inhabiting chicken isolates). The critical values for the *S. enterica* datasets were: $F(6,72)$ for $p$ value = 0.05 is 2.22740397 for the Chinese all isolates, $F(6,24)$ for $p$ value = 0.05 is 2.50818882 for both the Chinese co-inhabiting chicken isolates and the ENGAGE not-necessarily co-inhabiting chicken isolates. The post-hoc Nemenyi test was used to find if there is a single classifier or a group of classifiers that performs statistically better in terms of their average rank after the $F_F$ test has rejected the null hypothesis that the performance of the comparisons on the individual classifiers over the different datasets is similar. For the GSM models, we considered a pathway as significantly enriched in a cluster using hypergeometric enrichment tests if the false discovery rate (FDR) was <1% and used the Benjamini-Hochberg method for correction against multiple testing. For the validation experiment, the statistical comparison of the optical density at 600 nm between the mutant and the wild-type strains in different concentrations was performed using a two-sided *t*-test.

### Reporting summary

Further information on research design is available in the Nature Portfolio Reporting Summary linked to this article.

## Data availability

The sequencing data supporting the conclusions of this article are available in the NCBI database under BioProject accession numbers: PRJNA675772 (Shandong 1_1 and 1_2). PRJNA841811 (*E. coli* WGS from other farms). PRJNA841813 (*S. enterica* from all farms). Public datasets were obtained from the European Nucleotide Archive with accession numbers listed in Supplementary Data 11.

## Code availability

The code is available on Github: https://github.com/tan0101/Commercial_WGS2023 under https://doi.org/10.5281/zenodo.10210870[120]

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

## Acknowledgements

The authors gratefully acknowledge the support received from the University of Nottingham Research Beacon of Excellence: Future Food, the Innovate UK monitoring officer Lynn Viatge and the Turkish Ministry of National Education. We would also like to acknowledge the time and effort of Cemile Aksoy for helping with the experimental validation analysis. This study was funded by InnovateUK grant [104986], FARM-WATCH: Fight AbR with Machine learning and a Wide Array of sensing TeCHnologies (TD) and Ministry of Science and Technology of P. R. China under Grant Key Project of International Scientific and Technological Innovation Cooperation Between Governments (number 2018YFE0101500) (Z.P.).

## Author contributions

Designed and supervised the study: J.C., F.L., Z.P., X.Z., L.L., N.S. and T.D. Planned the methodology: Z.P., X.Z., L.L., F.L., J.C. and T.D. Writing—original draft: M.B., A.M.G., and T.D. Writing—review and editing: Z.P., J.C., F.L., M.B., A.M.G., N.S. and T.D. Carried out the experiments and collected the animal and environmental samples: Z.P., W.W., Y.D., M.H., S.H., Y.H., H.L., Z.T., M.Z., Y.G., L.Z., Z.H. and X.Z. Data analysis and visualization: A.M.G., M.B., K.B. and N.P. Acquired funding: Z.P., D.R. and T.D.

## Competing interests

The authors declare no competing interests.

## Additional information

[1]School of Veterinary Medicine and Science, University of Nottingham, College Road, Sutton Bonington, Loughborough, Leicestershire LE12 5RD, UK. [2]Shandong New Hope Liuhe Group Co. Ltd. and Qingdao Key Laboratory of Animal Feed Safety, Qingdao, Shandong 266000, P.R. China. [3]NHC Key Laboratory of Food Safety Risk Assessment, China National Center for Food Safety Risk Assessment, Beijing 100021, P. R. China. [4]Nimrod Veterinary Products Limited, 2, Wychwood Court, Cotswold Business Village, Moreton-in-Marsh, GL56 0JQ London, UK. [5]Shandong Kaijia Food Co. Ltd, Weifang, P. R. China. [6]Luoyang Center for Disease Control and Prevention, No. 9, Zhenghe Road, Luolong District, Luoyang City, Henan Province, Luolong 471000, P. R. China. [7]School of Life Sciences, University of Nottingham, East Drive, Nottingham, Nottinghamshire NG7 2RD, UK. [8]Liaoning Provincial Center for Disease Control and Prevention, No. 168, Jinfeng Street, Hunnan District, Shenyang CityLiaoning Province, 110072, P. R. China. [9]Agricultural Biopharmaceutical Laboratory, College of Chemistry and Pharmaceutical Sciences, Qingdao Agricultural University, No. 700 Changcheng Road, Chengyang District, Qingdao CityShandong Province, 266109, P. R. China. [10]Chinese Veterinary Medicine Innovation Center, College of Veterinary Medicine, China Agricultural University, Haidian District, Beijing City 100193, P. R. China. [11]Department of Engineering, University of Perugia, Perugia I06125, Italy. [12]Centre for Smart Food Research, Nottingham Ningbo China Beacons of Excellence Research and Innovation Institute, University of Nottingham Ningbo China, Ningbo 315100, P. R. China. [13]These authors contributed equally: Michelle Baker, Xibin Zhang, Alexandre Maciel-Guerra. ✉e-mail: pengzixin@cfsa.net.cn; lifengqin@cfsa.net.cn; tania.dottorini@nottingham.ac.uk

