## [Peer Review File · Nature Communications]

Convergence of resistance and evolutionary responses in *Escherichia coli* and *Salmonella enterica* co-inhabiting chicken farms in ChinaREVIEWER COMMENTS

Reviewer #1 (Remarks to the Author):

The study attempts to find genomic evidence for a reasonable hypothesis from the angle of evolutionary convergence driven by co-inhabitation. The study was skillfully and comprehensively executed, but I have some concerns over the study design.

The study does not include any control group, such as *E. coli* and *S. enterica* from poultry in other countries where the usage of antibiotics and/or microbial ecology is different. Would similar observations be made between poultry-associated *E. coli* and *S. enterica* that are not co-inhabiting? For example, are those similar plasmids and the same mobile ARGs commonly circulating between *E. coli* and *S. enterica* that are not necessarily cohabitants with each other? If the same machine learning models were trained by a different training set of strains that are not co-inhabiting, would the similar AMR-associated mutations and genes be found?

To make evolutionary inferences, an important assumption is that the bacteria of interest are native residents of the environment, subject to selections (e.g. antibiotic usage) and adaptations (e.g. HGT) long enough for (micro)evolution to occur and accumulate. The observation that *E. coli* isolates were not clonal may indicate different isolates were introduced by different flocks. Which isolates were sampled from the same flock?

Why *E. coli* isolates were not serotyped and analyzed (e.g. MCRA dating) by serotypes similar to *Salmonella*?

Not sure how much the machine learning and metabolic modeling part of the study contributes to the key conclusion that convergent evolution of AMR is driven by co-inhabitation of *E. coli* and *Salmonella*. This part of the paper feels only loosely adhered to the main storyline, more speculative in nature, and almost separable as a standalone study.

Reviewer #2 (Remarks to the Author):

An excellent, well-presented and interesting study. The authors have certainly done a very nice job of analyzing their data and as far as I can judge in a very clear and appropriate way. Some of their conclusions might be due to confounding factors that they have not had the possibility to adjust for, but considering the data available, all conclusions seem solid.

I only have a few very minor comments:

- Gentamycin should be spelled gentamicin.
- Page 3, line 99 (and throughout the manuscript): the authors conclude that their observations identify co-evolution. Would just be nice to acknowledge that this could also be that these features are a pre-requisite for occupying this niche.
- Page 24, line 698: Couldn't this enrichment step select specifically for specific *E. coli* strains?

Reviewer #3 (Remarks to the Author):

Baker/Zhang/Guerra and colleagues detail the genetic epidemiology of n=661 *Salmonella enterica* and *Escherichia coli* isolated from 10 chicken farms located in China over a 2.5 year period. They carry out phylogenetic, phylodynamic, AMR gene screening, and plasmid reconstruction analyses to identify AMR transmission events. The authors characterise the population structures of the two organisms circulating in the study settings, and present data suggestive of AMR and plasmid transmission between the two organisms. The authors also use machine learning approaches to identify genes which might contribute to AMR resistance or sensitivity, and demonstrate that many of these appear to be essential to the bacteria analysed via metabolic modelling. While the genome and AMR phenotype data collected are highly novel, the study has several weaknesses that need addressing as detailed in my comments below.

Major comments:

1. At present the rationale for the machine learning work is unclear as the authors do not present data on the correlation between AMR genotypes and phenotypes detailing if the molecular determinants of AMR detected using existing tools sufficiently explain the AMR phenotypes observed. While the machine learning approach used appears sound, the results mostly report already discovered molecular determinants of AMR described in existing databases. Subsequently, the data lack novelty, and relevance of some of the reported genetic loci remains unclear as there hasn't been any experimental validation carried out. Finally, it is unclear from the current text why the authors constructed their own machine learning pipeline for these analyses when existing GWAS approaches (e.g. PubMed IDs/PMIDs: 30419019, 27572646, 27887642, 29401456, 27633831, 30535304) have been validated which account for population structure and multiple testing correction. The advantages of the machine learning approach used over GWAS are unclear from the current manuscript text.

2. In terms of the metabolic modelling aspect, the shared essential genes between knockout strains across both organisms is not particularly surprising as there is a reasonable level of homology between the two (eg. see PubMed ID: 11677608). This may also explain the similarity between the ML predictions. Moreover, it is not clear how informative the knockout models are given that many of the associations were reported in the context of substitutions rather than nonsense mutations. However, this is difficult to assess as only a list of the genes containing these was provided as supplementary data. Metabolic modelling data were also not experimentally validated, and the added value of this work was not made clear in the text.

3. While the molecular dating analyses presented in the manuscript appeared to have used appropriate model selection processes, there was no validation of the temporal signal. The current field standards for validation of temporal signal are either date-randomisation testing (e.g. PMID: 28348834), or BETS analysis (e.g. PMID: 32895707). This is particularly problematic for the data presented as they were collected over a relatively short time frame of 2.5 years, with small sample sizes for each serovar/ST analysed, which can make it exceptionally difficult to infer accurate MRCA estimates. The data reported also lack genome-wide substitution rates, which would allow comparison/validation of the dates and rates inferred against those from previous studies. The authors also omit 95% HPD values for the point estimates, so the precision of the estimates is unknown. Without these additional analyses and reported data the robustness of the estimates presented cannot be assessed.

Minor comments:

- Line 45: It would aid clarity if the authors indicated here that they mean sharing of genes.

- Supplementary table 1: I noticed that gene *aac(6'')-Iaa* is reported for *S. enterica* under AMR genes. I understand this is probably listed for completeness, but please note that this gene is cryptic, and therefore, in its wildtype state, does not confer resistance in *S. enterica* (see PMIDs: 34135355 and 10542165).

- Line 127-128: Sample counts for *S. enterica* have been erroneously switched for Shandong, and Liaoning.

- Line 139-140: Please use the full drug names in text for clarity. I can see that these are defined in the methods section, but it's clearer if the reader doesn't have to go back and forth to look these up.

- Line 139-144: Please consider rephrasing 'greater resistance' to 'resistance was more frequently observed' if this is what is meant.

- Figure 1: Panels A and B are quite complex, and the amount of data subsequently makes them difficult to read. To improve readability, authors might consider summarising the data to show the number drug classes the samples are resistant to, rather than the individual drugs, or alternatively numbering the rings and showing these in the legend. If the former option is selected, the authors

could provide a rectangular tree showing the full dataset in supplementary data. The legend should also contain any abbreviations shown in the legend (e.g. farm names/identifiers). There is also some misalignment between the phylogroup classifications and the phylogenetic tree structure for *E. coli* which is not explained.

- Line 186-7: The higher level of clonality reported may relate to the selection of the same SNP threshold used for both genera at line 185 (i.e <15 SNPs). Including substitution rate estimates from phylodynamic analyses would assist in determining this, and if the 15 SNP threshold is appropriate/relevant for understanding the genetic epidemiology for both *Salmonella* spp. and *E. coli*, or if different thresholds should be used for each. The authors could also consider commenting on putative transmission patterns and events based on these data.

- Figure 2: Using a gradient colour scale to show increasing/decreasing genetic distance may aid clarity here, as this is currently difficult to understand with the current colour palette used.

- Figure S2: Please state the statistical test used in the legend.

- Table S2: At present it is unclear how to use the 'sample pairs' column to link the data. It may be clearer to use the accession number or strain ID of the samples here.

- Line 232: At present it is unclear if the counts referenced refer to the total number of plasmids or the total number of unique plasmids. Please clarify the meaning here.

- Line 257: The meaning of "regions" is a little unclear, please clarify the meaning to refer to clades or clusters (if this is what is intended). Please also note that the text states the MRCA of the bottom box on Figure 3 should be 2019, but the current MRCA is ~2017. Either the text or the box needs to be modified.

- Figure 4: The yellow shade used in the colour palette here, coupled with the thin lines for the genes makes the figure somewhat difficult to read. Please consider replacing yellow with a different colour.

- Line 303: 26 antimicrobials are referred to here, but 28 are listed at line 129. I think this is because there was one drug not tested for each organism, and 26 indicates the overlap of drugs tested on both organisms, but it would be helpful to clarify this for the reader.

- Figures 5-6: While I understand that Roary assigns "group" IDs to hypothetical proteins, the authors should attempt to identify the putative function of these genes via BLAST searches against GenBank and UniProt/SwissProt databases and use the appropriate gene names for the figures. Looking at Table S5, this may have been carried out, but the group names are still used in the figures and it's a lot of work for the reader to link up these data.

- Table S7: This is not currently mentioned in the body text of the manuscript.

- Line 590: While the authors somewhat correctly state that direct evidence of plasmid transmission between *Salmonella* and *E. coli* are scant, there are notable cases where this is likely, for example the IncY plasmid responsible for the emergence of XDR *Salmonella* Typhi appears to originate from *E. coli* (see: 29463654).

- Line 605: 26 drugs are listed here, but the text at line 129 states 28.

- Lines 703-705 & 718-719: Please detail primer design steps or cite the sources of the primers as appropriate.

- Lines-734-741: 27 drugs are listed but line 129 says 28 were tested in total. It appears Cefotaxime/Sulbactam may have been missed here.

- Line 757: Please provide a citation for the readfq software.

- Line 780-781: Please provide a citation with access dates for the database.

- Line 802-805: From the current text, is unclear which Roary files were used as input for Piggy. The current text reads as though Piggy was applied to an alignment of core genes, which would omit intergenic regions by nature. Moreover, the following lines suggest that only an alignment of intergenic steps was used for the network alignments. Please clarify the steps used in this analysis to aid clarity and reproducibility.
- Line 816: Please provide the python package used for plotting and any relevant citations.
- Line 820: Please provide details on alignment lengths, and the phylogenetic tree rooting method used.
- The manuscript lacks substantial discussion of the limitations of the analyses carried out and how these impact the findings. This needs addressing.

Reviewer #4 (Remarks to the Author):

Here Baker et al have characterised the population structure, evolution and AMR phenotypes of *E. coli* and *S. enterica*, highlighting differences across environments and hosts. The authors showed that there is significant potential for transfer of plasmids and mobile ARGs between *E. coli* and *S. enterica* in the environments studied, suggesting potential for the spread of AMR generally. Machine learning approaches revealed both *E. coli* and *S. enterica* had a common subset of features highly associated with AMR and that they were linked to the same functional pathways essential for growth. These results indicated the potential for common metabolic adaptations of the bacteria within the sampling environments.

This work is highly valuable for understanding the evolution and transmission of AMR, especially given the continuous global rise in AMR. It provides a framework for future studies and a platform for phenotypic studies.

The authors' machine learning approaches used are robust and innovative. They have expanded on previous methods to improve capture the co-occurrence of multiple mechanisms including the additive effect on resistance. They also employed robust validation models for this data.

Comments to address

Geography is a major determinant of circulating sequence types, and these can differ quite a lot across continents. Can you address how a study based in China might be more broadly applicable globally, i.e. what are the implications outside of China?

Are the larger clusters noted for *S. enterica* due to the smaller sample size (i.e. capturing less diversity than *E. coli*) or do you think this is a true representation of the greater population structure for *S. enterica*?

It would be important to note in the discussion that the *in silico* mutations made for metabolic genes should be tested in an *in vitro* model (e.g. natural host derived and human macrophages and/or epithelial cells) in order to validate the physiological relevance of the metabolic pathway findings.

Minor comments

Line 77, ARGs not yet defined

Fig 1. The phylogroup colours in the figure legend for *E. coli* don't match the shading on the tree very well, whereas for *Salmonella* it looks very accurate. For example, phylogroup F appears as a pink shade in the legend and red on the tree, and phylogroup A appears a lighter blue than on the tree. As the figure is already very complex, this just creates a bit of confusion.

Line 220, change 'potential' to 'potentially'

Line 248, delete the word 'many'

Line 251, change 'chicken' to 'chickens' and italicise the p for the p value indicated.

Fig S9A, one of the arrows needs correction.

Line 308, is the word data repeated accidentally?

Line 327, Fig 8B should read Fig S8B

Line 328, Fig 8C-G should read Fig S8C-G

Lines 332, 338, 339, should all read Fig S9 as they are supp figs, not main figs

Line 352, should this read 'resistance phenotype'?

Check grammar in Discussion, e.g., between lines 579 – 583, words like 'a' and 'the' are missing.

Line 625, double comma needs to be corrected.

Point-by-point response to reviewers

We thank the Reviewers for their useful comments, which no doubt contributed to make this a better paper. In the following, a point-by-point response to all the questions and comments is provided. The original questions are in blue, our replies in black.

Detailed information of which figures and supplementary material were changed is provided below and in the individual responses to the reviewers comments.

Items included in this submission:

- i. Cover letter
- ii. Point-by-point Response to reviewers (ResponsetoReviewers.pdf);
- iii. Revised manuscript marked-up copy (Manuscript_markedup.pdf)
- iv. Revised manuscript clean copy (Manuscript.pdf)
- v. Revised supplementary material (SupplementaryMaterial.pdf)

Reviewer #1 – Question #1

The study does not include any control group, such as *E. coli* and *S. enterica* from poultry in other countries where the usage of antibiotics and/or microbial ecology is different. Would similar observations be made between poultry-associated *E. coli* and *S. enterica* that are not co-inhabiting? For example, are those similar plasmids and the same mobile ARGs are commonly circulating between *E. coli* and *S. enterica* that are not necessarily cohabitants with each other? If the same machine learning models were trained by a different training set of strains that are not co-inhabiting, would the similar AMR-associated mutations and genes be found?

We thank the reviewer for this insightful comment. Our updated manuscript includes new analyses done using control groups to address a possible influence of both country and of co-inhabitation, as detailed in the following.

A comment on the challenge of obtaining controls for studying the influence of co-inhabitation

Firstly, we would like to comment on the challenge of obtaining a control group with demonstrated absence of co-inhabitation. To appreciate the challenge, the opposite one, i.e., collecting a sample with demonstrated co-inhabitation, must be discussed first. It is difficult to choose a moment of collection, as we need to make sure that both bacterial species are present in the sample at the same time point. The absence of a species in a collected sample may not mean absence of co-inhabitation, but simply be due to the second species having been present earlier (disappeared at the moment of collection, for example due to usage of antibiotics, or simply not being present in sufficient quantities at the time of culturing). There are further challenges related to processing samples with co-inhabiting species, as AST and sequencing data must be generated for all the targeted isolates in parallel. To date, we are

not aware of other studies in the literature where the characterisation of co-inhabiting bacterial species in chicken has been attempted, us being the first to do so with our Chinese samples. Under this perspective, it is clear that the demonstration of absence of a species in a sample is an even greater challenge, again because it is not easy to know what happened to the sample ahead of collection, and what would happen to it if the collection were to take place at a later moment in time. A way to strengthen a proof of absence of co-inhabitation would be to collect multiple times at multiple time points, minimising the likelihood of colonisation events that may go undetected. This approach of frequent collection is however difficult to implement, due to the costs and the logistics of a more frequent collection (in our study we were able to collect at only three time points over 42 days).

To respond to the Reviewer's question, i.e., to obtain control groups with no co-inhabitation, aware of the limitations of our datasets in terms of sampling frequency, we decided to adopt a necessarily limited approach: for any given sample, we tried to isolate both bacterial species. If we found only one, we would label the corresponding isolate as "not co-inhabitant", although clearly the absence of co-inhabitation is not demonstrated. In the following, we will frequently use terms such as "not co-inhabitant" and similars, but we always imply that such status is undemonstrated.

Construction of the control sets

Using the above criterion to identify (apparent) non co-inhabitation, we created a first control set was formed by considering the Chinese *E. coli* and *S. enterica* isolates collected from bird-related samples where no co-inhabitation was found. Then, to address the second part of the question, i.e. the possible influence of country of collection on the illustrated results, we opted for generating further control sets. In order to do so, we searched for *E. coli* and *S. enterica* in poultry in other countries where the usage of antibiotics and/or microbial ecology is different and where the type of interventions (vaccines against *S. enterica*) reduces co-inhabitation of *E. coli* and *S. enterica*. Our search included: (i) extensive literature review in PubMed (using the Boolean search terms "Escherichia coli"[All Fields] AND "poultry" AND "whole genome"[All Fields] and "salmonella"[All Fields] AND "poultry" AND "whole genome"[All Fields]), and (ii) searches on public databases such as BV-BRC, European Nucleotide Archive, NCBI Pathogen Browser for other comparable studies from which whole genomes sequences and phenotypic antibiotic resistance data from chicken faeces could be retrieved. We found two suitable studies with both whole genomes sequences and phenotypic antibiotic resistance data from chicken faeces, taken from countries known to feature a lower likelihood of co-inhabitation. These studies were: the EU project "EFFORT against AMR" (Leekitcharoenphon, Johansson et al. 2021) and the "ENGAGE" study (Alba, Leekitcharoenphon et al. 2020). The isolates included in both these studies had been collected from countries with strict antimicrobial usage policies, different from those enacted in China.

We collected our first European control group from "EFFORT against AMR", which provided us with 206 *E. coli* isolates from chicken faeces with AMR phenotype data. These isolates had been obtained from samples taken from chickens in five different European countries

(Denmark, Germany, Switzerland, Poland, and Spain) all subjected to the EU Salmonella control measures Regulation (EC) No. 200/2012 (European Food Safety Authority, 2018) resulting in strong Salmonella control/vaccination programmes and less than 0.5% Salmonella positive flocks in commercial broiler chickens. We therefore assumed that the isolates retrieved from “EFFORT against AMR” would be much less likely to feature co-inhabiting *S. enterica* compared to our samples.

The second European control group was collected from the “ENGAGE” study (Alba, Leekitcharoenphon et al. 2020) and provided us with a collection of 92 *S. enterica* chicken cecum and faeces isolates from Italy. Given that *E. coli* is an ubiquitous resident/commensal in the chicken gut, we cannot exclude the presence of co-inhabiting *E. coli* in the birds, therefore we labelled this set as “not-necessarily co-inhabiting”. Nevertheless, the set was interesting because ecologically distant from both the other European set, and our original Chinese set, in particular as Italy features a much higher proportion of antimicrobial resistance in poultry than the EU average, but lower than China (Vieira, Collignon et al. 2011, Allel, Day et al. 2023, European Food Safety Authority and European Centre for Disease Prevention Control 2023).

Tests done to evaluate the influence of co-inhabitation and country of collection

In the following, we illustrate the analyses done using the three new control sets (one Chinese and two European). The focus is on identifying changes in identified mobile genetic elements – MGEs (distinct plasmid types and mobile ARGs) found across the different cohorts.

MGEs in Chinese *S. enterica* isolates: co-inhabiting vs not co-inhabiting

We identified a total of 25 distinct plasmid types in the 70 Chinese co-inhabiting *S. enterica* isolates gathered from chicken faeces and cecal samples. On the contrary, the Chinese *S. enterica* isolates with (apparent) absence of co-inhabitation (n=13), also gathered from chicken faeces and cecal samples, exhibited 14 distinct plasmid types, with all 14 of them being in common with the co-inhabiting set. Of these 14, none were found in a significantly different proportions in either cohort (Proportion test, two-tailed, Bonferroni correction, all adjusted p-values > 0.05).

When comparing the mobile ARGs between the two cohorts, a total of 22 mobile ARGs were identified in the Chinese co-inhabiting isolates from faeces and cecal samples, and 17 were identified in the Chinese non co-inhabiting isolates from faeces and cecal, with all 17 of those of them being in common to both. Of these 17 common mobile ARGs, one was significantly more frequent in the non co-inhabiting cohort (IS6100-aadA7) whilst another was significantly more frequent in the co-inhabiting cohort, ISKpn19-QnrS1 (Proportion test, two-tailed, Bonferroni correction, all adjusted p-values < 0.05).

MGEs in Chinese E. coli isolates: co-inhabiting vs not co-inhabiting

We identified a total of 45 distinct plasmid types in the 70 Chinese co-inhabiting *E. coli* isolates gathered from chicken faeces and cecal samples. The Chinese *E. coli* isolates with no confirmed co-inhabitation (n=245), gathered chicken faeces and cecal samples, exhibited 52 distinct plasmid types, with 41 of them being in common with the co-inhabiting set. Of these 41, none were found in a significantly different proportions in either cohort (Proportion test, two-tailed, Bonferroni correction, all adjusted p-values > 0.05).

When comparing the mobile ARGs between the two cohorts, a total of 108 mobile ARGs were identified in the Chinese co-inhabiting cohort from faeces and cecal isolates, 155 were identified in the Chinese non co-inhabiting isolates from faeces and cecal, 82 of them being in common to both. Of these 82, none were found in a significantly different proportion in either cohort (Proportion test, two-tailed, Bonferroni correction, all adjusted p-values > 0.05).

MGEs: Chinese S. enterica co-inhabiting isolates vs ENGAGE S. enterica not necessarily co-inhabiting isolates

When considering the European *S. enterica* isolates (n=92) from Italy (ENGAGE study), 13 distinct plasmid types were found. The Chinese *S. enterica* chicken gut isolates (n=70) from our study (i.e., isolates from samples with confirmed co-inhabitation), exhibited 25 distinct plasmid types, of which 8 in common with the European cohort. Considering the overall number of distinct plasmid types found in the two cohorts (30), the Chinese cohort carried a significantly greater number of plasmid types compared to the European isolates (Proportion test across all samples, two-tailed, p value < 0.01). Out of the eight plasmids that overlapped, 2 were significantly more prevalent (Proportion test, two-tailed, Bonferroni correction) in the Chinese isolates compared to the European isolates; namely: Col156 (adj. p-value < 0.0001) and ColVC (adj. p-value < 0.0001). Conversely, four plasmid types were found in significantly higher proportions among the European isolates (IncX1, IncX3, IncX4 and IncFIB(K)), (Proportion test, two-tailed, Bonferroni correction, all adjusted p-values > 0.001). In total, 24 mobile ARG types were found in the ENGAGE cohort, whilst 22 mobile ARGs were found in the Chinese one, none in common between both cohorts. In conclusion, from this comparison the number of plasmid and mobile ARG types is significantly different between the Chinese cohort (*S. enterica* co-inhabiting with *E. coli*) and the ENGAGE cohort (*S. enterica* isolates not necessarily co-inhabiting with *E. coli*).

MGEs: Chinese E. coli co-inhabiting isolates vs EFFORT E. coli not co-inhabiting isolates

We identified a total of 50 distinct plasmid types in the 206 European *E. coli* isolates gathered from the European set. The Chinese *E. coli* gut isolates (n=70) from our study (again taken from samples with confirmed co-inhabitation) exhibited 45 distinct plasmid types, 42 of them being in common with the EFFORT set. Of these 42, six (IncHI2A, IncHI2, pKPC-CAV1321, IncI2(Delta), IncN, IncFII(pHN7A8)) were found in a significantly greater proportion in the Chinese cohort compared to the European cohort (Proportion test, two-tailed, Bonferroni

correction, all adjusted p-values < 0.05, Three of the 42 shared plasmid types (Incl1-I(Alpha), ColRNAI, Col(MG828)) were present in a significantly greater proportion in the European cohort compared to the Chinese one.

When comparing the mobile ARGs between the two cohorts, a total of 77 mobile ARGs were identified in the EFFORT cohort and 108 in the Chinese cohort, 22 of them being in common to both. Of the 22 common mobile ARGs, six were significantly more frequent in the Chinese cohort, namely: IS6100 -mphA, ISEc59 -APH(4)-Ia, ISEc59-AAC(3)-IV, ISKpn19-QnrS1, ISVsa3 -floR (Proportion test, two-tailed, Bonferroni correction, all adjusted p-values < 0.0001). However, ISVsa3-sul2 was found significantly more frequently in the European cohort (Proportion test, two-tailed, Bonferroni correction, adjusted p-value < 0.001).

Shared MGEs: set of MGEs shared by co-inhabiting Chinese E. coli and Chinese S. enterica vs. set of MGEs shared by EFFORT E. coli and ENGAGE S. enterica

We investigated the possibility of detecting similarities between proportions of distinct plasmid types and mobile ARGs shared by *E. coli* and *S. enterica* isolates, when comparing Chinese and European cohorts. The European cohort consisted of the aggregation of the EFFORT and ENGAGE sets. Larger numbers of shared plasmid types and mobile ARGs were found in the Chinese cohort compared to the European one: 30% vs 26% distinct plasmids types; 16% vs 4% mobile ARGs, though no statistically significant differences were present.

Shared MGEs: set of MGEs shared by co-inhabiting Chinese E. coli and Chinese S. enterica vs. set of MGEs shared by co-inhabiting Chinese E. coli and not-necessarily co-inhabiting ENGAGE S. enterica

Both plasmid types and mobile ARGs were found significantly more shared when analysing the Chinese isolates (*S. enterica* + *E. coli* isolates), compared to the combined set formed by ENGAGE *S. enterica* isolates and Chinese *E. coli* isolates (two-tailed chi-squared test with Yates and Bonferroni correction, adjusted p-value = 0.05).

Comments on the new results

Overall, our results show two things:

- a) There is a positive correlation between co-inhabitation of *E. coli* and *S. enterica* and the proportions of mobile genetic elements (plasmids and mobile ARGs) observed in the isolates (shared and not shared between the two bacterial species), as indicated in particular by the comparison of the Chinese cohorts with confirmed and not confirmed co-inhabitation, and by the comparison of Chinese *E. coli* isolates (with confirmed co-inhabitation) with the EFFORT *E. coli* isolates (with likely absence of co-inhabitation, due to the very low Salmonella prevalence in the five EU countries considered, due to the EU Salmonella control measures Regulation (EC) No. 200/2012

(European Food Safety Authority, 2018) resulting in strong Salmonella control/vaccination programmes and less than 0.5% Salmonella positive flocks in commercial broiler chickens).

- b) There is a higher occurrence of mobile genetic elements within the individual species (*S. enterica* and *E. coli*) in our Chinese cohort compared to the European cohorts, Analogously, we observe a greater prevalence of shared MGEs between the two species in the Chinese cohort, compared to the European ones. These observations indicate a likely influence of the country of collection.

Updates to the Submission

The Result section in the updated manuscript, Lines 324-352 has been updated as follows:

Results Lines 324-352: “To investigate the influence of co-inhabitation of *E. coli* and *S. enterica* as well as the influence of country of collection (antimicrobial usage and microbial ecology) on the observed results (proportion of distinct plasmid types and mobile ARGs, as well as amount of sharing between the two bacteria species), we considered three control sets. The first was formed of Chinese bird samples with apparent lack of co-inhabitation. The second and third were European, retrieved as publicly available data from two previous research projects. Namely: “EFFORT against AMR” (Leekitcharoenphon, Johansson et al. 2021) (206 *E. coli* isolates from chicken faeces with AMR phenotypes, collected in five different European countries - Denmark, Germany, Switzerland, Poland, and Spain - where the birds are subjected to strict control measures against Salmonella, thus co-presence of *S. enterica* is unlikely) and “ENGAGE” (Alba, Leekitcharoenphon et al. 2020) (92 *S. enterica* chicken cecum and faeces isolates from Italy, where samples may or may not contain co-inhabiting *E. coli*). We then performed multiple statistical comparison tests to search for differences in proportions of plasmids and mobile ARGs (including shared ones) when considering our Chinese isolates (faeces and caecal swabs) and the European ones (see Supplementary Note 3 for further details). In general, our results show that:

- A) both the number of distinct plasmid types and the number of mobile ARGs is higher in the Chinese cohort, indicating a likely influence of the country of collection in the observed results;
- B) there is a positive correlation between co-inhabitation of *E. coli* and *S. enterica*, and the proportions of mobile genetic elements (plasmids and mobile ARGs) observed in their isolates, as indicated in particular by the comparison of the Chinese co-inhabiting vs not co-inhabiting cohorts, and by the comparison of Chinese *E. coli* isolates (with confirmed co-inhabitation) with the EFFORT *E. coli* isolates (with likely absence of co-inhabitation, due to the very low *Salmonella spp.* prevalence, owing to the EU Salmonella control measures Regulation (EC) No. 200/2012 (European Food Safety Authority, 2018) - resulting in strong Salmonella control/vaccination programmes and less than 0.5% Salmonella positive flocks in commercial broiler chickens). “

A new sentence was added to the discussion, lines 731-734: “Our results indicate that sharing is influenced by many factors, and we have reported evidence on the effects of co-inhabitation, as well as country of collection, possibly reflecting differences in regulations, interventions and overall microbial ecology.”

Comments on the challenges of generating not co-inhabitant sets were added in the Supplementary Note 3:

Supplementary Material Lines 45-63: “Firstly, we would like to comment on the challenge of obtaining a control group with demonstrated absence of co-inhabitation. To appreciate the challenge, the opposite one, i.e., collecting a sample with demonstrated co-inhabitation, must be discussed first. It is difficult to choose a moment of collection, as we need to make sure that both bacterial species are present in the sample at the same time point. The absence of a species in a collected sample may not mean absence of co-inhabitation, but simply be due to the second species having been present earlier (disappeared at the moment of collection, for example due to usage of antibiotics, or simply not being present in sufficient quantities at the time of culturing). There are further challenges related to processing samples with co-inhabiting species, as AST and sequencing data must be generated for all the targeted isolates in parallel. To date, we are not aware of other studies in the literature where the characterisation of co-inhabiting bacterial species in chicken has been attempted, us being the first to do so with our Chinese samples. Under this perspective, it is clear that the demonstration of absence of a species in a sample is an even greater challenge, again because it is not easy to know what happened to the sample ahead of collection, and what would happen to it if the collection were to take place at a later moment in time. A way to strengthen a proof of absence of co-inhabitation would be to collect multiple times at multiple time points, minimising the likelihood of colonisation events that may go undetected. This approach of frequent collection is however difficult to implement, due the costs and the logistics of a more frequent collection (in our study we were able to collect at only three time points over 42 days). “

The Supplementary Material has been updated as follows:

Supplementary Material Lines 65-201: **“Supplementary Note 3:**

Creation of control sets and related analyses for co-inhabitation and country of collection

To test for the hypothetical influence on our results (proportion of plasmids and mobile ARGs, shared or not) of factors such as country of collection, as well as of co-inhabitation within the same sample of *S. enterica* and *E. coli*, we created three control sets. The first was formed by considering the Chinese *E. coli* and *S. enterica* isolates collected from bird-related samples where no co-inhabitation was found. Note that the absence of co-inhabitation is not demonstrated, but rather the result of not having found both bacteria in the analysed sample. Then, to obtain further control sets, we searched for *E. coli* and *S. enterica* from poultry in other countries where the usage of antibiotics and/or microbial ecology is different and

where the type of interventions (vaccines against *S. enterica*) reduces co-inhabitation of *E. coli* and *S. enterica*. Our search included: (i) extensive literature review in PubMed (using the Boolean search terms "Escherichia coli"[All Fields] AND "poultry" AND "whole genome"[All Fields] and "salmonella"[All Fields] AND "poultry" AND "whole genome"[All Fields]), and (ii) searches on public databases such as BV-BRC, European Nucleotide Archive, NCBI Pathogen Browser for other comparable studies from which whole genomes sequences and phenotypic antibiotic resistance data from chicken faeces could be retrieved. We found two suitable studies with both whole genomes sequences and phenotypic antibiotic resistance data from chicken faeces, taken from countries known to feature a lower likelihood of co-inhabitation. These studies were: the EU project "EFFORT against AMR" (Leekitcharoenphon, Johansson et al. 2021) and the "ENGAGE" study (Alba, Leekitcharoenphon et al. 2020). The isolates included in both these studies had been collected from countries with strict antimicrobial usage policies, different from those enacted in China.

We collected our first European control group from "EFFORT against AMR", which provided us with 206 *E. coli* isolates from chicken faeces with AMR phenotype data. These isolates had been obtained from samples taken from chickens in five different European countries (Denmark, Germany, Switzerland, Poland, and Spain) all subjected to the EU Salmonella control measures Regulation (EC) No. 200/2012 (European Food Safety Authority, 2018) resulting in strong Salmonella control/vaccination programmes and less than 0.5% Salmonella positive flocks in commercial broiler chickens. We therefore assumed that the isolates retrieved from "EFFORT against AMR" would be much less likely to feature co-habiting *S. enterica* compared to our samples.

The second European control group was collected from the "ENGAGE" study (Alba, Leekitcharoenphon et al. 2020) and provided us with a collection of 92 *S. enterica* chicken cecum and faeces isolates from Italy. Given that *E. coli* is an ubiquitous resident/commensal in the chicken gut, we cannot exclude the presence of co-inhabiting *E. coli* in the birds, therefore we labelled this set as "not-necessarily co-inhabiting". Nevertheless, the set was interesting because ecologically distant from both the other European set, and our original Chinese set, in particular as Italy features a much higher proportion of antimicrobial resistance in poultry than the EU average, but lower than China (Vieira, Collignon et al. 2011, Allel, Day et al. 2023, European Food Safety Authority and European Centre for Disease Prevention Control 2023)

In the following, we illustrate the analyses done using the new sets. The focus is on identifying changes in identified mobile genetic elements – MGEs (distinct plasmid types and mobile ARGs) found across the different cohorts.

MGEs in Chinese S. enterica isolates: co-inhabiting vs not co-inhabiting

We identified a total of 25 distinct plasmid types in the 70 Chinese co-inhabiting *S. enterica* isolates gathered from chicken faeces and cecal samples. On the contrary, the Chinese *S. enterica* isolates with no confirmed co-inhabitation (n=13), also gathered from chicken faeces and cecal samples, exhibited 14 distinct plasmid types, with all 14 of them being in common with the co-inhabiting set. Of these 14, none were found in a significantly different

proportions in either cohort (Proportion test, two-tailed, Bonferroni correction, all adjusted p-values > 0.05).

When comparing the mobile ARGs between the two cohorts, a total of 22 mobile ARGs were identified in the Chinese co-inhabiting isolates from faeces and cecal samples, and 17 were identified in the Chinese non co-inhabiting isolates from faeces and cecal, with all 17 of those of them being in common to both. Of these 17 common mobile ARGs, one was significantly more frequent in the non co-inhabiting cohort (*IS6100-aadA7*) whilst another was significantly more frequent in the co-inhabiting cohort, *ISKpn19-qnrS1* (Proportion test, two-tailed, Bonferroni correction, all adjusted p-values < 0.05).

MGEs in Chinese E. coli isolates: co-inhabiting vs not co-inhabiting

We identified a total of 45 distinct plasmid types in the 70 Chinese co-inhabiting *E. coli* isolates gathered from chicken faeces and cecal samples. The Chinese *E. coli* isolates with no confirmed co-inhabitation (n=245), gathered chicken faeces and cecal samples, exhibited 52 distinct plasmid types, with 41 of them being in common with the co-inhabiting set. Of these 41, none were found in a significantly different proportions in either cohort (Proportion test, two-tailed, Bonferroni correction, all adjusted p-values > 0.05).

When comparing the mobile ARGs between the two cohorts, a total of 108 mobile ARGs were identified in the Chinese co-inhabiting cohort from faeces and cecal isolates, 155 were identified in the Chinese non co-inhabiting isolates from faeces and cecal, 82 of them being in common to both. Of these 82, none were found in a significantly different proportion in either cohort (Proportion test, two-tailed, Bonferroni correction, all adjusted p-values > 0.05).

MGEs: Chinese S. enterica co-inhabiting isolates vs ENGAGE S. enterica not-necessarily co-inhabiting isolates

When considering the European *S. enterica* isolates (n=92) from Italy (ENGAGE study), 13 distinct plasmid types were found. The Chinese *S. enterica* chicken gut isolates (n=70) from our study (i.e., isolates from samples with confirmed co-inhabitation), exhibited 25 distinct plasmid types, of which 8 in common with the European cohort. Considering the overall number of distinct plasmid types found in the two cohorts (30), the Chinese cohort carried a significantly greater number of plasmid types compared to the European isolates (Proportion test across all samples, two-tailed, p value < 0.01). Out of the eight plasmids that overlapped, 2 were significantly more prevalent (Proportion test, two-tailed, Bonferroni correction) in the Chinese isolates compared to the European isolates; namely: Col156 (adj. p-value < 0.0001) and ColVC (adj. p-value < 0.0001). Conversely, four plasmid types were found in significantly higher proportions among the European isolates (*IncX1*, *IncX3*, *IncX4* and *IncFIB(K)*), (Proportion test, two-tailed, Bonferroni correction, all adjusted p-values > 0.001). In total, 24 mobile ARG types were found in the ENGAGE cohort, whilst 22 mobile ARGs were found in the Chinese one, none in common between both cohorts. In conclusion, from this comparison the number of plasmid and mobile ARG types is significantly different between the Chinese cohort (*S. enterica* co-inhabiting with *E. coli*) and the ENGAGE cohort (*S. enterica* isolates not necessarily co-inhabiting with *E. coli*).

MGEs: Chinese E. coli co-inhabiting isolates vs EFFORT E. coli not co-inhabiting isolates

We identified a total of 50 distinct plasmid types in the 206 European *E. coli* isolates gathered from the European set. The Chinese *E. coli* gut isolates (n=70) from our study (again taken from samples with confirmed co-inhabitation) exhibited 45 distinct plasmid types, 42 of them being in common with the EFFORT set. Of these 42, six (IncHI2A, IncHI2, pKPC-CAV1321, IncI2(Delta), IncN, IncFII(pHN7A8)) were found in a significantly greater proportion in the Chinese cohort compared to the European cohort (Proportion test, two-tailed, Bonferroni correction, all adjusted p-values < 0.05). Three of the 42 shared plasmid types (IncI1-I(Alpha), ColRNAI, Col(MG828)) were present in a significantly greater proportion in the European cohort compared to the Chinese one.

When comparing the mobile ARGs between the two cohorts, a total of 77 mobile ARGs were identified in the EFFORT cohort and 108 in the Chinese cohort, 22 of them being in common to both. Of the 22 common mobile ARGs, six were significantly more frequent in the Chinese cohort, namely: IS6100 -mphA, ISEc59-APH(4)-I α , ISEc59-AAC(3)-IV, ISKpn19-qnrS1, ISVsa3-floR (Proportion test, two-tailed, Bonferroni correction, all adjusted p-values < 0.0001). However, ISVsa3-sul2 was found significantly more frequently in the European cohort (Proportion test, two-tailed, Bonferroni correction, adjusted p-value < 0.001).

Shared MGEs: set of MGEs shared by co-inhabiting Chinese E. coli and Chinese S. enterica vs. set of MGEs shared by EFFORT E. coli and ENGAGE S. enterica

We investigated the possibility of detecting similarities between proportions of distinct plasmid types and mobile ARGs shared by *E. coli* and *S. enterica* isolates, when comparing Chinese and European cohorts. The European cohort consisted of the aggregation of the EFFORT and ENGAGE sets. Larger numbers of shared plasmid types and mobile ARGs were found in the Chinese cohort compared to the European one: 30% vs 26% distinct plasmids types; 16% vs 4% mobile ARGs, though no statistically significant differences were present.

Shared MGEs: set of MGEs shared by co-inhabiting Chinese E. coli and Chinese S. enterica vs. set of MGEs shared by co-inhabiting Chinese E. coli and not-necessarily co-inhabiting ENGAGE S. enterica

Both plasmid types and mobile ARGs were found significantly more shared when analysing the Chinese isolates (*S. enterica* + *E. coli* isolates), compared to the combined set formed by ENGAGE *S. enterica* isolates and Chinese *E. coli* isolates (two-tailed chi-squared test with Yates and Bonferroni correction, adjusted p-value = 0.05).

Overall, our results show two things:

- a) There is a positive correlation between co-inhabitation of *E. coli* and *S. enterica* and the proportions of mobile genetic elements (plasmids and mobile ARGs) observed in the isolates (shared and not shared between the two bacterial species), as indicated in particular by the comparison of the Chinese cohorts with confirmed and not

confirmed co-inhabitation, and by the comparison of Chinese *E. coli* isolates (with confirmed co-inhabitation) with the EFFORT *E. coli* isolates (with likely absence of co-inhabitation, due to the very low Salmonella prevalence in the five EU countries considered, due to the EU Salmonella control measures Regulation (EC) No. 200/2012 (European Food Safety Authority, 2018) resulting in strong Salmonella control/vaccination programmes and less than 0.5% Salmonella positive flocks in commercial broiler chickens).

- b) There is a higher occurrence of mobile genetic elements within the individual species (*S. enterica* and *E. coli*) in our Chinese cohort compared to the European cohorts, Analogously, we observe a greater prevalence of shared MGEs between the two species in the Chinese cohort, compared to the European ones. These observations indicate a likely influence of the country of collection.

References for the response:

- Allel, K., L. Day, A. Hamilton, L. Lin, L. Furuya-Kanamori, C. E. Moore, T. Van Boeckel, R. Laxminarayan and L. Yakob (2023). "Global antimicrobial-resistance drivers: an ecological country-level study at the human–animal interface." *The Lancet Planetary Health* 7(4): e291-e303.
- European Food Safety Authority and European Centre for Disease Prevention Control (2023). "The European Union Summary Report on Antimicrobial Resistance in zoonotic and indicator bacteria from humans, animals and food in 2020/2021." *EFSA Journal* 21(3): e07867.
- Vieira, A. R., P. Collignon, F. M. Aarestrup, S. A. McEwen, R. S. Hendriksen, T. Hald and H. C. Wegener (2011). "Association between antimicrobial resistance in *Escherichia coli* isolates from food animals and blood stream isolates from humans in Europe: an ecological study." *Foodborne Pathog Dis* 8(12): 1295-1301.
- Leekitcharoenphon P, et al. Genomic evolution of antimicrobial resistance in *Escherichia coli*. *Scientific Reports* 11, 15108 (2021).
- European Food Safety Authority, European Centre for Disease Prevention Control. The European Union summary report on trends and sources of zoonoses, zoonotic agents and food-borne outbreaks in 2017. *EFSA Journal* 16, e05500 (2018).
- Alba P, et al. Molecular epidemiology of *Salmonella* *Infantis* in Europe: insights into the success of the bacterial host and its parasitic pESI-like megaplasmid. *Microb Genom* 6, (2020).

Reviewer #1 – Question #2

If the same machine learning models were trained by a different training set of strains that are not co-inhabiting, would the similar AMR-associated mutations and genes be found?

This issue ties into the previous as the goal is to understand the influence of co-inhabitation into obtaining the observed results. In principle, as in the previous response (Reviewer #1 – Question #1) we have now documented further experiments indicating that co-inhabitation seems indeed to have an influence, we would expect machine learning to react accordingly when trained with a not co-inhabiting data set. To verify this, we performed additional tests using machine learning, now included in the updated manuscript and illustrated in the following. These experiments address the machine learning dependency on co-inhabitation, but– to follow up on the previous questions – also on country on collection.

Methods and datasets

We first selected the set of *E. coli* and *S. enterica* isolates that had been confirmed to be co-inhabitants in our Chinese cohort. This set consisted of instances in which positive cultures verified the presence of both *E. coli* and *S. enterica* species in chicken, with a total of 98 such cases. The 98 isolates included: 47 chicken faeces, 23 chicken caecal droppings, 16 chicken carcass, 12 chicken feather. Since the number of confirmed co-inhabiting isolates in chicken faeces alone was limited to 47, insufficient for training the machine learning model, and by including chicken caecal droppings we increase to 70 (still insufficient), we opted to perform the training using all confirmed co-inhabiting isolates found in chicken sources (including faeces, ceca, carcasses, and feathers). Then, we selected the not co-inhabiting isolates from the same Chinese cohort of chicken isolates (see previous answer for a discussion on the lack of co-inhabitation being observed but not demonstrated). Next, we selected the data from the European EFFORT and ENGAGE sets (also illustrated in the previous answer) to perform an analogous training of the machine learning models, in the control experiments.

The same machine learning pipeline (see Fig. S8 and S9, Results line 357-429, Materials and Methods 1088-1143) was used in both the test experiments (Chinese *E. coli* and Chinese *S. enterica* data) and control experiments (not co-inhabiting Chinese and European data).

Machine learning results for the Chinese co-inhabiting sets

When training the machine learning models on the Chinese data, the only occasional modification to the pipeline was the p-value of the chi-square test, set to 0.05 for *E. coli* due to the small number of isolates. The following features were encoded as inputs to the machine learning models: presence/absence of accessory genes, core genome SNPs and intergenic region SNPs (therefore covering both mutations and genes). When training the machine learning models for the test experiments (Chinese data), for some of the 28 antimicrobials, because of the limited number of isolates, the distribution of resistance/susceptibility profiles was skewed, with most isolates being either susceptible or resistant. Consequently, these data could not be used for constructing predictive models. Given the above limitations we were able to run the machine learning pipeline on 3 antibiotics (chloramphenicol - CHL, ciprofloxacin - CIP, nalidixic acid – NAL) for the co-inhabiting *E. coli* dataset and on 5 antibiotics (cefotaxime – CTX, gentamycin – GEN, minocycline – MIN, tetracycline – TET and

trimethoprim / sulfamethoxazole – SXT) for the co-inhabiting *S. enterica* dataset. Figures 1 and 2 below show the performance metrics and the number of features for the co-inhabiting *E. coli* and *S. enterica* Chinese datasets.

Figure 1. The supervised machine learning pipeline effectively predicts the resistance/susceptibility profiles of co-inhabiting *E. coli* isolates. This prediction is based on a dataset comprising 98 isolates collected from a Chinese cohort and originating from various chicken sources, including faeces, ceca, carcasses, and feathers. Machine learning performance results are given for six performance indicators: (I) area under the curve AUC, (II) accuracy, (III) sensitivity, (IV) Specificity, (V) Cohen's kappa score, and (VI) precision. All indicators were calculated from 30 training runs for each antimicrobial model. (VII) Number and type of features (accessory genes, core genome SNPs and intergenic region SNPs) selected by each antibiotic model.

Figure 2. The supervised machine learning pipeline effectively predicts the resistance/susceptibility profiles of co-inhabiting *S. enterica* isolates. This prediction is based on a dataset comprising 98 isolates collected from a Chinese cohort and originating from various chicken sources, including faeces, ceca, carcasses, and feathers. Machine learning performance results are given for six performance indicators: (I) area under the curve AUC, (II) accuracy, (III) sensitivity, (IV) Specificity, (V) Cohen’s kappa score, and (VI) precision. All indicators were calculated from 30 training runs for each antimicrobial model. (VII) Number and type of features (accessory genes, core genome SNPs and intergenic region SNPs) selected by each antibiotic model.

The application of the remaining part of the pipeline, led to the results illustrated in Table 1 for the Chinese datasets.

Table 1. Number of AMR-associated accessory genes, core genome SNPs, and core genome genes found by the machine learning pipeline, in the Chinese co-inhabiting *E. coli* and *S. enterica* isolates, with hypothetical genes excluded.

Dataset	Antibiotics	Accessory genes	Core genome SNPs	Core genome Genes
Chinese co-inhabiting E. coli	CHL	46	618	431
	CIP	50	443	324
	NAL	38	502	360
Chinese co-inhabiting S. enterica	CTX	41	257	227
	GEN	26	220	205
	MIN	18	89	87
	SXT	42	210	185
	TET	42	290	255

Machine learning results for the Chinese not co-inhabiting sets

When training the machine learning models on the Chinese not co-inhabiting sets, no modification was done to the original proposed pipeline. There were 337 not co-inhabiting *E. coli* isolates from chicken sources, a sufficient number to train the machine learning pipeline to create predictor models for *E. coli*. However, there were only 17 not co-inhabiting *S. enterica* isolates from chicken sources, which was not sufficient to train the machine learning models. Therefore, the analyses focused solely on training the machine learning models for *E. coli*. The same features used in the test experiments (with co-inhabitant isolates) were encoded as inputs, i.e., presence/absence of accessory genes, core genome SNPs and intergenic region SNPs. The predictive models for some of the antibiotics could not be trained, because of excessive disproportion of resistance or susceptibility phenotypes for those antibiotics. We were able to complete the training for 15 antibiotics (amikacin – AMI, aztreonam – AZM, ceftazidime - CAZ, ceftazidime/clavulanic acid – CAZ-C, ceftiofur – CFT, chloramphenicol - CHL, cefotaxime – CTX, cefepime – FEP, gentamycin – GEN, kanamycin – KAN, minocycline – MIN, nalidixic acid – NAL, streptomycin – STR, sulphafurazole – SUL and trimethoprim/sulfamethoxazole – SXT). The performance metrics and the number of features selected as relevant by the machine learning predictors of AMR to specific antibiotics in *E. coli*, are summarised in Figure 3. These models ran using the not co-inhabiting *E. coli* isolates.

Figure 3. The machine learning pipeline effectively predicts the resistance/susceptibility profiles of Chinese not co-inhabiting *E. coli* isolates. This prediction is based on a dataset comprising 337 isolates collected from a Chinese cohort and originating from various chicken sources, including faeces, ceca, carcasses, and feathers. Machine learning performance results are given for six performance indicators: (I) area under the curve AUC, (II) accuracy, (III) sensitivity, (IV) Specificity, (V) Cohen's kappa score, and (VI) precision. All indicators were calculated from 30 training runs for each antimicrobial model. (VII) Number and type of features (accessory genes, core genome SNPs and intergenic region SNPs) selected by each antibiotic model.

The application of the pipeline, led to the results illustrated in Table 2 for the Chinese not co-inhabiting datasets.

Table 2. Number of AMR-associated accessory genes, core genome SNPs, and core genome genes found by the machine learning pipeline, in the Chinese not co-inhabiting datasets for *E. coli*, with hypothetical genes excluded.

Dataset	Antibiotics	Accessory genes	Core genome SNPs	Core genome Genes
Chinese not co-inhabiting E. coli	AMI	94	832	478
	AZM	70	255	163
	CAZ	4	32	30
	CAZ-C	24	183	118
	CFX	70	902	500
	CHL	98	1007	574
	CTX	83	510	331
	FEP	31	81	55
	GEN	43	173	128
	KAN	64	295	209
	MIN	87	802	488
	NAL	81	730	461
	STR	110	946	547
	SUL	101	809	467
SXT	56	411	249	

Comparison between Chinese not co-inhabiting and Chinese co-inhabiting machine learning results

The comparison between not co-inhabiting and co-inhabiting results is summarised in Table 3, where the degree of overlap of identified genetic elements is illustrated as percentages. The overlaps (i.e., percentages of genes identified by machine learning in both datasets) were estimated by mapping the findings back to the pangenome. Note that in the table, the co-inhabiting sample (test) is taken as the reference (as opposed to selecting the control, i.e. the not co-inhabiting sample). This is to allow comparison of overlaps with the following tests. Despite the overlaps in Table 3 being in some cases relatively large, the differences in prediction generated by the machine learning models provide an additional hint at co-inhabitation having a likely effect on the machine learning prediction.

Table 3. Percentages of AMR-associated accessory genes, core genome genes, and overall number of genes identified by machine learning in both the Chinese co-inhabiting and not co-inhabiting *E. coli* isolates. The co-inhabiting set is taken as reference. Hypothetical genes were excluded, each column gives the percentages of genes in common between the sets.

Datasets	Antibiotics	Accessory genes	Core genome genes	Overall number of genes
Not co-inhabiting Chinese E. coli	CHL	8.70%	43.39%	40.04%
	NAL	5.26%	42.22%	38.69%

An average of 39.37% of genes were found in common across antibiotics. This includes both accessory and core genes. In more detail, 723 genes were found in common, representing accessory genes and core genes related to SNPs across all antibiotic modes. Interestingly, only five of these genes are known antibiotic resistance genes: *kpdE* (aminoglycoside), *emrB* (fluoroquinolone), *mdtA* (multi drug resistance), *cpxA* (multi drug resistance), *acrB* (multi drug resistance).

Machine learning results for the European not-necessarily co-inhabiting sets

As anticipated earlier, a second set of experiments was conducted using machine learning, to incorporate in the analysis the investigation of a possible influence of country of collection. The aforementioned EFFORT and ENGAGE datasets were used to train machine learning models in order to obtain further control results. It should be reiterated that the EFFORT *E. coli* set is labelled as “not co-inhabiting” due to the less likely presence of *S. enterica* consequent to bird treatment; whilst the ENGAGE *S. enterica* set is labelled as not-necessarily co-inhabiting because the presence of *E. coli* is unknown.

For *E. coli* (EFFORT dataset) we had 206 isolates with AMR phenotype data, of which 7 antibiotics (ampicillin - AMP, chloramphenicol - CHL, ciprofloxacin - CIP, nalidixic acid – NAL, sulfisoxazole – SUL, tetracycline – TET and trimethoprim – TMP) had enough isolates in each class for the machine learning models to be trained. For *S. enterica* (ENGAGE dataset) 92 isolates were available with AMR phenotype data, of which 5 antibiotics (ampicillin – AMP, chloramphenicol – CHL, cefotaxime – CTX, tetracycline – TET, trimethoprim – TMP) had enough isolates in each class for the machine learning models to be trained. Figures 4 and 5, and Table 4 below, report the performance metrics and the features identified by the machine learning pipeline for the control experiments.

Figure 4. The supervised machine learning pipeline effectively predicts the resistance/susceptibility profiles of EFFORT *E. coli* isolates. This prediction is based on a dataset comprising 206 isolates collected from a European cohort and originating from chicken faeces, with strong likelihood of absence of co-inhabiting *S. enterica*. Machine learning performance results are given for six performance indicators: (I) area under the curve AUC, (II) accuracy, (III) sensitivity, (IV) Specificity, (V) Cohen's kappa score, and (VI) precision. All indicators were calculated from 30 training runs for each antimicrobial model. (VII) Number and type of features (accessory genes, core genome SNPs and intergenic region SNPs) selected by each antibiotic model.

Figure 5. The supervised machine learning pipeline effectively predicts the resistance/susceptibility profiles of ENGAGE *S. enterica* isolates (from samples which may or may not contain also co-inhabiting *E. coli*). The prediction is based on a dataset comprising 92 isolates collected from an Italian cohort and originating from chicken faeces and cecum. Machine learning performance results are given for six performance indicators: (I) area under the curve AUC, (II) accuracy, (III) sensitivity, (IV) Specificity, (V) Cohen's kappa score, and (VI) precision. All indicators were calculated from 30 training runs for each antimicrobial model. (VII) Number and type of features (accessory genes, core genome SNPs and intergenic region SNPs) selected by each antibiotic model.

Table 4. Number of AMR-associated accessory genes, core genome SNPs, and core genome genes found by the machine learning, in the European EFFORT and ENGAGE datasets, with hypothetical genes excluded.

Dataset	Antibiotics	Accessory genes	Core genome SNPs	Core genome Genes
E. coli (EFFORT)	AMP	100	744	378
	CHL	105	685	368
	CIP	98	672	284
	NAL	83	533	277
	SUL	39	272	154
	TET	87	719	335
	TMP	59	321	177
S. enterica (ENGAGE)	AMP	21	266	106
	CHL	18	217	81
	CTX	18	118	71
	TET	16	244	74
	TMP	34	265	102

Comparison between European non-necessarily co-inhabiting and the Chinese co-inhabiting machine learning results

In Table 5 (below), the degrees of overlap between the results on the European sets and the original results obtained in the Chinese co-inhabitant set are shown. As the Table 5 shows, the degrees of overlap are very low, indicating a likely influence of country of collection, although the result may also be influenced by a likely lesser co-inhabitation featured in the European sets. Also, note that the results (overlaps) in Table 3 and Table 5 are comparable, as they have been computed using the same reference (results found using the co-inhabitant Chinese set).

Table 5. Percentages of the AMR-associated accessory genes, core genome genes, and overall number of genes, identified by machine learning, that are shared between the Chinese co-inhabiting *E. coli* and *S. enterica* isolates (serving as the reference) and the Chinese isolates of not co-inhabiting *E. coli* and the European isolates of *E. coli* and *S. enterica* (EFFORT + ENGAGE). Hypothetical genes were excluded, each column gives the percentages of genes in common between the sets.

Dataset	Antibiotics	Accessory genes	Core genome genes	Overall number of genes
E. coli (EFFORT)	CHL	0.00%	14.71%	14.71%
	CIP	4.17%	15.71%	15.08%
	NAL	2.70%	13.21%	13.39%
S. enterica (ENGAGE)	CTX	0.00%	1.79%	1.90%
	TET	2.44%	3.66%	3.48%
	TMP	2.44%	3.35%	3.65%

A lower overlap (14.19%) was observed when comparing EFFORT not co-inhabiting *E. coli* with Chinese co-inhabiting *E. coli*, compared to Chinese not co-inhabiting *E. coli* vs Chinese co-inhabiting *E. coli*. (39.37%). In the comparison involving the EFFORT *E. coli*, 246 genes were found in common, representing accessory genes and core genome genes related to SNPs across all antibiotic models. Interestingly, only four of these genes are known antibiotic resistance genes: *kpdE* (aminoglycoside), *emrB* (fluoroquinolone), *mdtA* (multi drug resistance), *eptA* (peptide).

Lower rates of overlap were observed when comparing the results for the ENGAGE not-necessarily co-inhabiting *S. enterica* isolates with the Chinese co-inhabiting *S. enterica* isolates. On average, only 3.01% of genes were shared across three antimicrobial models, with the following antibiotic-specific percentages: TMP = 3.65%, CTX = 1.90%, and TET = 3.48%. In more detail, 55 genes were found in common, representing accessory genes and core genome genes related to SNPs across all antibiotic models. There was only one known antibiotic resistance gene (*oprM*), which is actually associated with multi-drug resistance.

The scenario presented by Table 3 and Table 5 indicates that both country of collection and presence/absence of co-inhabitation have an influence in the genetic elements discovered by machine learning. Note that these results are consistent with what illustrated in the previous response for the more conventional analysis of MGEs (i.e. country of collection appears to be more influential than presence/absence of co-inhabitation). The results are particularly consistent with the possible differences existing in the microbial ecosystems between Chinese and European anthropogenic settings, driven by variations in antibiotic usage, intervention strategies (such as vaccines against *Salmonella*). There is a possibility also that country-related differences may influence the rate of co-inhabitation, possibly shedding some light on the currently unknown co-inhabitation situation in the ENGAGE *S. enterica* dataset. Nevertheless, we acknowledge that our analysis may be influenced by further confounding effects. To better

explore and separate potential confounding factors a long-term evolutionary case-control study would be required.

Updates to the submission

Results Lines 513-542: **“Machine learning reveals differences in AMR-associated mutations and accessory genes between co-inhabiting *E. coli* and *S. enterica* in chickens and those of *E. coli* and *S. enterica* that do not necessarily co-inhabit.**

Similar to what done when searching for plasmids and mobile genetic elements carrying AMR genes and present in co-inhabiting *E. coli* and *S. enterica*, we wanted to investigate whether the results (identified genetic elements associated with AMR) predicted by the machine learning pipeline trained with the Chinese *E. coli* and *S. enterica* cohorts (with confirmed co-inhabitation of the bacterial species) would change if the same pipeline was trained with different cohorts (e.g., from another country, and/or with no confirmed co-inhabitation). To this purpose, we resorted to our Chinese *E. coli* not co-inhabiting chicken isolates and again to the previously described EFFORT (*E. coli*) and ENGAGE (*S. enterica*) European datasets, and investigated how the predictions would change once the machine learning pipelines were trained using those sets. The full results are given in the **Supplementary Note 4**.

In summary, when looking at the European cohorts and at the Chinese not co-inhabiting isolates, the comparative analyses were only possible for subsets of antibiotics with sufficient data amount for ML training. The results (fully documented in the supplementary materials) indicate a larger overlap of results (genetic elements shared by *E. coli* and *S. enterica*, identified by ML) when comparing co-inhabiting and not co-inhabiting isolates collected in China, with respect to overlaps observed when comparing results from co-inhabiting Chinese isolates with European isolates. In more detail, a 39.67% prediction overlap was observed when comparing Chinese co-inhabiting and not co-inhabiting datasets, whilst a 14.19% prediction overlap was observed when comparing Chinese co-inhabiting *E. coli* vs European not co-inhabiting *E. coli* (EFFORT dataset), and 3% prediction overlap was observed when comparing Chinese co-inhabiting *S. enterica* vs European not-necessarily co-inhabiting *S. enterica* (ENGAGE). As stated earlier, the absence of co-inhabitation in the European samples cannot be fully demonstrated and may be unlikely at least for the ENGAGE set, hinting at the observed differences being likely influenced in no small amount by country of collection, although more investigation should be needed to fully assess the influence of both co-inhabitation and country of collection in the machine learning predictions.

Discussion, lines 731-734: “Our results indicate that sharing is influenced by many factors, and we have reported evidence on the effects of co-inhabitation, as well as country of collection, possibly reflecting differences in regulations, interventions and overall microbial ecology.”

Supplementary material Lines 202-341: **“Supplementary Note 4:**

To investigate whether the results (identified genetic elements associated with AMR) predicted by the machine learning pipeline trained with the Chinese *E. coli* and *S. enterica* cohorts (with confirmed co-inhabitation of the bacterial species) would change if the same pipeline was trained with different cohorts (e.g., from another country, and/or with no co-inhabitation) we performed a further investigation by training the machine learning pipeline multiple times. In the “test” runs, using Chinese *E. coli* and *S. enterica* isolates (with confirmed co-inhabitation) for training; in the “control” runs, using the Chinese not co-inhabiting chicken isolates and the European datasets retrieved from the EFFORT and ENGAGE projects (previously described in **Supplementary Note 3**). Then, we compared the results to observe the degree of overlap of the predictions (identified genetic elements associated to AMR).

Methods and datasets

We first selected the set of *E. coli* and *S. enterica* isolates that had been confirmed to be co-inhabitants in our Chinese cohort. This set consisted of instances in which positive cultures verified the presence of both *E. coli* and *S. enterica* species in chicken, with a total of 98 such cases. The 98 isolates included: 47 chicken faeces, 23 chicken caecal droppings, 16 chicken carcass, 12 chicken feather. Since the number of confirmed co-inhabiting isolates in chicken faeces alone was limited to 47, insufficient for training the machine learning model, and by including chicken caecal droppings we increase to 70 (still insufficient), we opted to perform the training using all confirmed co-inhabiting isolates found in chicken sources (including faeces, ceca, carcasses, and feathers). Then, we selected the not co-inhabiting isolates from the same Chinese cohort of chicken isolates. Next, we selected the data from the European EFFORT and ENGAGE sets to perform an analogous training of the machine learning models, in the control experiments.

The same machine learning pipeline (see **Materials and Methods** in the manuscript) was used in both the test experiments (Chinese *E. coli* and Chinese *S. enterica* data) and control experiments (not co-inhabiting Chinese and European data).

Machine learning results for the Chinese co-inhabiting sets

When training the machine learning models on the Chinese data, the only occasional modification to the pipeline was the p-value of the chi-square test, set to 0.05 for *E. coli* due to the small number of isolates. The following features were encoded as inputs to the machine learning models: presence/absence of accessory genes, core genome SNPs and intergenic region SNPs (therefore covering both mutations and genes). When training the machine learning models for the test experiments (Chinese data), for some of the 28 antimicrobials, because of the limited number of isolates, the distribution of resistance/susceptibility profiles was skewed, with most isolates being either susceptible or resistant. Consequently, these data could not be used for constructing predictive models. Given the above limitations we were able to run the machine learning pipeline on 3 antibiotics (chloramphenicol - CHL,

ciprofloxacin - CIP, nalidixic acid – NAL) for the co-inhabiting *E. coli* dataset and on 5 antibiotics (cefotaxime – CTX, gentamycin – GEN, minocycline – MIN, tetracycline – TET and trimethoprim / sulfamethoxazole – SXT) for the co-inhabiting *S. enterica* dataset. **Figs. S17 and S18** show the performance metrics and the number of features for the co-inhabiting *E. coli* and *S. enterica* Chinese datasets. The performance metrics for all classifiers are in **Table S4**, while the selected features for each antibiotic model are in **Table S11**.

The application of the remaining part of the pipeline, led to the results illustrated in **Table S12** for the Chinese datasets.

Machine learning results for the Chinese not co-inhabiting sets

When training the machine learning models on the Chinese not co-inhabiting sets, no modification was done to the original proposed pipeline. There were 337 not co-inhabiting *E. coli* isolates from chicken sources, a sufficient number to train the machine learning pipeline to create predictor models for *E. coli*. However, there were only 17 not co-inhabiting *S. enterica* isolates from chicken sources, which was not sufficient to train the machine learning models. Therefore, the analyses focused solely on training the machine learning models for *E. coli*. The same features used in the test experiments (with co-inhabitant isolates) were encoded as inputs, i.e., presence/absence of accessory genes, core genome SNPs and intergenic region SNPs. The predictive models for some of the antibiotics could not be trained, because of excessive disproportion of resistance or susceptibility phenotypes for those antibiotics. We were able to complete the training for 15 antibiotics (amikacin – AMI, aztreonam – AZM, ceftazidime - CAZ, ceftazidime/clavulanic acid – CAZ-C, ceftoxitin – CFX, chloramphenicol - CHL, cefotaxime – CTX, cefepime – FEP, gentamycin – GEN, kanamycin – KAN, minocycline – MIN, nalidixic acid – NAL, streptomycin – STR, sulphafurazole – SUL and trimethoprim/sulfamethoxazole – SXT). The performance metrics and the number of features selected as relevant by the machine learning predictors of AMR to specific antibiotics in *E. coli*, are summarised in **Fig. S19**. These models ran using the not co-inhabiting *E. coli* isolates. The performance metrics for all classifiers are in **Table S4**, while the selected features for each antibiotic model are in **Table S13**.

The application of the pipeline, led to the results illustrated in **Table S14** for the Chinese not co-inhabiting datasets.

Comparison between Chinese not co-inhabiting and Chinese co-inhabiting machine learning results

The comparison between not co-inhabiting and co-inhabiting results is summarised in **Table S15**, where the degree of overlap of identified genetic elements is illustrated as percentages. The overlaps (i.e., percentages of genes identified by machine learning in both datasets) were estimated by mapping the findings back to the pangenome. Note that in the table, the co-inhabiting sample (test) is taken as the reference (as opposed to selecting the control, i.e. the not co-inhabiting sample). This is to allow comparison of overlaps with the

following tests. Despite the overlaps in **Table S16** being in some cases relatively large, the differences in prediction generated by the machine learning models provide an additional hint at co-inhabitation having a likely effect on the machine learning prediction.

An average of 39.37% of genes were found in common across antibiotics. This includes both accessory and core genes. In more detail, 723 genes were found in common, representing accessory genes and core genes related to SNPs across all antibiotic modes. Interestingly, only five of these genes are known antibiotic resistance genes: *kpdE* (aminoglycoside), *emrB* (fluoroquinolone), *mdtA* (multi drug resistance), *cpxA* (multi drug resistance), *acrB* (multi drug resistance).

Machine learning results for the European sets

As anticipated earlier, a second set of experiments was conducted using machine learning, to incorporate in the analysis the investigation of a possible influence of country of collection. The aforementioned EFFORT and ENGAGE datasets were used to train machine learning models in order to obtain further control results. It should be reiterated that the EFFORT *E. coli* set is labelled as “not co-inhabiting” due to the less likely presence of *S. enterica* consequent to bird treatment; whilst the ENGAGE *S. enterica* set is labelled as not-necessarily co-inhabiting because the presence of *E. coli* is unknown.

For *E. coli* (EFFORT dataset) we had 206 isolates with AMR phenotype data, of which 7 antibiotics (ampicillin - AMP, chloramphenicol - CHL, ciprofloxacin - CIP, nalidixic acid – NAL, sulfisoxazole – SUL, tetracycline – TET and trimethoprim – TMP) had enough isolates in each class for the machine learning models to be trained. For *S. enterica* (ENGAGE dataset) 92 isolates were available with AMR phenotype data, of which 5 antibiotics (ampicillin – AMP, chloramphenicol – CHL, cefotaxime – CTX, tetracycline – TET, trimethoprim – TMP) had enough isolates in each class for the machine learning models to be trained. **Figs. S20 and S21** and **Table S16**, report the performance metrics and the features identified by the machine learning pipeline for the control experiments. The performance metrics for all classifiers are in **Table S4**, while the selected features for each antibiotic model are in **Table S17**.

Comparison between European non-necessarily co-inhabiting and the Chinese co-inhabiting machine learning results

In **Table S18**, the degrees of overlap between the results on the European sets and the original results obtained in the Chinese co-inhabitant set are shown. As the **Table S18** shows, the degrees of overlap are very low, indicating a likely influence of country of collection, although the result may also be influenced by a likely lesser co-inhabitation featured in the European sets. Also, note that the results (overlaps) in **Table S15** and **Table S18** are comparable, as they have been computed using the same reference (results found using the co-inhabitant Chinese set).

A lower overlap (14.19%) was observed when comparing EFFORT not co-inhabiting *E. coli* with Chinese co-inhabiting *E. coli*, compared to Chinese not co-inhabiting *E. coli* vs Chinese co-inhabiting *E. coli*. (39.37%). In the comparison involving the EFFORT *E. coli*, 246 genes were found in common, representing accessory genes and core genome genes related to SNPs across all antibiotic models. Interestingly, only four of these genes are known antibiotic resistance genes: *kpdE* (aminoglycoside), *emrB* (fluoroquinolone), *mdtA* (multi drug resistance), *eptA* (peptide).

Lower rates of overlap were observed when comparing the results for the ENGAGE not-necessarily co-inhabiting *S. enterica* isolates with the Chinese co-inhabiting *S. enterica* isolates. On average, only 3.01% of genes were shared across three antimicrobial models, with the following antibiotic-specific percentages: TMP = 3.65%, CTX = 1.90%, and TET = 3.48%. In more detail, 55 genes were found in common, representing accessory genes and core genome genes related to SNPs across all antibiotic models. There was only one known antibiotic resistance gene (*oprM*), which is actually associated with multi-drug resistance.

The scenario presented by **Table S15** and **Table S18** indicates that both country of collection and presence/absence of co-inhabitation have an influence in the genetic elements discovered by machine learning. Note that these results are consistent with what illustrated in the previous response for the more conventional analysis of MGEs (i.e. country of collection appears to be more influential than presence/absence of co-inhabitation). The results are particularly consistent with the possible differences existing in the microbial ecosystems between Chinese and European anthropogenic settings, driven by variations in antibiotic usage, intervention strategies (such as vaccines against *Salmonella*). There is a possibility also that country-related differences may influence the rate of co-inhabitation, possibly shedding some light on the currently unknown co-inhabitation situation in the ENGAGE *S. enterica* dataset. Nevertheless, we acknowledge that our analysis may be influenced by further confounding effects. To better explore and separate potential confounding factors a long-term evolutionary case-control study would be required.”

Fig. S17. The supervised machine learning pipeline effectively predicts the resistance/susceptibility profiles of co-inhabiting *E. coli* isolates. This prediction is based on a dataset comprising 98 isolates collected from a Chinese cohort and originating from various chicken sources, including faeces, ceca, carcasses, and feathers. Machine learning performance results are given for six performance indicators: (I) area under the curve AUC, (II) accuracy, (III) sensitivity, (IV) Specificity, (V) Cohen's kappa score, and (VI) precision. All indicators were calculated from 30 training runs for each antimicrobial model. (VII) Number and type of features (accessory genes, core genome SNPs and intergenic region SNPs) selected by each antibiotic model. The results shown are for the best classifier RBF-SVM, as defined by the Nemenyi test (Fig. S10c)

Fig. S18. The supervised machine learning pipeline effectively predicts the resistance/susceptibility profiles of co-inhabiting *S. enterica* isolates. This prediction is based on a dataset comprising 98 isolates collected from a Chinese cohort and originating from various chicken sources, including faeces, ceca, carcasses, and feathers. Machine learning performance results are given for six performance indicators: (I) area under the curve AUC, (II) accuracy, (III) sensitivity, (IV) Specificity, (V) Cohen's kappa score, and (VI) precision. All indicators were calculated from 30 training runs for each antimicrobial model. (VII) Number and type of features (accessory genes, core genome SNPs and intergenic region SNPs) selected by each antibiotic model. The results shown are for the best classifier Linear SVM, as defined by the Nemenyi test (Fig. S10d)

Fig. S19. The machine learning pipeline effectively predicts the resistance/susceptibility profiles of Chinese not co-inhabiting *E. coli* isolates. This prediction is based on a dataset comprising 337 isolates collected from a Chinese cohort and originating from various chicken sources, including faeces, ceca, carcasses, and feathers. Machine learning performance results are given for six performance indicators: (I) area under the curve AUC, (II) accuracy, (III) sensitivity, (IV) Specificity, (V) Cohen’s kappa score, and (VI) precision. All indicators were calculated from 30 training runs for each antimicrobial model. (VII) Number and type of features (accessory genes, core genome SNPs and intergenic region SNPs) selected by each antibiotic model. The results shown are for the best classifier RBF-SVM, as defined by the Nemenyi test (Fig. S10e)

Fig. S20. The supervised machine learning pipeline effectively predicts the resistance/susceptibility profiles of EFFORT *E. coli* isolates. This prediction is based on a dataset comprising 206 isolates collected from a European cohort and originating from chicken faeces, with strong likelihood of absence of co-inhabiting *S. enterica*. Machine learning performance results are given for six performance indicators: (I) area under the curve AUC, (II) accuracy, (III) sensitivity, (IV) Specificity, (V) Cohen's kappa score, and (VI) precision. All indicators were calculated from 30 training runs for each antimicrobial model. (VII) Number and type of features (accessory genes, core genome SNPs and intergenic region SNPs) selected by each antibiotic model. The results shown are for the best classifier Logistic Regression, as defined by the Nemenyi test (Fig. S10f)

Fig. S21. The supervised machine learning pipeline effectively predicts the resistance/susceptibility profiles of ENGAGE *S. enterica* isolates (from samples which may or may not contain also co-inhabiting *E. coli*). The prediction is based on a dataset comprising 92 isolates collected from an Italian cohort and originating from chicken faeces and cecum. Machine learning performance results are given for six performance indicators: (I) area under the curve AUC, (II) accuracy, (III) sensitivity, (IV) Specificity, (V) Cohen's kappa score, and (VI) precision. All indicators were calculated from 30 training runs for each antimicrobial model. (VII) Number and type of features (accessory genes, core genome SNPs and intergenic region SNPs) selected by each antibiotic model. The results shown are for the best classifier RBF-SVM, as defined by the Nemenyi test (Fig. S10g)

Table S11. (Separate File)

Features selected by the ML pipeline as correlated to the resistance-susceptibility of either *E. coli* to a panel of 3 antibiotics or *S. enterica* to a panel of 5 antibiotics using the Chinese co-inhabiting chicken isolates. The features are either accessory genes, core genome SNPs or intergenic region SNPs. The number “1” indicates that the feature was found in the respective antibiotic model.

Table S12.

Number of AMR-associated accessory genes, core genome SNPs, and core genome genes found by the machine learning pipeline, in the Chinese co-inhabiting *E. coli* and *S. enterica* isolates, with hypothetical genes excluded.

Dataset	Antibiotics	Accessory genes	Core genome SNPs	Core genome Genes
Chinese co-inhabiting E. coli	CHL	46	618	431
	CIP	50	443	324
	NAL	38	502	360
Chinese co-inhabiting S. enterica	CTX	41	257	227
	GEN	26	220	205
	MIN	18	89	87
	SXT	42	210	185
	TET	42	290	255

Table S13. (Separate File)

Features selected by the ML pipeline as correlated to the resistance-susceptibility of *E. coli* to a panel of 15 antibiotics using the Chinese not co-inhabiting chicken isolates. The features are either accessory genes, core genome SNPs or intergenic region SNPs. The number “1” indicates that the feature was found in the respective antibiotic model.

Table S14.

Number of AMR-associated accessory genes, core genome SNPs, and core genome genes found by the machine learning pipeline, in the Chinese not co-inhabiting datasets for *E. coli*, with hypothetical genes excluded.

Dataset	Antibiotics	Accessory genes	Core genome SNPs	Core genome Genes
Chinese not co-inhabiting E. coli	AMI	94	832	478
	AZM	70	255	163
	CAZ	4	32	30
	CAZ-C	24	183	118
	CFX	70	902	500
	CHL	98	1007	574
	CTX	83	510	331
	FEP	31	81	55
	GEN	43	173	128
	KAN	64	295	209
	MIN	87	802	488
	NAL	81	730	461
	STR	110	946	547
	SUL	101	809	467
	SXT	56	411	249

Table S15.

Percentages of AMR-associated accessory genes, core genome genes, and overall number of genes identified by machine learning in both the Chinese co-inhabiting and not co-inhabiting *E. coli* isolates. The co-inhabiting set is taken as reference. Hypothetical genes were excluded, each column gives the percentages of genes in common between the sets.

Datasets	Antibiotics	Accessory genes	Core genome genes	Overall number of genes
Not co-inhabiting Chinese E. coli	CHL	8.70%	43.39%	40.04%
	NAL	5.26%	42.22%	38.69%

Table S16.

Number of AMR-associated accessory genes, core genome SNPs, and core genome genes found by the machine learning, in the European EFFORT and ENGAGE datasets, with hypothetical genes excluded.

Dataset	Antibiotics	Accessory genes	Core genome SNPs	Core genome Genes
E. coli (EFFORT)	AMP	100	744	378
	CHL	105	685	368
	CIP	98	672	284
	NAL	83	533	277
	SUL	39	272	154
	TET	87	719	335
	TMP	59	321	177
S. enterica (ENGAGE)	AMP	21	266	106
	CHL	18	217	81
	CTX	18	118	71
	TET	16	244	74
	TMP	34	265	102

Table S17. (Separate File)

Features selected by the ML pipeline as correlated to the resistance-susceptibility of either *E. coli* to a panel of 7 antibiotics or *S. enterica* to a panel of 5 antibiotics using the European not co-inhabiting chicken isolates. The features are either accessory genes, core genome SNPs or intergenic region SNPs. The number “1” indicates that the feature was found in the respective antibiotic model.

Table S18.

Percentages of the AMR-associated accessory genes, core genome genes, and overall number of genes, identified by machine learning, that are shared between the Chinese co-inhabiting *E. coli* and *S. enterica* isolates from China (serving as the reference) and the Chinese isolates of not confirmed co-inhabiting *E. coli* and the European isolates of *E. coli* and *S. enterica* (EFFORT + ENGAGE). Hypothetical genes were excluded, each column gives the percentages of genes in common between the sets.

Dataset	Antibiotics	Accessory genes	Core genome genes	Overall number of genes
E. coli (EFFORT)	CHL	0.00%	14.71%	14.71%
	CIP	4.17%	15.71%	15.08%
	NAL	2.70%	13.21%	13.39%
S. enterica (ENGAGE)	CTX	0.00%	1.79%	1.90%
	TET	2.44%	3.66%	3.48%
	TMP	2.44%	3.35%	3.65%

Reviewer #1, Question #3

To make evolutionary inferences, an important assumption is that the bacteria of interest are native residents of the environment, subject to selections (e.g. antibiotic usage) and adaptations (e.g. HGT) long enough for (micro)evolution to occur and accumulate. The observation that *E. coli* isolates were not clonal may indicate different isolates were introduced by different flocks. Which isolates were sampled from the same flock?

Both our *E. coli* and the *S. enterica* Chinese datasets were clonal (the salmonella data presenting higher clonality than the *E. coli* data), hence the correction we used for the population structure in our machine learning pipeline (see Results line 180-200 and Material and Methods lines 965-975). We apologise if the wording used in our original sentence, below, made this unclear.

Original sentence: "Using the standard association index (I^S_A) to measure for clonality in the population, we found the I^S_A to be 0.2126 (p -value < 0.0001) at whole cohort level and 0.1313 (p -value < 0.0001) at ST type level, indicating there was little clonality observed amongst our *E. coli* cohort"

In the original paper we described the use of the standard association index (I^S_A) to measure for clonality in the population (Haubold & Hudson, 2000; Peng et al. 2022). For the *E. coli* cohort, we found the I^S_A to be 0.2126 (p -value < 0.0001) at whole-cohort level and 0.1313 (p -value < 0.0001) at ST type level, indicating presence of clonality. For the *Salmonella* cohort the I^S_A was 0.9077 (p -value < 0.0001) at whole-cohort level and 0.3883 (p -value < 0.0001) at ST type level indicating stronger clonality compared to the *E. coli*.

Each farm had its own flock. Supplementary Table 1 indicated the farms where each sample was collected for both *E. coli* and *Salmonella*.

To address the reviewer comment we conducted an additional analysis to determine the similarity or dissimilarity in clonality between each flock. To do this we calculated the I^S_A at the cohort level for each flock (Table 6 below) for both the *E. coli* and the *Salmonella* datasets. Table 1 indicates that in all the flocks there was clonality in *E. coli* and *Salmonella* samples, and also that for most farms the clonality in each farm was not much different from the overall clonality of each species, which indicates that generally the different flocks do not have a direct influence in the clonality of our data. For the Shandong farms, a greater amount of clonality was observed in *E. coli*, however, the population correction employed in the machine learning pipeline would correct for this and reduce any impact.

Table 6. ISA at cohort level for all the samples (overall) and at each farm. The N/A indicates that there were insufficient samples to calculate the ISA. The symbol “-” indicates that there were no samples for that farm.

	E. coli	Salmonella
Overall	0.2126	0.9106
Henan 1	0.2967	0.8988
Henan 2	0.2476	0.9481
Henan 3	0.3325	0.9788
Shandong 1_1	0.3682	-
Shandong 1_2	0.4784	-
Shandong 2	0.3461	N/A
Shandong 3	0.3287	N/A
Shandong 4	0.2628	-
Liaoning 1	0.3102	0.8463
Liaoning 2	0.2903	-
Liaoning 3	0.2819	N/A

Updates to the Submission

Results Lines 185-188: “Using the standard association index (I_A^S) to measure for clonality in the population (Haubold & Hudson, 2000; Peng et al. 2022), for the *E. coli* cohort we found the I_A^S to be 0.2126 (p -value < 0.0001) at whole-cohort level and 0.1313 (p -value < 0.0001) at ST type level, indicating presence of clonality”

Results Lines 198-200: “For *S. enterica*, the I_A^S of 0.9077 (p -value < 0.0001) at whole-cohort level and 0.3883 (p -value < 0.0001) at ST type level indicated stronger clonality compared to *E. coli*.”

References

Haubold B, Hudson RR. LIAN 3.0: detecting linkage disequilibrium in multilocus data. Linkage Analysis. *Bioinformatics* **16**, 847-848 (2000).

Peng Z, et al. Whole-genome sequencing and gene sharing network analysis powered by machine learning identifies antibiotic resistance sharing between animals, humans and environment in livestock farming. *PLoS Comput Biol* **18**, e1010018 (2022).

Reviewer #1 – Question #4

Why *E. coli* isolates were not serotyped and analyzed (e.g. MCRA dating) by serotypes similar to Salmonella?

The *E. coli* isolates were serotyped in silico. The H and O serogroups can be found in Table S1, although this was not mentioned in the original submission, this has now been corrected. The *E. coli* isolates exhibited a wide range of diversity, leading to the identification of 246 distinct serotypes within the cohort. The number of isolates per serotype varied between 11 and 1. Due to the relatively low number of isolates per serotype, we decided to utilize phylogroups instead. Phylogroups offered fewer subgroupings and thus allowed for a more meaningful comparison with the *S. enterica* dataset. In the updated manuscript we have also added the analysis of the two of the most prevalent serotypes O83:H42 and O8:H16. The serotype O83:H42 contained 11 sequences from six different farms and strictly clustered by farm, with the MRCA within farms being less than one year, while the MRCA between farms ranged between 2 - 20 years. For the serotype O8:H16, containing 8 isolates from five different farms, some inter-farm clustering was present with the MRCA between isolates from SD1 and HN2 being less than 1 year.

Updates to the Submission

Supplementary Material lines 3-18: **“Supplementary Note 1:** To construct the phylogenetic tree, the following steps were taken. The serotyping of the *E. coli* isolates revealed 246 distinct serotypes, with varying numbers of isolates per serotype, ranging from 1 to 11. Due to the limited number of isolates within each serotype, we were unable to assess clustering within serotypes. Therefore, we conducted an assessment of isolate clustering based on larger phylogroups to draw more meaningful conclusions. Isolates were primarily from phylogroups A (n=181) or B1(n=221), both associated with commensal *E. coli*, however all other phylogroups were also seen in smaller numbers: C (n=10), D (n=32), E (n=28), F (n=31), G (n=9), Clade I (n=2), Unknown (n=4).”

Results lines 233-240: “The vast phylogenetic and genomic (SNPs) diversity of *E. coli* and *S. enterica* was supported by the Bayesian divergence analysis results showing that *S. enterica* strains were more recently evolved, with MRCAs of *S. enterica* being around 5 years (Fig. S3), compared to MRCAs for *E. coli* phylogroups (Fig. S4) which was in the range of hundreds of years. For the two largest *E. coli* serotypes, for which a temporal signal was present (O83:H42 and O8:H16), the MRCA being also longer, compared to *S. enterica* (approximately 20 years) with some clustering by farm observed (Fig. S5).”

Material and Methods Lines 1052-1054: “In addition to the phylogroup analyses, the BEAST analysis was also conducted for two *E. coli* serotypes, O83:H42 and O8:H16, using the same models used for the phylogroup analyses.”

Supplementary Figure S5:

Fig S5. Bayesian divergence analysis of *E. coli* isolates in two serotypes (A) O83:H42; (B) O8:H16. the farm the isolates were taken from, sample source, and the timepoint for each sample are displayed as coloured strips.

Reviewer #1 – Question #5

Not sure how much the machine learning and metabolic modeling part of the study contributes to the key conclusion that convergent evolution of AMR is driven by co-inhabitation of *E. coli* and *Salmonella*. This part of the paper feels only loosely adhered to the main storyline, more speculative in nature, and almost separable as a standalone study.

In the original manuscript the machine learning analysis was not done to obtain a direct proof that co-inhabitation is a driver of convergent evolution. Rather, the analysis was done to obtain separate information on the degree of sharing (between *E. coli* and *S. enterica*) of genetic elements related to AMR, as sharing between the bacterial species may be an indicator supporting convergence, though not being a proof of it. Moreover, our original machine learning analysis was conducted across all hosts and environments, and considering all samples, not just those with co-inhabitant bacteria. That is, it was not aimed at identifying any correlation with co-inhabitation.

However, in the updated manuscript, in response to a previous question by this Reviewer (question #2), we applied our machine learning pipeline to search for possible correlations between sharing of genetic elements related to AMR and co-inhabitation, adding to this analysis also the search for correlations with country of collection (because of potentially different microbial ecology and uses of antimicrobials, also suggested by the Reviewer). The results of this analyses have been discussed in the response to question #2.

A separate comment must be made on the general importance of the machine learning (ML) analysis as complementary, not redundant, analysis method with respect to the more conventional search for mobile ARGs also illustrated in the manuscript.

The conventional procedure that we applied to find mobile ARGs consisted of searching for known ARGs (presence/absence) in the bioinformatics data, followed by identifying the presence of MGEs within a given distance threshold from them. Then, positive or negative resistance phenotypes were associated to the patterns of mobile ARGs so that pattern similarities across *S. enterica* and *E. coli* isolates could be investigated, under the assumption that similarity of pattern may imply coevolution. On the contrary, to apply the ML pipeline, we did not start from known ARGs and we did not search for MGEs. We simply gave the whole dataset of features obtained by bioinformatics, combined with resistance/susceptibility data, to ML-powered regressors/classifiers to train them into predictors of resistance. Once trained, introspection on the fitted models was used to retrieve the genetic elements found to be most important, within the fitted models, in affecting the prediction results. In general, these genetic elements could be anything, not necessarily MGEs, ARGs or plasmids, because the only criterion for retrieval is relative importance within the fitted prediction model.

The benefits of our ML pipeline are multiple: a) the conventional analysis is constrained to using known genetic elements. This is often insufficient when dealing with rapidly mutating resistance. Conversely, ML is not constrained to annotated genes, and simply returns anything which may be correlated to the observed change in phenotype; b) the ML pipeline returns results (genetic elements) ranked by importance (in the internal prediction model), hence it provides a richer information set to drive further analyses.

Note that there may or may not be overlap between the genetic elements returned by the conventional analysis and those returned by the ML pipeline. Overlapping elements may be interpreted as further confirmation of putative importance, new elements found by ML may provide hints at new investigation targets. Clearly the ML results require extra work, because genetic elements are returned solely based on relevance within the fitted prediction model, whilst a high correlation score may be coincidental, e.g., due to other interactions or confounding factors in the model. Each genetic element candidate returned by ML therefore needs further screening, usually aimed at reconstructing putative functionality, in order to obtain further information which may help ascertain the role of the element within the observed phenotype (resistance). This further investigation can be done for example with the help of genome-scale metabolic models, as illustrated in this work.

To better help understand the importance of using our machine learning pipeline + genome-scale metabolic model to the analysis (in addition to the more conventional search for mobile ARGs), in the updated paper we have documented the identification of three promising new genes, present in both *E. coli* and *S. enterica* isolates extracted from samples with proven co-inhabitation, that are showing to be correlated to AMR, at least according to our preliminary data (see also response to Reviewer #3, Question #2). Importantly, these three genes could not be found using the more conventional method of search for mobile ARGs, because they are not present in the AMR databases. The reason they were found by the ML pipeline is that, as stated earlier, the ML method does not start from known genetic elements, but instead processes all the available information in the search for the strongest correlations to the phenotype.

Updates to the submission

The following text was added to the Introduction section, lines 96-100, to introduce the reader to the role of ML compared with the previous analysis:

Introduction lines 96-100: “The new data analysis pipeline was designed to provide a wider perspective on the relationships between genetic elements of *E. coli* and *S. enterica* isolates and AMR, by data mining all the possible correlations between SNPs in the coding and non-coding regions of the core genome as well as all the accessory genes, and AMR resistance/susceptibility to multiple antibiotics.”

The following text was added to the results section, lines 365-369, with the same clarification purpose:

Results lines 365-369: “Note that whilst the previously described analysis specifically focused on plasmids and mobile ARGs, the machine learning pipeline was designed to provide a wider perspective by investigating all the SNPs in the coding and non-coding regions of the core genome as well as all the accessory genes in search for correlation to AMR traits. “

A section in the Supplementary Material (Supplementary Notes 6) was added to reply to Reviewer #3 Question #3 in relation to the comparison of our custom ML pipeline and existing GWAS method, also provides further details on the justification for using our method.

Reviewer #2 – Question #1:

An excellent, well-presented and interesting study. The authors have certainly done a very nice job of analyzing their data and as far as I can judge in a very clear and appropriate way. Some of their conclusions might be due to confounding factors that they have not had the possibility to adjust for, but considering the data available, all conclusions seem solid.

We thank the reviewer for providing their valuable assessment of our work. Acknowledging the reviewer's feedback, we concur that confounding factors unaccounted for in our analysis may have influenced our results. Amongst those factors more likely to be influential:

- Factors related to the place of collection, for example: a) differences in rearing practices between the covered Chinese regions and farms (including feed, antibiotic treatment, use of net housing vs cage housing, etc.), with the added complexity that some of these practices may be poorly documented; b) differences in environmental variables, such as temperature and humidity. For example, in recent works (Baker *et al.*, 2023, Maciel-Guerra *et al.*, 2022) we have demonstrated that there is a correlation between AMR in *E. coli* residing in the chicken gut, and environmental temperature and humidity in the barn; c) other region-related differences, possibly even more difficult to account for (for example differences in local microbial ecologies across regions).
- Factors related to health and welfare of the birds. Note that our samples were pooled, thus averaging the conditions of the birds in the net or cage of collection. However, there likely is bias still due to this factor, as our sample size (977 samples over 10 farms) is very little considering that a typical Chinese barn may host between 12k and 33k birds, a farm usually has multiple barns, the collection covered 10 farms, and there are 6-7 weeks of bird life + slaughtering, which in this research were sampled in only three time points.
- Issues in sample collection and analysis. For example, when culturing it is very difficult to demonstrate absence of co-inhabitation of the two bacterial species *E. coli* and *S. enterica*, as such conclusion was inferred when one of the two species would not be found in the sample (when in reality the species might have been present earlier in time, or in undetectable quantities, and might have been found with more frequent/larger sampling – we have commented extensively on this point in the answer to Reviewer #1 question #1). Several other confounding factors may be involved in the analysis.

In the updated manuscript, we introduced further analyses aimed at investigating in greater depth the possible influences of country of collection, as well as the influence of presence/absence of co-inhabitation. These analyses are described in detail in the answers to Reviewer #1 – Question #1 and Reviewer #1 - Question #2 and summarised in the following. More in detail, we investigated how our prediction of important genetic elements linked to AMR would change, if the investigation transitioned from considering the Chinese cohort of co-inhabiting isolates (“test” set), to considering three new “control” sets, featuring various

likelihoods of absence of co-inhabitation, as well as different countries of collection. The three new control sets included: a) Chinese isolates (*E. coli* and *S. enterica*) with absence of co-inhabitation (though not rigorously demonstrated, as commented earlier and in the response to Reviewer #1 – Question #1); b) European isolates of *E. coli*, sampled from five countries, with more likely absence of *S. enterica*, as the birds had been vaccinated against Salmonella (data from the previous EFFORT project (Leekitcharoenphon, Johansson et al. 2021)); b) European isolates of *S. enterica*, sampled from one country (Italy), with less likely absence of *E. coli* (data from the previous ENGAGE project (Alba, Leekitcharoenphon et al. 2020)). More details on these datasets are now reported in the updated supplementary material. In summary, the estimation of genetic elements possibly tied to AMR changed when using “test” vs “control” data, with country of collection having a stronger influence on the prediction, compared to presence-absence of co-inhabitation, the latter still being a significant influencing factor.

In summary (see also response to Reviewer #1 Question #2) in terms of difference between predictions (quantified in opposite terms, as percentage of prediction overlap – i.e., proportion of genetic elements equally predicted using the test and control set), the largest difference of predictions was observed for *S. enterica*: European ENGAGE (not necessarily co-inhabiting) vs the Chinese (co-inhabiting) set, with only 3.01% overlap. The second largest difference was observed for *E. coli*: European EFFORT (not co-inhabiting) vs the Chinese (co-inhabiting) set: with 14.19% overlap. The smallest difference was observed for the combined *E. coli* and *S. enterica* Chinese sets: co-inhabiting vs not co-inhabiting (39%). These results seem to indicate that the country of collection is a stronger factor than co-inhabitation, but the latter is still appreciable, in particular when the samples are collected within the same country.

Other potentially relevant confounding factors, for example from the list of candidate factors provided above, were not analysed as the available sampling size and coverage was deemed insufficient to cope with the complexity of the issue (as correctly appreciated by this Reviewer) and should be targeted as the subject of future work.

References

Baker M, et al. Machine learning and metagenomics reveal shared antimicrobial resistance profiles across multiple chicken farms and abattoirs in China. *Nature Food* 4, 707-720 (2023).

Maciel-Guerra A, et al. Dissecting microbial communities and resistomes for interconnected humans, soil, and livestock. *The ISME Journal*, (2022).

Leekitcharoenphon P, et al. Genomic evolution of antimicrobial resistance in *Escherichia coli*. *Scientific Reports* 11, 15108 (2021).

Alba P, et al. Molecular epidemiology of *Salmonella* Infantis in Europe: insights into the success of the bacterial host and its parasitic pESI-like megaplasmid. *Microb Genom* 6, (2020).

Updates to the submission

The description of the new experiments (tests vs controls) aimed at investigating the influence of presence/absence of co-inhabitation and country of origin has been added to the Material and Methods, results and supplementary material in the updated submission. The contents of these updates have already been illustrated in the answers to Reviewer #1, questions #1 and #2. To better clarify that the provided results may be affected by unknown confounding factors, and that these may not be fully addressed with the current data, we also added the new **Supplementary Note 6** (Supplementary Material lines 402-433), to specifically discuss confounding factors.

Supplementary Material lines 402-433: *“Confounding factors possibly influencing the illustrated results*

It is important to discuss potential confounding factors, unaccounted for in our analysis, which may have influenced our results. Amongst those more likely to be influential, we can list:

- Factors related to the place of collection, for example: a) differences in rearing practices between the covered Chinese regions and farms (including feed, antibiotic treatment, use of net housing vs cage housing, etc.), with the added complexity that some of these practices may be poorly documented; b) differences in environmental variables, such as temperature and humidity). For example, in recent works works (Baker *et al.*, 2023, Maciel-Guerra *et al.*, 2022) we have demonstrated that there is a correlation between AMR in *E. coli* in the chicken gut, and environmental temperature and humidity in the barn; c) other region-related differences, possibly even more difficult to account for (for example differences in local microbial ecologies across regions).
- Factors related to health and welfare of the birds. Note that our samples were pooled, thus averaging the conditions of the birds in the net or cage of collection. However, there likely is still bias due to this factor, as our sample size (977 samples over 10 farms) is very little considering that a typical Chinese barn may host between 12k and 33k birds, a farm usually has multiple barns, the collection covered 10 farms, and there are 6-7 weeks of bird life + slaughtering, which in this research were sampled in only three time points.
- Issues in sample collection and analysis. For example, when culturing it is very difficult to demonstrate absence of co-inhabitation of the two bacterial species *E. coli* and *S. enterica*, as such conclusion was inferred when one of the two species would not found in the sample (when in reality the species might have been present earlier in time, or in undetectable quantities – see elsewhere in this discussion) . Several other confounding factors may be involved in the analysis.

There will clearly be many other factors that have not been explicitly listed above. Whilst the effects of the country of origin have been touched upon, with quantitative experiments (see Results and Supplementary Material, and a similar analysis was run on various degrees of likelihood of absence of co-inhabitation, also in the supplementary material), several others

of the factors listed above, as well as many unlisted others, remain unaddressed. Clearly the main limitation is in the size/coverage our sample, which was sufficient to support the presented conclusions, but insufficient in terms of size and diversity, to guarantee sufficient coverage of all the potential conditions that should be investigated in order to isolate the many factors that are surely involved in the study of genetic convergence of AMR.”

Other questions

I only have a few very minor comments:

- Gentamycin should be spelled gentamicin.

Thanks. Corrected on Results lines 405-409 and 428-429.

Results Lines 405-409: “Performance across all metrics was high, Fig. S9b-g, and gentamicin (GEN) performed best across all metrics. Unlike for *E. coli* the poorer performing antibiotics models came from three different antibiotic classes: beta lactams (AMC), diaminopyrimidines (SXT) and tetracycline (TET) and poorer performing models did not necessarily have a low number of features, Fig. S11b.”

Results Lines 428-429: “Most notably, for gentamicin, the presence of 9 of the top 10 genes were positively correlated to the susceptible phenotype.”

- Page 3, line 99 (and throughout the manuscript): the authors conclude that their observations identify co-evolution. Would just be nice to acknowledge that this could also be that these features is a pre-requisite for occupying this niche.

We have now updated introduction lines 100-105 and discussion lines 709-713 to acknowledge also this suggestion.

Introduction Lines 100-105: “We found that most isolates of *E. coli* and *S. enterica*, within the same host and environment, possessed the same AMR-carrying MGEs, which also appear to have co-evolved, which in real-world settings pose a high risk of AMR transfer to humans and the environment (Manyi-Loh et al. 2018). These AMR-carrying MGEs could also potentially be a pre-requisite for the bacteria to occupy the same host and environment, as the horizontal gene transfer process may drive the development of host adaptation (Silva et al. 2012, Stokes et al. 2011).”

Discussion lines 709-713: “In particular we found IncHI2, a plasmid known to be a very important carrier of AMR genes (Coelho et al. 2012) in a number of *S. enterica* and *E. coli* strains isolated from the same samples. Bayesian divergence analysis highlighted two recently co-evolved clusters of IncHI2 plasmids isolated from both species, indicative of relatively recent HGT between species which may also indicate a host adaptation process for these species.”

References

Coelho A, *et al.* Role of IncHI2 plasmids harbouring blaVIM-1, blaCTX-M-9, aac(6')-Ib and qnrA genes in the spread of multiresistant *Enterobacter cloacae* and *Klebsiella pneumoniae* strains in different units at Hospital Vall d'Hebron, Barcelona, Spain. *Int J Antimicrob Agents* **39**, 514-517 (2012).

Manyi-Loh C, Mamphweli S, Meyer E, Okoh A. Antibiotic use in agriculture and its consequential resistance in environmental sources: potential public health implications. *Molecules* **23**, 795 (2018).

Silva, Claudia, Magdalena Wiesner, and Edmundo Calva. "The importance of mobile genetic elements in the evolution of *Salmonella*: pathogenesis, antibiotic resistance and host adaptation." *Salmonella-A Diversified Superbug* 231 (2012): 254.

Stokes, Hatch W., and Michael R. Gillings. "Gene flow, mobile genetic elements and the recruitment of antibiotic resistance genes into Gram-negative pathogens." *FEMS microbiology reviews* 35, no. 5 (2011): 790-819.

- Page 24, line 698: Couldn't this enrichment step select specifically for specific *E. coli* strains?

The reviewer is correct when pointing out that the step conducted on lines 851-853 will select only *E. coli* strains. The specific selection process for the *Salmonella* strains was indicated on lines 867-869. Both methods are indicated below for clarity.

Methods lines 851-853: "For isolation of *E. coli*, approximately 1mL dilution (of any of the above samples) was then added to 9 mL *E. coli* (EC) broth (Luqiao Inc.) and incubated at 37°C for 16-20h in order to enumerate presumptive *E. coli* populations."

Methods lines 867-869: "Similarly, for isolation of *S. enterica*, approximately 1 mL enriched dilution was added to 9 mL RV broth and SC broth (Luqiao Inc.) and incubated at 42 °C and 37 °C for 16-20 h, respectively, in order to enumerate presumptive *Salmonella* populations."

Reviewer #3 – Question #1

Baker/Zhang/Guerra and colleagues detail the genetic epidemiology of n=661 *Salmonella enterica* and *Escherichia coli* isolated from 10 chicken farms located in China over a 2.5 year period. They carry out phylogenetic, phylodynamic, AMR gene screening, and plasmid reconstruction analyses to identify AMR transmission events. The authors characterise the population structures of the two organisms circulating in the study settings, and present data suggestive of AMR and plasmid transmission between the two organisms. The authors also use machine learning approaches to identify genes which might contribute to AMR resistance or sensitivity, and demonstrate that many of these appear to be essential to the bacteria analysed via metabolic modelling. While the genome and AMR phenotype data collected are highly novel, the study has several weaknesses that need addressing as detailed in my comments below.

We thank the reviewer for appreciating the novelty of the genome and AMR phenotype data.

In the response below we clarify further aspects of novelty which we had failed to highlight in the original manuscript, and address the weaknesses identified by the Reviewer. In addition, in the updated submission, we have performed additional analyses and obtained further results that in our opinion may help to reduce these weaknesses.

Reviewer #3 – Question #2

At present the rationale for the machine learning work is unclear as the authors do not present data on the correlation between AMR genotypes and phenotypes detailing if the molecular determinants of AMR detected using existing tools sufficiently explain the AMR phenotypes observed. While the machine learning approach used appears sound, the results mostly report already discovered molecular determinants of AMR described in existing databases. Subsequently, the data lack novelty, and relevance of some of the reported genetic loci remains unclear as there hasn't been any experimental validation carried out.

To answer the first part of this question, it is useful to firstly provide a more in-depth illustration of how the ML method differs in objectives and implementation with respect to the more conventional method of identification of molecular determinants. A similar illustration has been also provided in response to Reviewer #1 – Question #5.

The conventional procedure that we applied to find mobile ARGs consisted of searching for known ARGs (presence/absence) in the bioinformatics data, followed by identifying the presence of MGEs within a given distance threshold from them. Then, positive or negative resistance phenotypes were associated to the patterns of mobile ARGs so that pattern similarities across *S. enterica* and *E. coli* isolates could be investigated, under the assumption that similarity of pattern may imply coevolution.

On the contrary, to apply the ML pipeline, we did not start from known ARGs and we did not search for MGEs. We simply gave the whole dataset of features obtained by bioinformatics, combined with resistance/susceptibility data, to ML-powered regressors/classifiers to train them into predictors of resistance. Once trained, introspection on the fitted models was used to retrieve the genetic elements found to be most important, within the fitted models, in relation to affecting the prediction results. In general, these genetic elements could be anything, not necessarily MGEs, ARGs or plasmids, because the only criterion for retrieval is relative importance within the fitted prediction model.

The benefits of our ML pipeline are multiple: a) the conventional analysis is constrained to using known genetic elements. This is often insufficient when dealing with rapidly mutating resistance. Conversely, ML is not constrained to annotated genes, and simply returns anything which may be correlated to the observed change in phenotype; b) the ML pipeline returns results (genetic elements) ranked by importance (in the internal prediction model), hence it provides a richer information set to drive further analyses. Clearly though, there may or may not be overlap between the genetic elements returned by the conventional analysis and those returned by the ML pipeline, thus candidate elements found by ML usually need further screening.

In our specific case, the ML analysis identified a large number of accessory genes and mutations correlated to AMR phenotypes, many of which were found to be novel. In *E. coli*, of the 4119 genes selected by ML, only 1% were known AMR genes (as found in public databases, CARD(Alcock, Raphenya et al. 2020), ARG-Annot(Gupta, Padmanabhan et al. 2014), Resfinder(Bortolaia, Kaas et al. 2020) and AMRfinder(Feldgarden, Brover et al. 2019)) whilst 99% were novel AMR associations. For *S. enterica*, of the 3501 genes selected, only 1.1% were known molecular determinants of AMR as found in public databases, whilst 98.9% were novel associations.

In summary, for both *E. coli* and *S. enterica*, with machine learning, we were able to identify thousands of known and unknown genetic elements, relying solely on the power of correlation strength, going well beyond the approach of simply focusing on preselected genes with known AMR annotation. In addition, because with ML we could observe SNPs in coding and non-coding core genome regions, and accessory genes, we could investigate co-occurrences as well as different mechanisms of mutation and horizontal gene transfer possibly involved in AMR. We have reported all this information in the new Supplementary Table 6 in the supplementary material.

The last part of this answer addresses the issue with the relevance of the reported genetic loci and the lack of experimental validation.

Admittedly, in the original paper: we ran the ML pipeline to identify a set of genetic elements involved with the phenotype (resistance), ranked by relative importance/influence on the phenotype; we selected the top 10% (highest-ranked), resulting in a list of core genes, accessory genes and SNPs in the core genome (the latter mapped back to genes) and finally

we further reduced the set by running genome-scale metabolic models. However, we did not perform any experimental validation on any gene found in the final selected set.

In the updated manuscript, we selected three genes from the remaining subset using the following criteria: a) present in the *E. coli* and *S. enterica* co-inhabiting set; b) harbouring non-synonymous SNPs; c) significantly impacting reaction flux when knocked out, as highlighted by the genome-scale metabolic model; d) availability of the knockout strain. This resulted in the three candidate genes, top-ranked by ML for the antibiotics ampicillin and doxycycline, namely: *hisA* (obtained by mapping the SNP: P109Q back onto the genome), *argI* (SNP: A153T) and *fhuB* (SNP: N47D). Interestingly, neither of these three genes is currently present in any AMR databases.

HisA is a $(\beta\alpha)_8$ barrel enzyme that catalyzes the Amadori rearrangement of *N'*-[(5'-phosphoribosyl)formimino]-5-aminoimidazole-4-carboxamide ribonucleotide (ProFAR) to *N'*-[(5'-phosphoribulosyl) formimino]-5-aminoimidazole-4-carboxamide-ribonucleotide (PRFAR) in the histidine biosynthesis pathway. The amino acid position 109, identified by machine learning as associated to the doxycycline resistance phenotype, is found in the histidine biosynthesis domain (1-234). The gene *argI* encodes for catalysing ornithine to produce a substrate L-citrulline for argininosuccinate synthetase that is part of the arginine biosynthesis (ornithine to produce a substrate L-citrulline) (Thongbhubate et al., 2021). The gene *fhuB* is part of the ferrichrome iron transporter Fhu operon, containing genes *fhuA*, *fhuB*, *fhuC*, and *fhuF*. The protein FhuB forms the transmembrane complex comprised of ten transmembrane helices creating a pore for transmembrane transport.

For the experimental validation of these three genes, we used the publicly available Keio gene *E. coli* knockout collection (Baba et al, 2006). Both *argI* and *fhuB* mutants exhibited increased antimicrobial susceptibility compared to the parental wild type, as shown by a reduced yield after 20 hours of incubation in LB in the presence of ampicillin at a concentration of 3.55 $\mu\text{g/ml}$ with a confidence level of 99.9% (Figure 6 below). Similarly, the *hisA* mutant failed to reach the same optical densities as the wild type in the presence of doxycycline over a range of concentrations between 0.19 and 1.77 $\mu\text{g/ml}$ with a confidence level of 99% (Figure 6 below). While this analysis is limited to only knocking out genes rather than introducing mutations, these results give a preliminary experimental support for possible the involvement of these genetic elements as AMR-determinants, as predicted by machine learning. It is important to recognise that none of the experimentally observed changes in growth were sufficient to cause a change to the resistance/susceptibility classification of the *E. coli* strain. It is likely that individual mutations alone are not strong enough influencers of resistance, but may require the co-presence and co-occurrence of further AMR-associated genetic determinants (Johnson et al, 2010, Mobegi et al, 2017).

Figure 6. Susceptibility against antimicrobials of selected mutants with knocked-down genes vs wild type. *E. coli* strains (*E. coli* K12, $\Delta argI$, $\Delta fhuB$ and $\Delta hisA$ mutants) were inoculated in LB in 96-well plates amended with ampicillin or doxycycline at the indicated concentrations and incubated for 20 h at 35°C without shaking, after which the optical density at 600 nm (OD600) was measured. Significance of *t*-tests: *** $p \leq 0.001$; ** $0.001 \leq p \leq 0.01$; * $0.01 \leq p \leq 0.05$; N.S not significant.

Following knockout, further considerations may be made in relation to the possible importance of the three genes *hisA*, *argI* and *fhuB* in relation to AMR.

Updates to the submission

A new paragraph addressing validation was added to the updated manuscript, at the bottom of the Results section.

Results lines 668-681: “To perform a preliminary validation of the AMR-related genetic elements (SNPs and accessory genes) identified by the ML pipeline, we selected three of the top-ranked SNP candidates, prioritizing the following aspects in relation to the associated genes: i) presence in the *E. coli* and *S. enterica* co-inhabiting set; ii) harbouring non-synonymous SNP; c) significantly impacting reaction flux when knocked out, as highlighted by the genome-scale metabolic model; d) availability of the knockout strain. This resulted in three genes, top-ranked by ML for the antibiotics ampicillin and doxycycline, namely: *hisA* (obtained by mapping the SNP: P109Q back onto the genome), *argI* (SNP: A153T) and *fhuB* (SNP: N47D). To our knowledge, neither of these three genes is currently present in any AMR databases.

Although knocking out the entire gene cannot replace the in vitro study of the effects of the individual mutations onto the AMR phenotype, our preliminary results showed that all three gene mutants exhibited increased antimicrobial susceptibility compared to the parental wild type (Supplementary Note 5).”

The new Supplementary Note 5 was added, reading as follows:

Supplementary Material lines 343-383: **“Supplementary Note 5:**

Further investigations on the validity of the machine learning results

To obtain some experimental validation on the results obtained by running the ML method, we selected three genes from the top-ranked 10% identified by the pipeline, using the following criteria: a) present in the *E. coli* and *S. enterica* co-inhabiting set; b) harbouring non-synonymous SNPs; c) significantly impacting reaction flux when knocked out, as highlighted by the genome-scale metabolic model; d) availability of the knockout strain. This resulted in the three candidate genes, top-ranked by ML for the antibiotics ampicillin and doxycycline, namely: *hisA* (obtained by mapping the SNP: P109Q back onto the genome), *argI* (SNP: A153T) and *fhuB* (SNP: N47D). Interestingly, neither of these three genes is currently present in any AMR databases.

HisA is a $(\beta\alpha)_8$ barrel enzyme that catalyzes the Amadori rearrangement of *N'*-[(5'-phosphoribosyl)formimino]-5-aminoimidazole-4-carboxamide ribonucleotide (ProFAR) to *N'*-[(5'-phosphoribulosyl) formimino]-5-aminoimidazole-4-carboxamide-ribonucleotide (PRFAR) in the histidine biosynthesis pathway. The amino acid position 109, identified by machine learning as associated to the doxycycline resistance phenotype, is found in the histidine biosynthesis domain (1-234). The gene *argI* encodes for catalysing ornithine to produce a substrate L-citrulline for argininosuccinate synthetase that is part of the arginine biosynthesis (ornithine to produce a substrate L-citrulline) Thongbhubate et al., 2021). The gene *fhuB* is part of the ferrichrome iron transporter Fhu operon, containing genes *fhuA*, *fhuB*, *fhuC*, and *fhuF*. The protein FhuB forms the transmembrane complex comprised of ten transmembrane helices creating a pore for transmembrane transport.

For the experimental validation of these three genes, we used the publicly available Keio gene *E. coli* knockout collection (Baba et al, 2006). Bacterial growth was determined by using an automated luminometer-spectrometer (TECAN). Overnight cultures of *E. coli* strains were diluted 1:1000 in fresh lysogeny broth (LB), and 0.2 ml cultures were grown at 35°C in 96-well microtitre plates. Turbidity was measured as optical density at 600 nanometres (OD600). Antimicrobial susceptibility to ampicillin and doxycycline was determined by broth microdilution and interpreted according to the Clinical & Laboratory Standards Institute interpretive criteria (CLSI 2009). *E. coli* ATCC 25922 was used as a control for the antimicrobial susceptibility testing.

Both *argI* and *fhuB* mutants exhibited increased antimicrobial susceptibility compared to the parental wild type, as shown by a reduced yield after 20 hours of incubation in LB in the presence of ampicillin at a concentration of 3.55 µg/ml with a confidence level of 99.9% (**Fig. S22**). Similarly, the *hisA* mutant failed to reach the same optical densities as the wild type in the presence of doxycycline over a range of concentrations between 0.19 and 1.77 µg/ml with a confidence level of 99% (**Fig. S22**). While this analysis is limited to only knocking out genes rather than introducing mutations, these results give a preliminary experimental support for possible the involvement of these genetic elements as AMR-determinants, as predicted by

machine learning. It is important to recognise that none of the experimentally observed changes in growth were sufficient to cause a change to the resistance/susceptibility classification of the *E. coli* strain. It is likely that individual mutations alone are not strong enough influencers of resistance, but may require the co-presence and co-occurrence of further AMR-associated genetic determinants (Johnson et al, 2010, Mobegi et al, 2017).”

Fig. S22 Susceptibility against antimicrobials of selected mutants with knocked-down genes vs wild type. *E. coli* strains (*E. coli* K12, $\Delta argI$, $\Delta fhuB$ and $\Delta hisA$ mutants) were inoculated in LB in 96-well plates amended with ampicillin or doxycycline at the indicated concentrations and incubated for 20 h at 35°C without shaking, after which the optical density at 600 nm (OD600) was measured. Significance of *t*-tests: *** $p \leq 0.001$; ** $0.001 \leq p \leq 0.01$; * $0.01 \leq p \leq 0.05$; N.S not significant.”

References

- Chen, Y., Lu, H., Zhang, N., Zhu, Z., Wang, S., & Li, M. (2020). PremPS: Predicting the impact of missense mutations on protein stability. *PLoS computational biology*, 16(12), e1008543.
- Ha, Y., McCann, M. T., Tuchman, M., & Allewell, N. M. (1997). Substrate-induced conformational change in a trimeric ornithine transcarbamoylase. *Proceedings of the National Academy of Sciences*, 94(18), 9550-9555.
- Pettersen, E. F., Goddard, T. D., Huang, C. C., Couch, G. S., Greenblatt, D. M., Meng, E. C., & Ferrin, T. E. (2004). UCSF Chimera—a visualization system for exploratory research and analysis. *Journal of computational chemistry*, 25(13), 1605-1612.
- Pires, D. E., Ascher, D. B., & Blundell, T. L. (2014a). DUET: a server for predicting effects of mutations on protein stability using an integrated computational approach. *Nucleic acids research*, 42(W1), W314-W319.
- Pires, D. E., Ascher, D. B., & Blundell, T. L. (2014b). mCSM: predicting the effects of mutations in proteins using graph-based signatures. *Bioinformatics*, 30(3), 335-342.
- Rodrigues, C. H., Pires, D. E., & Ascher, D. B. (2018). DynaMut: predicting the impact of mutations on protein conformation, flexibility and stability. *Nucleic acids research*, 46(W1), W350-W355.

Worth, C. L., Preissner, R., & Blundell, T. L. (2011). SDM—a server for predicting effects of mutations on protein stability and malfunction. *Nucleic acids research*, 39(suppl_2), W215-W222.

Thongbhubate K, Irie K, Sakai Y, Itoh A, Suzuki H. Improvement of putrescine production through the arginine decarboxylase pathway in *Escherichia coli* K-12. *AMB Express* **11**, 168 (2021).

Reviewer #3 – Question #3

Finally, it is unclear from the current text why the authors constructed their own machine learning pipeline for these analyses when existing GWAS approaches (e.g. PubMed IDs/PMIDs: 30419019, 27572646, 27887642, 29401456, 27633831, 30535304) have been validated which account for population structure and multiple testing correction. The advantages of the machine learning approach used over GWAS are unclear from the current manuscript text.

A primary difference is that conventional GWAS methods only consider one feature type, most commonly SNPs, or k-mers (Jaillard et al 2018 pmid: 30419019; Earle et al 2016 pmid: 27572646, Lees et al 2016 pmid: 27633831, Lees et al 2018 pmid: 30535304), accessory genes (Brynildsrud et al 2016 pmid: 27887642), or any type characterised by a binary state (Collins et al 2018 pmid: 29401456). To address multiple feature types, separate runs are made on each type and the results combined (Collins et al 2018 pmid: 29401456, Pearcy et al 2022). On the contrary, the approach presented in this paper implies processing both SNPs and accessory genes within the same model and at the same time. By simultaneously addressing multiple types of genetic elements, our correlation models can potentially capture more complex relationships between genotype and phenotype, compared to what possible by considering only one feature type (Tam et al. 2019; Vermeulen 2023). For example, by including both SNPs and accessory genes, our method can generate models that can simultaneously highlight both mutation and horizontal gene transfer events underlying AMR. Further investigations may be needed to assess whether the simultaneous consideration of different types of genetic elements leads to superior results or not, compared to combining results obtained from running separate analysis on individual types, like done in (Collins et al 2018 pmid: 29401456, Pearcy et al 2022).

Another main difference is that conventional GWAS methods typically perform a sequential scan of the candidate genetic elements, testing each for correlation with the phenotype. Many pre-processing or post-processing approaches are available to cope with hidden dependencies amongst elements, but ultimately only limited multi-collinearity can be handled. On the contrary, our method considers all genetic elements at the same time, regardless of co-dependencies, and solves a regression problem producing a model where the genetic elements act as factors, contributing in different measures to phenotype manifestation, where co-dependencies are implicitly captured.

The general advantages of using machine learning (ML) to investigate the correlation between genotype and phenotype have been recognised by the scientific literature, to the point that even recent GWAS methods have begun to incorporate machine learning (Enoma et al 2022). The main benefits of ML (whether used with GWAS or within a different method like ours) are the following:

- ML can better handle regression problems where the dimensionality of the feature set is larger than the available number of observations;
- ML can better handle multi-collinearity of the input features, because co-dependencies are implicitly captured by the model and do not need to be addressed. To test for presence of multicollinearity in our data, we computed variance inflation factors (VIFs) using the StatsModels package in Python (VIF values >2.5 are considered indicative of multi-collinearity). We found significant multi-collinearity in all antibiotic models: in E coli models, VIFs ranged from 2 to 265 (mean: 109), with many values at infinity. In S. enterica models, VIFs ranged from 7 to 50 (mean: 20), with many at infinity.

In addition to the advantages intrinsic to the adoption of ML, illustrated above, our method provides two additional features that result in a significant advantage with respect to alternatives:

- when ML is implemented within our phenotype predictors using technologies such as extra trees, random forests and decision trees, we analyse the features finally selected by ML to be part of the correlation model to determine their relative importance in affecting the phenotype, using the Gini feature importance method. The assessment of relative importance leads to feature ranking, which in turn is very useful to remove low-ranked features while at the same time highlighting the most important ones, essentially supporting dimensionality reduction (Enoma et al. 2022; Menze et al. 2009);
- for unbalanced sets, i.e., when the numerosity of observations available for any phenotype manifestation are significantly different with each other (for example, when there may be more susceptible isolates vs the resistant ones), we have developed a solution based on the SMOTE technique to generate additional observations and rebalance the sets, which is essential for ensuring appropriate representativeness of the classes that must be modelled in the regression problem.

Further details and applications of our proposed ML approach to link genotype information to AMR phenotypes can be found in previous work (Pearcy et al. 2022, Peng et al. 2022, Wang et al. 2021).

Updates to the manuscript

Results Lines 369-373: "Firstly, a test for presence of multicollinearity was carried out on the ML features. The variance inflation factors (VIFs) were computed using the StatsModels package in Python. Multicollinearity was found in all antibiotic models: in *E. coli* models VIFs ranged from 2 to 265 (mean 109), with many at infinity; in *S. enterica* models VIFs ranged from 7 to 50 (mean 20), with many at infinity."

Supplementary Material lines 386-399: "*Proposed machine learning pipeline compared to existing GWAS approaches*

Recent works published by us and others have demonstrated that culture based approaches with whole genome sequencing of individual pathogens, antibiotic susceptibility testing and ML techniques are effective predictors of genomic characteristics linked to AMR for both *E. coli* isolates (Her & Wu 2018, Hyun et al. 2020, Pearcy et al. 2022, Peng et al. 2022) and other bacteria (Kavvas et al. 2018, Kavvas et al. 2020, Liu et al. 2020, ValizadehAslani et al. 2020, Wang et al. 2022, Wang et al. 2021). Therefore, this work developed a methodology based on ML to accurately associate multiple genomic features (accessory genes, core genome SNPs and intergenic region SNPs) with *E. coli* and *Salmonella* antibiotic resistances. Compared to conventional GWAS methods only processing a single genetic element type at each run (most frequently SNPs), our method simultaneously considers SNPs and accessory genes, allowing to capture complex interactions in relation to both mutation and horizontal gene transfer. The use of machine learning allows for better handling problems with large numbers of features compared to the number of observations, and can better cope with multi-collinearity of in the feature set, while also providing tools for ranking and filtering the identified features based on influence on the phenotype."

References

Brynildsrud, O., Bohlin, J., Scheffer, L. *et al.* Rapid scoring of genes in microbial pan-genome-wide association studies with Scoary. *Genome Biol* **17**, 238 (2016)

Collins, Caitlin, and Xavier Didelot. "A phylogenetic method to perform genome-wide association studies in microbes that accounts for population structure and recombination." *PLoS computational biology* 14, no. 2 (2018): e1005958.

Earle, Sarah G., Chieh-Hsi Wu, Jane Charlesworth, Nicole Stoesser, N. Claire Gordon, Timothy M. Walker, Chris CA Spencer et al. "Identifying lineage effects when controlling for population structure improves power in bacterial association studies." *Nature microbiology* 1, no. 5 (2016): 1-8.

Enoma, David O., Janet Bishung, Theresa Abiodun, Olubanke Ogunlana, and Victor Chukwudi Osamor. "Machine learning approaches to genome-wide association studies." *Journal of King Saud University-Science* 34, no. 4 (2022): 101847.

Her, H.-L. & Wu, Y.-W. A pan-genome-based machine learning approach for predicting antimicrobial resistance activities of the *Escherichia coli* strains. *Bioinformatics* 34, i89-i95 (2018).

Jaillard, Magali, Leandro Lima, Maud Tournoud, Pierre Mahé, Alex Van Belkum, Vincent Lacroix, and Laurent Jacob. "A fast and agnostic method for bacterial genome-wide association studies: Bridging the gap between k-mers and genetic events." *PLoS genetics* 14, no. 11 (2018): e1007758.

Kavvas, E. S. et al. Machine learning and structural analysis of *Mycobacterium tuberculosis* pan-genome identifies genetic signatures of antibiotic resistance. *Nat. Commun.* 9, 4306 (2018).

Kavvas, E. S., Yang, L., Monk, J. M., Heckmann, D. & Palsson, B. O. A biochemically-interpretable machine learning classifier for microbial GWAS. *Nat. Commun.* 11, 2580 (2020).

Lees, J., Vehkala, M., Välimäki, N. *et al.* Sequence element enrichment analysis to determine the genetic basis of bacterial phenotypes. *Nat Commun* 7, 12797 (2016)

Lees, John A., Marco Galardini, Stephen D. Bentley, Jeffrey N. Weiser, and Jukka Corander. "Pyseer: a comprehensive tool for microbial pangenome-wide association studies." *Bioinformatics* 34, no. 24 (2018): 4310-4312.

Liu, Z. et al. Evaluation of Machine Learning Models for Predicting Antimicrobial Resistance of *Actinobacillus pleuropneumoniae* From Whole Genome Sequences. *Front. Microbiol.* 11 (2020).

Menze, B.H., Kelm, B.M., Masuch, R. et al. A comparison of random forest and its Gini importance with standard chemometric methods for the feature selection and classification of spectral data. *BMC Bioinformatics* 10, 213 (2009)

Pearcy, N. et al. Genome-scale metabolic models and machine Learning reveal genetic determinants of antibiotic resistance in *Escherichia coli* and unravel the underlying metabolic adaptation mechanisms. *mSystems* 6, e00913-00920, (2021).

Peng, Z. et al. Whole-genome sequencing and gene sharing network analysis powered by machine learning identifies antibiotic resistance sharing between animals, humans and environment in livestock farming. *PLoS Comput. Biol.* 18, e1010018 (2022).

Tam, V., Patel, N., Turcotte, M. *et al.* Benefits and limitations of genome-wide association studies. *Nat Rev Genet* 20, 467–484 (2019).

ValizadehAslani, T., Zhao, Z., Sokhansanj, B. A. & Rosen, G. L. Amino Acid k-mer Feature Extraction for Quantitative Antimicrobial Resistance (AMR) Prediction by Machine Learning and Model Interpretation for Biological Insights. *Biology* 9, 365 (2020).

Vermeulen, Sander. "Bacterial GWAS: A Comprehensive Assessment of Challenges, Methods and Alternatives." Master's thesis (2023)

Wang, W. et al. Novel SCCmec type XV (7A) and two pseudo-SCCmec variants in foodborne MRSA in China. *J. Antimicrob. Chemother.* (2022).

Wang, W. et al. Whole-genome sequencing and machine learning analysis of *Staphylococcus aureus* from multiple heterogeneous sources in China reveals common genetic traits of antimicrobial resistance. *mSystems* 6, e01185-01120 (2021)

Reviewer #3 – Question #4

Moreover, it is not clear how informative the knockout models are given that many of the associations were reported in the context of substitutions rather than nonsense mutations. However, this is difficult to assess as only a list of the genes containing these was provided as supplementary data.

Any mutation, not just the nonsense ones, can cause in structural destabilisation and/or loss of function, i.e., resulting in a knockout condition. Synonymous mutations, for example in mRNA, can influence the stability and secondary structure affecting translation efficiency. Non synonymous mutations in the protein can alter its 3D structure, causing unfolding or affecting ligand binding capacity, again resulting in loss of function.

Considering our data, For *E. coli* there were six nonsense mutations selected by the ML: *argI* 335, *fadJ* 714, *potB* 286, *trmJ* 246, *ybiT* 531, *yfiH* 196. Of these, two genes (*argI* and *fadJ*) used were found in the GSM model. Neither were found to be essential genes or significant in the flux variability analysis.

For *S. enterica* there was only one nonsense mutation selected by the ML, which was in a hypothetical protein (named group_39 in our analysis), this was not used as input for the GSM model.

We concur that not enough information was given in the original manuscript to properly assess the identified genetic elements. We have now done additional analyses to characterize the core genome SNPs found by ML in both *E. coli* and *S. enterica* datasets.

In detail, we performed *in silico* characterization of both the synonymous and non-synonymous mutations to better understand their roles, and finally validated three of the most significant ones identified by the machine learning and genome-scale metabolic (GMS) models via preliminary experimental analysis. For the *E. coli* dataset, the machine learning pipeline selected a total of 11315 mutations over 1763 different genes across the 17 antimicrobials studied with an AUC over 0.9. Out of these 11315 mutations, we have 1306 that are non-synonymous and 10009 that are synonymous. Moreover, 54 non-synonymous

mutations were related to 36 genes that were significant in the GSM model analysis (which considered only the most important 10% of features). For the *S. enterica* dataset, the machine learning pipeline selected a total of 3496 core genome SNPs over 1403 different genes across the 13 antimicrobials models with an AUC over 0.9. Out of these 3496 mutations, we have 478 mutations that are non-synonymous and 1181 that are synonymous. Moreover, 12 of the 478 non-synonymous mutations were related to 11 genes that were significant in the GSM model analysis. In addition, for each of the non-synonymous and synonymous substitutions, we have now analysed and reported, if synonymous or not synonymous, the specific antibiotic model where the substitution was identified as significant, the p value and coefficient value obtained by the associated machine learning antibiotic models, the gene name and description, the aminoacid position in the corresponding protein, whether the substitution was related to genes that were significant in the GSM model (and which ones), the StringDB protein annotation (functional annotation of the protein), the StringDB gene ontology function annotation, and (when relevant) the metabolic pathway associated to the gene/protein where the substitution is occurring. All this information has been added in new Supplementary Table 6 , already mentioned in relation to a previous question (Reviewer #3, Question #2)

To provide an example of how substitutions may cause knockout even without being nonsense mutations, we illustrate in the following the structural analysis done on HisA, one of the three proteins investigated in the response to a previous question by this Reviewer (Reviewer #3, Question #2).

We used the published X-ray crystal structure of *S. enterica* (homolog to *E. coli*) pdbID: 5AC6 (2.0 Å resolution) and performed a 3D mutational modelling of the SNP P109Q to see if the transition from proline (P) to glutamine (Q) would destabilise the protein structure. We observed that Q109 is indeed causing a $\Delta\Delta G = -0.886$ kcal/mol, indicative of structural destabilisation. Furthermore, the prediction shows Q109 establishing a new van der Waals, hydrophobic interaction with Val106, which is actively involved in binding the ligand ProFAR (Figure 7 below), potentially implying a change of function, if the ligand binding capacity is altered.

Figure 7. Structural analysis of the SNP P109Q, identified as relevant to AMR by the ML pipeline. The *S. enterica* 3D structure (pdbID: 5AC6) of the associated protein HisA is shown in pink. P109Q is shown in red (ball and stick model), whilst interacting aminoacids are labelled. The interaction of P109Q with V106 (active residue, shown in yellow) changes between the P and Q mutations. The $\Delta\Delta G$ values associated to the mutations are also

reported. Green lines represent van der Waals, hydrophobic interactions; blue lines represent hydrogen bonds.

Updated to the submission

Additional information, for example related to the synonymous / non-synonymous or nonsense nature of the mutations was added in new Supplementary Table 6, already mentioned in relation to a previous question (Reviewer #3, Question #2)

References

Pettersen, E. F., T. D. Goddard, C. C. Huang, E. C. Meng, G. S. Couch, T. I. Croll, J. H. Morris and T. E. Ferrin (2021). "UCSF ChimeraX: Structure visualization for researchers, educators, and developers." Protein Sci **30**(1): 70-82.

Baba, T., T. Ara, M. Hasegawa, Y. Takai, Y. Okumura, M. Baba, K. A. Datsenko, M. Tomita, B. L. Wanner and H. Mori (2006). "Construction of Escherichia coli K-12 in-frame, single-gene knockout mutants: the Keio collection." Mol Syst Biol **2**: 2006.0008.

Bernhofer, M., C. Dallago, T. Karl, V. Satagopam, M. Heinzinger, M. Littmann, T. Olenyi, J. Qiu, K. Schütze, G. Yachdav, H. Ashkenazy, N. Ben-Tal, Y. Bromberg, T. Goldberg, L. Kajan, S. O'Donoghue, C. Sander, A. Schafferhans, A. Schlessinger, G. Vriend, M. Mirdita, P. Gawron, W. Gu, Y. Jarosz, C. Trefois, M. Steinegger, R. Schneider and B. Rost (2021). "PredictProtein - Predicting Protein Structure and Function for 29 Years." Nucleic Acids Res **49**(W1): W535-w540.

Chen, Y., H. Lu, N. Zhang, Z. Zhu, S. Wang and M. Li (2020). "PremPS: Predicting the impact of missense mutations on protein stability." PLoS Comput Biol **16**(12): e1008543.

McWilliam, H., W. Li, M. Uludag, S. Squizzato, Y. M. Park, N. Buso, A. P. Cowley and R. Lopez (2013). "Analysis tool web services from the EMBL-EBI." Nucleic acids research **41**(W1): W597-W600.

Page, A. J., C. A. Cummins, M. Hunt, V. K. Wong, S. Reuter, M. T. Holden, M. Fookes, D. Falush, J. A. Keane and J. Parkhill (2015). "Roary: rapid large-scale prokaryote pan genome analysis." Bioinformatics **31**(22): 3691-3693.

Pancotti, C., S. Benevenuta, G. Birolo, V. Alberini, V. Repetto, T. Sanavia, E. Capriotti and P. Fariselli (2022). "Predicting protein stability changes upon single-point mutation: a thorough comparison of the available tools on a new dataset." Brief Bioinform **23**(2).

Pettersen, E. F., T. D. Goddard, C. C. Huang, E. C. Meng, G. S. Couch, T. I. Croll, J. H. Morris and T. E. Ferrin (2021). "UCSF ChimeraX: Structure visualization for researchers, educators, and developers." Protein Sci **30**(1): 70-82.

Pires, D. E., D. B. Ascher and T. L. Blundell (2014). "DUET: a server for predicting effects of mutations on protein stability using an integrated computational approach." Nucleic acids research **42**(W1): W314-W319.

Pires, D. E., D. B. Ascher and T. L. Blundell (2014). "mCSM: predicting the effects of mutations in proteins using graph-based signatures." Bioinformatics **30**(3): 335-342.

Ramachandran, S., P. Kota, F. Ding and N. V. Dokholyan (2011). "Automated minimization of steric clashes in protein structures." Proteins: Structure, Function, and Bioinformatics **79**(1): 261-270.

Rodrigues, C. H., D. E. Pires and D. B. Ascher (2018). "DynaMut: predicting the impact of mutations on protein conformation, flexibility and stability." Nucleic Acids Res **46**(W1): W350-w355.

Sievers, F. and D. G. Higgins (2014). "Clustal Omega, accurate alignment of very large numbers of sequences." Multiple sequence alignment methods: 105-116.

Johnson, J.R., Johnston, B., Clabots, C., Kuskowski, M.A. and Castanheira, M., 2010. Escherichia coli sequence type ST131 as the major cause of serious multidrug-resistant E. coli infections in the United States. Clinical infectious diseases, 51(3), pp.286-294.

Mobegi, F.M., Cremers, A.J., De Jonge, M.I., Bentley, S.D., Van Hijum, S.A. and Zomer, A., 2017. Deciphering the distance to antibiotic resistance for the pneumococcus using genome sequencing data. Scientific reports, 7(1), p.42808.

Other questions

Minor comments:

- Line 45: It would aid clarity if the authors indicated here that they mean sharing of genes.

We thank the reviewer for the suggestion. We have now corrected lines 43-45 and below:

Abstract lines 43-45: "Sharing of genetic elements among different pathogens and commensals inhabiting same hosts and environments has significant implications for antimicrobial resistance (AMR)"

- Supplementary table 1: I noticed that gene *aac(6'')*-*laa* is reported for *S. enterica* under AMR genes. I understand this is probably listed for completeness, but please note that this gene is cryptic, and therefore, in its wildtype state, does not confer resistance in *S. enterica* (see PMIDs: 34135355 and 10542165).

We thank the reviewer for indicating this. The AMR genes in Supplementary Table 1 were acquired by screening the whole genome sequences against the CARD database with a minimum coverage of 70% and identity of 90% to identify known AMR-associated genes in the isolate cohort for both *E. coli* and *S. enterica*. These are indicative results that may change if using a different database.

We added the following disclaimer at the end of the caption for Supplementary Table 1: “The annotation of resistance determinants may change if using a different database. For example, the gene *aac(6'')-laa* is annotated as not conferring resistance in *S. enterica* when using the AMRFinderPlus database (Feldgarden et al. 2021).”

Feldgarden M, et al. AMRFinderPlus and the Reference Gene Catalog facilitate examination of the genomic links among antimicrobial resistance, stress response, and virulence. *Sci Rep* 11, 12728 (2021).

•Line 127-128: Sample counts for *S. enterica* have been erroneously switched for Shandong, and Liaoning.

Corrected on results lines 133-135 and below.

Results Lines 133-135: “Analogously, for *S. enterica* the most represented province was Henan ($n=111$), followed by Liaoning ($n=25$) then Shandong ($n=7$).”

•Line 139-140: Please use the full drug names in text for clarity. I can see that these are defined in the methods section, but it’s clearer if the reader doesn’t have to go back and forth to look these up.

Corrected on results lines 145-150 and below.

Results lines 145-150: “When considering individual antibiotics, the three tetracycline antibiotics (tetracycline, doxycycline and minocycline) were significantly different between species (adjusted p -value <0.001 , chi-squared test with Bonferroni correction), for Doxycycline (DOX) and Minoocycline (MIN) the resistance was more frequently observed in *S. enterica* and for Tetracycline (TET) it was more frequently observed in *E. coli*.”

•Line 139-144: Please consider rephrasing ‘greater resistance’ to ‘resistance was more frequently observed’ if this is what is meant.

Corrected on lines 145-157 and below.

Results lines 145-157: “When considering individual antibiotics, the three tetracycline antibiotics (tetracycline, doxycycline and minocycline) were significantly different between species (adjusted p -value <0.001 , chi-squared test with Bonferroni correction), for Doxycycline (DOX) and Minoocycline (MIN) the resistance was more frequently observed in *S. enterica* and for Tetracycline (TET) it was more frequently observed in *E. coli*. Similarly,

both polymyxins (polymyxin B and colistin) had a resistance frequency significantly higher in *E. coli* (corrected p -value <0.001 , chi-squared test with Bonferroni correction). Two 3rd generation cephalosporins (cefepime and ceftazidime) were proportionally more resistant in *S. enterica* (corrected p -value <0.001 , chi-squared test with Bonferroni correction). Finally, the aminoglycoside streptomycin was found to have a similar proportion of resistant cases, but more intermediate and less susceptible isolates in *S. enterica* compared to *E. coli* (corrected p -value <0.001 , chi-squared test with Bonferroni correction)."

•Figure 1: Panels A and B are quite complex, and the amount of data subsequently makes them difficult to read. To improve readability, authors might consider summarising the data to show the number drug classes the samples are resistant to, rather than the individual drugs, or alternatively numbering the rings and showing these in the legend. If the former option is selected, the authors could provide a rectangular tree showing the full dataset in supplementary data. The legend should also contain any abbreviations shown in the legend (e.g. farm names/identifiers). There is also some misalignment between the phylogroup classifications and the phylogenetic tree structure for *E. coli* which is not explained.

We thank the reviewer for this point and agree the figure was difficult to read. As suggested, we have improved the readability by summarising the data to show the number of classes each isolate is resistant to and removing the phenotypic resistance rings. As also suggested, the phenotypic resistance has been added as an additional supplementary figure, Figure S2. In addition to the suggested changes to Figure 1 we have also changed the colours of the phylogroups/serotypes to pastels to improve the visual appearance and readability. The full farm names have been added to the Figure legend. Regarding the misalignment of phylogroups with the tree, the phylogenetic tree was constructed using the alignment of 3118 genes in the core genome, with only branches with $> 95\%$ likelihood kept (ultrafast bootstrap probability). The phylogroup prediction was done using conventional Clermont typing (Beghain, Bridier-Nahmias et al. 2018) which uses an *in silico* prediction based on primer sequences to 15 genes. As a result of the application of these two different methodologies, occasionally there are isolates that are predicted to be in one phylogroup by Clermont typing but are overall more similar to isolates within another phylogroup, so they will be placed within a different phylogroup in the tree.

Updates to the submission

Supplementary Material lines 12-18: "Phylogroup prediction was done using *in silico* Clermont typing (Beghain, Bridier-Nahmias et al. 2018) based on primer sequences to 15 genes. In contrast, the phylogenetic tree was constructed using the alignment of 3118 genes in the core genome, with only branches with $> 95\%$ likelihood kept (ultrafast bootstrap probability). As a result of the application of these two different methodologies, occasionally there were isolates that are predicted to be in one phylogroup by Clermont typing but were overall more similar to isolates within another phylogroup, so they would be placed within a different phylogroup in the tree."

Figure 1 Legend: “Figure 1. Phylogeny of *E. coli* and *S. enterica* isolates collected from 10 commercial broiler farms in China (A) Maximum likelihood phylogenetic tree of the whole cohort of *E. coli* isolates based on core genome of the 518 isolates, cultured from the animal and environmental samples collected from the 10 farms and 4 abattoirs. Phylogroups are shown as coloured sections. Sample source, region and number of resistance classes are shown as rings around the tree. (B) Maximum likelihood phylogenetic tree of the whole cohort of *S. enterica* isolates based on core genome of the 143 isolates, cultured from the animal and environmental samples collected from the 10 farms and 4 abattoirs. Serotypes are shown as coloured sections. Sample source, region and number of resistance classes are shown as rings around the tree. Farm names are abbreviated: Farm names are abbreviated: Henan 1 (HN1), Henan 2 (HN2) Henan 3 (HN3), Liaoning 1 (LN1), Liaoning 2 (LN2), Liaoning 3 (LN3), Shandong 1 (SD1), Shandong 2 (SD2), Shandong 3 (SD3), Shandong 4 (SD4).”

a *Escherichia coli*

E. coli Phylogroup

b *Salmonella enterica*

Serotype

Sample Source

Farm

Phenotypic resistance

Fig. S2: Phenotypic Resistance patterns of *E. coli* and *S. enterica* isolates collected from 10 commercial broiler farms in China (A) Maximum likelihood phylogenetic tree of the whole

cohort of *E. coli* isolates based on core genome of the 518 isolates, cultured from the animal and environmental samples collected from the 10 farms and 4 abattoirs. Phylogroups are shown as coloured sections. Sample source, region and phenotypic resistances are shown as strips beside the tree. (B) Maximum likelihood phylogenetic tree of the whole cohort of *S. enterica* isolates based on core genome of the 143 isolates, cultured from the animal and environmental samples collected from the 10 farms and 4 abattoirs. Serotypes are shown as coloured sections. Sample source, region and phenotypic resistances are shown as strips beside the tree. In both panels A and B the phenotypic resistance rings are grouped by antibiotic class, with presence (coloured) defined as resistance to at least one antibiotic in the class and absence (grey) as susceptibility to all tested antibiotics in the class. Farm names are abbreviated: Henan 1 (HN1), Henan 2 (HN2), Henan 3 (HN3), Liaoning 1 (LN1), Liaoning 2 (LN2), Liaoning 3 (LN3), Shandong 1 (SD1), Shandong 2 (SD2), Shandong 3 (SD3), Shandong 4 (SD4).

Beghain, J., A. Bridier-Nahmias, H. Le Nagard, E. Denamur and O. Clermont (2018). "ClermonTyping: an easy-to-use and accurate in silico method for Escherichia genus strain phylotyping." *Microbial genomics* 4(7): e000192.

•Line 186-7: The higher level of clonality reported may relate to the selection of the same SNP threshold used for both genera at line 185 (i.e <15 SNPs). Including substitution rate estimates from phylodynamic analyses would assist in determining this, and if the 15 SNP threshold is appropriate/relevant for understanding the genetic epidemiology for both *Salmonella* spp. and *E. coli*, or if different thresholds should be used for each. The authors could also consider commenting on putative transmission patterns and events based on these data.

We thank the reviewer for this comment. The substitution rate estimates from the Bayesian analysis have now been added as Supplementary Table 10. The substitution rates for *E. coli* and *S. enterica* in our cohort predict *E. coli* mutating slightly faster than *S. enterica* but the difference is less than an order of magnitude, supporting the premise of using the same threshold for both species. However, we agree with the reviewer that the effect of changing this threshold should be explored to ensure it is appropriate. To that end, we have now looked at whether changing the threshold, significantly changes the clusters observed. Considering a range of thresholds between 0 and 50 SNPs, we found that the clustering of neither species changed significantly (see Figures 8 and 9 below).

Figure 8. SNP network analysis of *S. enterica* isolates with different SNPs thresholds. Network diagram showing pairwise connections between chicken (circle) and environmental (star) isolates with different SNPs thresholds (5, 10, 15, 20, 25, 30, 35, 40 and 50) for *S. enterica*. Each panels show a different threshold with its value indicated in the title of the panel. The nodes are colour-coded according to serotype.

Figure 9. SNP network analysis of *E. coli* isolates with different SNPs thresholds. Network diagram showing pairwise connections between chicken (circle) and environmental (star) isolates with different SNPs thresholds (5, 10, 15, 20, 25, 30, 35, 40 and 50) for *E. coli*. Each panels show a different threshold with its value indicated in the title of the panel. The nodes are colour-coded according to phylogroup.

Given this, we believe the threshold of <15 SNPs for both species to be acceptable. Regarding putative transmission patterns and events, we thank the reviewer for this suggestion. However, given that the timepoints were 3 weeks apart and that the mutation rates were relatively high, we feel that commenting on transmission events would be speculative and beyond the scope of this study.

•Figure 2: Using a gradient colour scale to show increasing/decreasing genetic distance may aid clarity here, as this is currently difficult to understand with the current colour palette used.

Corrected. This figure has now been moved to the Supplementary Material Fig. S14.

Fig. S14. SNP network analysis of highly connected isolates in *S. enterica* and *E. coli*. Network diagram showing pairwise connections between chicken (circle) and environmental (star) isolates with less than 15 pairwise SNP differences for (a) *S. enterica* and (b) *E. coli*. The panels in each row show the same network with the nodes colour-coded according to time point (left), source type (centre) or serotype/phylogroup (right). The lines between pairs of isolates are colour-coded by SNP number.

•Figure S2: Please state the statistical test used in the legend.

Figure S2 and S3 (now S15 and S16) used an ANOVA test with post hoc Tukey comparisons. This information has been added to the legend and also to the Material and Methods lines 1173-1175 and below.

Figure S15 Legend: “**Fig. S15.** Distribution of pairwise comparisons of SNP distance between (A) *E. coli* and (B) *S. enterica* isolates across source types, different farms, and provinces. Only pairs with less than 100 SNPs were included in the analysis (19). Differences in the means were tested using a pairwise Tukey HSD test with FWER control (<0.05), with significance measured as adjusted p-values < 0.05. Adjusted p-values of significant tests are shown on the plot.”

Figure S16 Legend: “**Fig. S16.** Distribution of pairwise comparisons of SNP distance between (A) *E. coli* and (B) *S. enterica* isolates across collection dates. Only pairs with less than 100 SNPs were included in the analysis (19). The distribution categories refer to the difference in dates between each isolate pair. Differences in the means were tested using a pairwise Tukey HSD test with FWER control (<0.05), with significance measured as adjusted p -values < 0.05 . Adjusted p -values of significant tests are shown on the plot.”

Material and Methods Lines 1173-1175: “ANOVA tests with post hoc Tukey comparisons to consider intra- and inter-cluster variability across geography and time for the network analysis.”

- Table S2: At present it is unclear how to use the ‘sample pairs’ column to link the data. It may be clearer to use the accession number or strain ID of the samples here.

Corrected.

- Line 232: At present it is unclear if the counts referenced refer to the total number of plasmids or the total number of unique plasmids. Please clarify the meaning here.

Corrected results lines 255-256 and below.

Results lines 255-256: “The total number of plasmids reconstructed was 3169 in *E. coli* (mean of 6.1 per isolate) and 572 in *S. enterica* (mean of 4 per isolate).”

- Line 257: The meaning of “regions” is a little unclear, please clarify the meaning to refer to clades or clusters (if this is what is intended). Please also note that the text states the MRCA of the bottom box on Figure 3 should be 2019, but the current MRCA is ~2017. Either the text or the box needs to be modified.

We thank the reviewer for this question. The “regions” in the sentence were related to the red rectangles on Fig. 3, The first region date it back to 2019 while the second region date it back to 2017. We have now updated the text to clarify it.

Results lines 280-283: “However, there were two regions (red rectangles on **Fig. 2**) of more recent evolution, with the MRCA dating back to 2019 in the upper region and to 2017 in the bottom region, and within these we have pairs of *E. coli* and *S. enterica* strains isolated from the same sample, potentially indicative of recent transmission of this plasmid between species.”

- Figure 4: The yellow shade used in the colour palette here, coupled with the thin lines for the genes makes the figure somewhat difficult to read. Please consider replacing yellow with a different colour.

Corrected.

•Line 303: 26 antimicrobials are referred to here, but 28 are listed at line 129. I think this is because there was one drug not tested for each organism, and 26 indicates the overlap of drugs tested on both organisms, but it would be helpful to clarify this for the reader.

Corrected on results lines 354-361 and below.

Results lines 354-361: **“Machine learning unravels known and novel AMR-associated core genome SNPs and accessory genes correlated with resistance/susceptibility profiles to multiple antimicrobials in both species**

Given the similar proportions of resistance, against 28 antimicrobials across the 518 *E. coli* isolates and against 26 antimicrobials across 143 *S. enterica* isolates, despite phylogenetic differences, we further investigated which genetic determinants (features) were underlying the experimentally determined resistance/susceptibility profiles and if they were in common between the two species.”

•Figures 5-6: While I understand that Roary assigns “group” IDs to hypothetical proteins, the authors should attempt to identify the putative function of these genes via BLAST searches against GenBank and UniProt/SwissProt databases and use the appropriate gene names for the figures. Looking at Table S5, this may have been carried out, but the group names are still used in the figures and it's a lot of work for the reader to link up these data.

We thank the reviewer for this suggestion. We have now identified putative gene identities for these group IDs where possible using a BLAST search against the suggested databases, with an 80% identity threshold. We have added the names of these genes to all relevant figures and Supplementary Table 5. In addition, we have described this in the Material and Methods on Lines 978-988 and below:

Material and Methods Lines 978-988: “For each species, all annotated genomes were taken as input for pan-genome analysis with core gene alignments through Roary v3.13. For hypothetical proteins, assigned with group IDs by Roary, we identified putative gene names via a BLAST search of the GenBank and SwissProt databases at 80% identity.”

•Table S7: This is not currently mentioned in the body text of the manuscript.

Corrected on results lines 548-554 and below.

Results Lines 548-554: “To further investigate the systemic relationships connecting the identified AMR genetic signatures on a mechanistic level and to elucidate their mechanistic effects beyond genes encoding proteins targeted by drugs (i.e., positive selection in basal biosynthetic, regulation, and repair pathways), we integrated the top 10% ranked genetic determinants identified by ML with the genome-scale metabolic (GSM) models of *E. coli* (K-12 MG1655, iML1515 (Monk et al. 2017)) and *S. enterica* (STM v1.0 (Thiele et al. 2011), (Thiele, Hyduke et al. 2011) (Thiele, Hyduke et al. 2011), see **Table S8.**”

Monk JM, *et al.* iML1515, a knowledgebase that computes *Escherichia coli* traits. *Nature Biotechnology* **35**, 904-908 (2017).

Thiele I, *et al.* A community effort towards a knowledge-base and mathematical model of the human pathogen *Salmonella Typhimurium* LT2. *BMC Syst Biol* **5**, 8 (2011).

•Line 590: While the authors somewhat correctly state that direct evidence of plasmid transmission between *Salmonella* and *E. coli* are scant, there are notable cases where this is likely, for example the IncY plasmid responsible for the emergence of XDR *Salmonella* Typhi appears to originate from *E. coli* (see: 29463654).

We thank the reviewer for this comment and have added this to the Discussion lines 713-718:

Discussion Lines 713-718: "This plasmid has been previously associated with cross species horizontal transfer(Fernández, Cloeckeaert *et al.* 2007), however, to our knowledge direct transfer of this plasmid (IncHI2) between strains isolated from these two species has not previously been observed, although it has been observed in a small number of other plasmid types e.g. IncY (Klemm, Shakoor *et al.* 2018)"

Fernández, A. G., A. Cloeckeaert, A. Bertini, K. Praud, B. Doublet, F.-X. Weill and A. Carattoli (2007). "Comparative Analysis of IncHI2 Plasmids Carrying *bla*_{CTX-M-2} or *bla*_{CTX-M-9} from *Escherichia coli* and *Salmonella enterica* Strains Isolated from Poultry and Humans." *Antimicrobial Agents and Chemotherapy* **51**(11): 4177-4180.

Klemm, E. J., S. Shakoor, A. J. Page, F. N. Qamar, K. Judge, D. K. Saeed, V. K. Wong, T. J. Dallman, S. Nair, S. Baker, G. Shaheen, S. Qureshi, M. T. Yousafzai, M. K. Saleem, Z. Hasan, G. Dougan and R. Hasan (2018). "Emergence of an Extensively Drug-Resistant *Salmonella enterica* Serovar Typhi Clone Harboring a Promiscuous Plasmid Encoding Resistance to Fluoroquinolones and Third-Generation Cephalosporins." *mBio* **9**(1).

•Line 605: 26 drugs are listed here, but the text at line 129 states 28.

Corrected on discussion lines 737-739 and below.

Discussion lines 737-739: "Antibiotic susceptibility testing done against a panel of up to 28 antimicrobials showed similar proportions of resistant isolates in both species for many antibiotics, in line with previous studies"

•Lines 703-705 & 718-719: Please detail primer design steps or cite the sources of the primers as appropriate.

Apologies for this omission. These references have now been added to the Material and Methods Lines 856-860 and 872-875 and below.

Material and Methods Lines 856-860: "The positive isolates identified were further confirmed by PCR using *E. coli*-specific primers ITS-F (5'-CAATTTTCGTGTCCCCTTCG-3') and ITS-R (5'-GTTAATGATAGTGTGTCGAAAC-3')(Maheux, Picard et al. 2009)."

Material and Methods Lines 872-875: "The positive isolates identified were further confirmed by PCR using Salmonella specific primers invA-F (5'-GTGAAATTATCGCCACGTTCCGGCAA-3') and invA-R (5'-TCATCGCACCGTCAAAGGAACC-3')(Sharma and Das 2016)."

Sharma I. and Das K. (2016). "Detection of invA Gene in Isolated Salmonella from Marketed Poultry Meat by PCR Assay." *Journal of Food Processing & Technology* 7(3): 564.

Maheux, A. F., F. J. Picard, M. Boissinot, L. Bissonnette, S. Paradis and M. G. Bergeron (2009). "Analytical comparison of nine PCR primer sets designed to detect the presence of *Escherichia coli*/Shigella in water samples." *Water Research* 43(12): 3019-3028.

•Lines-734-741: 27 drugs are listed but line 129 says 28 were tested in total. It appears Cefotaxime/Sulbactam may have been missed here.

Corrected on Material and Methods lines 888-899 and below.

Material and Methods lines 888-899: "The resistance/susceptibility of 28 and 26 antimicrobial compounds were measured for each of the *E. coli* and *S. enterica* isolates, respectively, and these included ampicillin (AMP), ampicillin/sulbactam (AMS), tetracycline (TET), chloramphenicol (CHL), trimethoprim/sulfamethoxazole (SXT), cefazolin (CFZ), cefotaxime (CTX), cefotaxime/clavulanic acid (CTX-C – *E. coli* only), ceftazidime (CAZ), ceftazidime/clavulanic acid (CAZ-C – *E. coli* only), ceftazidime (CFX), gentamicin (GEN), imipenem (IMI), nalidixic acid (NAL – *E. coli* only), azithromycin (AZI – *S. enterica* only), sulfisoxazole (SUL), ciprofloxacin (CIP), amoxicillin/clavulanic acid (AMC), polymyxin E (CT), polymyxin B (PB), minocycline (MIN), amikacin (AMI), aztreonam (AZM), cefepime (FEP), meropenem (MEM), levofloxacin (LEV), doxycycline (DOX), kanamycin (KAN) and streptomycin (STR). *E. coli* ATCC 25922 was used as a control for the antimicrobial susceptibility testing"

•Line 757: Please provide a citation for the readfq software.

Corrected on Material and Methods line 914 and below.

Material and Methods line 914: "All sequences were pre-processed through readfq v10 (Li, 2019)"

Heng Li. *readfq*. unpublished software. 2019; Available from: <https://github.com/lh3/readfq>.

- Line 780-781: Please provide a citation with access dates for the database.

The citation and access date have been added to the Material and Methods lines 960-963 and below.

Material and Methods Lines 960-963: "To compare plasmid presence with the level observed previously we downloaded from BV-BRC(Olson, Assaf et al. 2023) (accessed 9th March 2023), all the good quality *E. coli* (n= 705) and *S. enterica* (n=265) isolates collected from chicken in China (Table S9) and conducted plasmid screening on these using the PlasmidFinder(Carattoli, Zankari et al. 2014) database in Abricate(Seemann 2018)."

Carattoli, A. *et al.* (2014). "In silico detection and typing of plasmids using PlasmidFinder and plasmid multilocus sequence typing." *Antimicrobial agents and chemotherapy* 58(7): 3895-3903.

Olson, R. D. *et al.* (2023). "Introducing the Bacterial and Viral Bioinformatics Resource Center (BV-BRC): a resource combining PATRIC, IRD and ViPR." *Nucleic Acids Res* 51(D1): D678-d689.

Seemann, T. (2018). "ABRicate: mass screening of contigs for antimicrobial and virulence genes." Department of Microbiology and Immunology, The University of Melbourne, Melbourne, Australia. Available online: <https://github.com/tseemann/abricate> (accessed on 28 February 2019).

- Line 802-805: From the current text, is unclear which Roary files were used as input for Piggy. The current text reads as though Piggy was applied to an alignment of core genes, which would omit intergenic regions by nature. Moreover, the following lines suggest that only an alignment of intergenic steps was used for the network alignments. Please clarify the steps used in this analysis to aid clarity and reproducibility.

Apologies that this was not clear, the input for Piggy is a folder of all output files from Roary, in addition the Roary input files (GFF files for each isolate) are also required. The text has been edited to clarify. In addition, the network analysis title and opening sentence have been edited to clarify that core genome SNPs were used. This has been changed in the Material and Methods on lines 992-1001 and below.

Material and Methods Lines 992-1001: "To map intergenic SNPs, for each species an alignment of core intergenic regions was created using Piggy v1.5(Thorpe, Bayliss et al. 2018). As input all the output files from Roary (Page, Cummins et al. 2015) and gff files for each isolate were used. An alignment of the intergenic clusters was generated by Piggy v1.5(Thorpe, Bayliss et al. 2018) and SNPs were called from this alignment using SNP-sites.

Network analysis based on core genome SNPs

Networks of *E. coli* and *S. enterica* isolates collected from different samples sources in the farms and different regions of China were created using a pairwise hamming distance comparison based on core genome SNPs."

Page, A. J., C. A. Cummins, M. Hunt, V. K. Wong, S. Reuter, M. T. Holden, M. Fookes, D. Falush, J. A. Keane and J. Parkhill (2015). "Roary: rapid large-scale prokaryote pan genome analysis." *Bioinformatics* 31(22): 3691-3693.

Thorpe, H. A., S. C. Bayliss, S. K. Sheppard and E. J. Feil (2018). "Piggy: a rapid, large-scale pan-genome analysis tool for intergenic regions in bacteria." *GigaScience* 7(4).

•Line 816: Please provide the python package used for plotting and any relevant citations.

Corrected on Material and Methods lines 1005-1010 and below.

Material and Methods lines 1005-1010: "To consider intra- and intercluster variability across geography and time a threshold of <100 SNPs was applied as done previously(Shaw, Chau et al. 2021) and this subset of isolate pairs was plotted using Python (Matplotlib v3.6.2 (Hunter, 2007))"

J. D. Hunter, "Matplotlib: A 2D Graphics Environment", *Computing in Science & Engineering*, vol. 9, no. 3, pp. 90-95, 2007

•Line 820: Please provide details on alignment lengths, and the phylogenetic tree rooting method used.

We apologise for this omission. The alignment length of the *E. coli* maximum likelihood tree was 2,214,946 nucleotide sites of which 198,217 were informative. For the *S. enterica* maximum likelihood tree the alignment length was 3,489,080 nucleotide sites with 87,084 informative sites. Both of these trees were unrooted. For the Bayesian trees, alignment lengths ranged between 510,551 and 4,261,565 bases and rooting was done in BEAST based on the clock model predictions. This information has now been added to Material and Methods lines 1014-1038 and below.

Material and Methods Lines 1014-1038: "*Whole genome phylogenetic analysis and Bayesian evolutionary analysis*

IQTree v2.1.4-beta (Minh, Schmidt et al. 2020) was used to construct the maximum-likelihood phylogenetic trees from the core genome alignments of *E. coli* and *S. enterica*. The alignment length of the *E. coli* core genome was 2,214,946 nucleotide sides of which 198,217 were informative. For the *S. enterica* core genome the alignment length was 3,489,080 nucleotide

sites with 87,084 informative sites. For both *E. coli* and *S. enterica* different nucleotide replacement models were tested automatically with the GTR (+F+R3) replacement model selected as the best model. The Ultrafast bootstrap algorithm was used with 10000 replicates to assess branch support. The generated trees were unrooted. The phylogenetic trees were subsequently visualised through iTOLv5(Letunic and Bork 2021). Subsets of sequences from this study from individual phylogroups (*E. coli*) or serotypes (*S. enterica*), were selected for Bayesian evolutionary analysis using BEAST v 1.10.4(Suchard, Lemey et al. 2018). Analysis was conducted on a core genome alignment of each lineage by using Roary v3.13(Page, Cummins et al. 2015). The alignment lengths varied between 510,551 and 4,261,565 and can be found in Table S9. For each species, all combinations of three clock models (strict, uncorrelated log normal, and uncorrelated exponential) and four tree priors (constant coalescent, logistic growth, Bayesian skyline, and birth-death model) were tested using steppingstone sampling on a subset of the isolates to identify the best model. The GTR-gamma nucleotide substitution model was used, as selected for the maximum likelihood tree. The analysis was run for 3 independent chains until the effective sample size (ESS), that is, the effective number of independent draws from the posterior distribution, for all parameters was greater than 200 per chain. Convergence was assessed in Tracer v1.7.1(Rambaut, Drummond et al. 2018), and chains were subsequently combined using LogCombiner v1.10.4(Drummond and Rambaut 2007). The maximum clade credibility tree was selected using TreeAnnotator v1.10.4(Drummond and Rambaut 2007) and then visualized in iTOL v5(Letunic and Bork 2021)."

Drummond, A. J. and A. Rambaut (2007). "BEAST: Bayesian evolutionary analysis by sampling trees." *BMC Evol Biol* 7: 214.

Letunic, I. and P. Bork (2021). "Interactive Tree Of Life (iTOL) v5: an online tool for phylogenetic tree display and annotation." *Nucleic Acids Res.*

Minh, B. Q., H. A. Schmidt, O. Chernomor, D. Schrempf, M. D. Woodhams, A. Von Haeseler and R. Lanfear (2020). "IQ-TREE 2: New models and efficient methods for phylogenetic inference in the genomic era." *Molecular Biology and Evolution* 37(5): 1530-1534.

Page, A. J., C. A. Cummins, M. Hunt, V. K. Wong, S. Reuter, M. T. Holden, M. Fookes, D. Falush, J. A. Keane and J. Parkhill (2015). "Roary: rapid large-scale prokaryote pan genome analysis." *Bioinformatics* 31(22): 3691-3693.

Rambaut, A., A. J. Drummond, D. Xie, G. Baele and M. A. Suchard (2018). "Posterior Summarization in Bayesian Phylogenetics Using Tracer 1.7." *Syst Biol* 67(5): 901-904.

Suchard, M. A., P. Lemey, G. Baele, D. L. Ayres, A. J. Drummond and A. Rambaut (2018). "Bayesian phylogenetic and phylodynamic data integration using BEAST 1.10." *Virus Evol* 4(1): vey016.

- The manuscript lacks substantial discussion of the limitations of the analyses carried out and how these impact the findings. This needs addressing.

We have now added a section to our discussion lines 783-798, as well as supplementary information discussing the limitations of the study. The new discussion section incorporates the answer to this Reviewer #4 - Question 3 commenting on shortcomings of the current validation, that also gave origin to Supplementary Note 5; as well as the answer to Reviewer #2 – Question #1, that gave origin to a section in Supplementary Note 6 commenting on confounding factors.

Discussion lines 783-798: “A limitation of this study is that the analysis of association of AMR resistance to genetic features was primarily *in silico*, with experimental gene validation restricted to a small number of gene and a knockout approach (**Supplementary Note 5**). To fully validate the predictions made in this study experimental validation of the specific genetic mutations would be required. It is important to note that the *in silico* mutations made for metabolic genes should be tested in an in vitro model (e.g. natural host derived and human macrophages and/or epithelial cells) in order to validate the physiological relevance of the metabolic pathway findings. In relation to the analyses done to understand the influence of co-inhabitation as well as country of collection in the observed results (genetic elements tied to AMR), another limitation of our study is intrinsic to the challenge of accounting for the many further confounding factors which may have influenced our results. Further insight is given in **Supplementary Note 6**. One further limitation of the results is the close genomic homology of *E. coli* and *S. enterica*, which may have contributed to some of the overlapping AMR-associated genes selected by the machine learning and metabolic modelling. Future studies should consider comparisons between co-habiting gut bacteria species less closely related, for example *Enterococcus spp.* or *Campylobacter spp.*”

Reviewer #4

Here Baker et al have characterised the population structure, evolution and AMR phenotypes of *E. coli* and *S. enterica*, highlighting differences across environments and hosts. The authors showed that there is significant potential for transfer of plasmids and mobile ARGs between *E. coli* and *S. enterica* in the environments studied, suggesting potential for the spread of AMR generally. Machine learning approaches revealed both *E. coli* and *S. enterica* had a common subset of features highly associated with AMR and that they were linked to the same functional pathways essential for growth. These results indicated the potential for common metabolic adaptations of the bacteria within the sampling environments.

This work is highly valuable for understanding the evolution and transmission of AMR, especially given the continuous global rise in AMR. It provides a framework for future studies and a platform for phenotypic studies.

The authors' machine learning approaches used are robust and innovative. They have expanded on previous methods to improve capture the co-occurrence of multiple mechanisms including the additive effect on resistance. They also employed robust validation models for this data.

We are very grateful to the reviewer for their useful comments. The suggestion for additional analysis, figure edits and consideration of the implications of the research beyond China were all very helpful in improving the manuscript.

Reviewer #4 – Question #1

Comments to address

Geography is a major determinant of circulating sequence types, and these can differ quite a lot across continents. Can you address how a study based in China might be more broadly applicable globally, i.e. what are the implications outside of China?

Antimicrobial resistance poses a major threat to human/animal health around the world, and we agree with the Reviewer that geography can play an important role on circulating sequence types. In particular for livestock farming, rearing practices, including antibiotic treatments, and the microbial ecologies at play in the anthropogenic environments typical of each different country, surely have an effect on circulating species, genomes and relationships with AMR. As our methods are predominantly data-driven, fundamentally relying on various forms of statistical analyses and in particular on fitting regression models to available observations, the obtained results will reflect the datasets we started with.

It is interesting to evaluate how results may be influenced by country of collection in a more quantitative manner, for example by assessing how many genetic elements predicted as relevant to AMR in one country, may be also found when working with data from a different country, as well as how many are instead unique to each specific country. To address this

issue, and also in response to another Reviewer who had similarly asked for comments on the influence of country of collection (Reviewer #1, Question #1 and Question #2), in the updated submission we reported the results of comparisons performed between the machine learning predictions (genetic elements more strongly associated to AMR) obtained using Chinese isolates (*E. coli* and *S. enterica*) and new sets of European isolates, retrieved from the literature; more specifically from the EFFORT project (Leekitcharoenphon, Johansson et al. 2021)) (isolates of *E. coli*, sampled from five EU countries: Denmark, Germany, Poland, Switzerland and Spain), and from the ENGAGE project (Alba, Leekitcharoenphon et al. 2020) *S. enterica*, sampled from Italy.

One of the main challenges to tackle in any study aimed at isolating the effects of a specific factor (in this case, country of collection) is to make sure that the effects of additional, unaccounted for (confounding) factors is kept to a minimum. For example, in this work a big, further influence factor was the presence/absence of co-inhabiting *E. coli* and *S. enterica* in the same sample, being the influence of co-inhabitation on AMR prediction one of the core scientific questions addressed in this research. As we already had both co-inhabiting and not co-inhabiting isolates in our Chinese cohort, we selected European sets with various situations in terms of co-inhabitation: the EFFORT set was comprised of *E. coli* isolates with more likely absence of *S. enterica*, as the birds had been vaccinated against Salmonella (the complete absence of co-inhabitation is challenging to demonstrate, as commented in Supplementary Note 3 “A comment on the challenge of obtaining controls for studying the influence of co-inhabitation”); the ENGAGE set was comprised of *S. enterica* isolates with no checks for presence of *E. coli*, so co-inhabitation is unknown (more details on the European datasets are provided in the updated supplementary material).

The comparison of the machine learning prediction (also detailed in the response to Reviewer #1 – Question #2) indicates that the largest difference of predictions was observed for *S. enterica*: European ENGAGE (not necessarily co-inhabiting) vs the Chinese (co-inhabiting) set, with only 3.01% overlap. The second largest difference was observed for *E. coli*: European EFFORT (not co-inhabiting) vs the Chinese (co-inhabiting) set: with 14.19% overlap. The smallest difference was observed for the combined *E. coli* and *S. enterica* Chinese sets: co-inhabiting vs not co-inhabiting (39%). These results seem to indicate that the country of collection is a stronger factor than co-inhabitation, but the latter is still appreciable, in particular when the samples are collected within the same country.

It is also interesting to speculate onto whether country-related influences in this type of analysis will increase or decrease with time. AMR can have multiple routes of transmission with spread from animal to human by direct contact or via food; and global spread by export/import of live animals and products is the most significant one in the context of food production (Teuber 1999, Jung et al. 2022). Poultry meat production accounts for approximately one third of the overall meat production worldwide and has increased rapidly over the last decades (FAO 2020, FAO 2023). The poultry industry will drive meat production growth in the coming decade, and it is projected to account for half of all additional meat produced in that time period (OECD/FAO 2020). The USA, Brazil, the European Union, and

China continue to be the leading producers of poultry meat globally. Out of the total meat trade globally (37.6 million tonnes), 14.1 million tonnes (37.5%) relate to poultry meat exported by leading producing countries, making chicken the most exported type of meat (OECD/FAO 2020). Specifically in China, industrial livestock production has increased in recent years (Cheng et al. 2018), with poultry the meat in highest demand. In 2021, China exported \$668M in Poultry Meat, making it the 11th largest exporter of Poultry Meat in the world. At the same year, Poultry Meat was the 489th most exported product in China. The main destination of Poultry Meat exports from China were: Hong Kong (\$500M), Macau (\$51.1M), Malaysia (\$40.3M), Mongolia (\$24.7M), and Cambodia (\$13.2M), but more destinations may be added in the future, increasing the risk of AMR spread.

Finally, as the importance of adopting data-driven approaches gets gradually more recognised as an important means to gain further insights on the relationships between genetic elements and the antimicrobial resistance phenotype, it becomes increasingly more evident that appropriate data, needed to perform similar studies, is often difficult to come by. Although the presence/absence of bacterial pathogens is often documented, as well as their prevalence, comprehensive collections of large amounts of observations are seldom available, in particular if we require such data to carry isolate information along with AST and sequencing data. Even more challenging is to find datasets also reporting co-inhabitation of multiple bacteria, ours being the only one so far, at least to our knowledge.

References

(Chaisatit et al 2011, Patchanee et al 2017, Shi 2021, Barua et al 2014, Le Hello 2011, Sun et al, 2022, Shi 2023, Ćwiek 2021, Tavares 2022, Rasmussen, 2015, Fuentes-Castillo 2020)”

Abreu, Raquel, Teresa Semedo-Lemsaddek, Eva Cunha, Luís Tavares, and Manuela Oliveira. "Antimicrobial Drug Resistance in Poultry Production: Current Status and Innovative Strategies for Bacterial Control." *Microorganisms* 11, no. 4 (2023): 953.

Adeyanju, Gladys Taiwo, and Olayinka Ishola. "Salmonella and Escherichia coli contamination of poultry meat from a processing plant and retail markets in Ibadan, Oyo State, Nigeria." *Springerplus* 3 (2014): 1-9.

Leekitcharoenphon P, et al. Genomic evolution of antimicrobial resistance in Escherichia coli. *Scientific Reports* 11, 15108 (2021).

European Food Safety Authority, European Centre for Disease Prevention Control. The European Union summary report on trends and sources of zoonoses, zoonotic agents and food-borne outbreaks in 2017. *EFSA Journal* 16, e05500 (2018).

Alba P, et al. Molecular epidemiology of Salmonella Infantis in Europe: insights into the success of the bacterial host and its parasitic pESI-like megaplasmid. *Microb Genom* 6, (2020).

Allel, Kasim, Lucy Day, Alisa Hamilton, Leesa Lin, Luis Furuya-Kanamori, Catrin E. Moore, Thomas Van Boeckel, Ramanan Laxminarayan, and Laith Yakob. "Global antimicrobial-

resistance drivers: an ecological country-level study at the human–animal interface." *The Lancet Planetary Health* 7, no. 4 (2023): e291-e303.

Bacanlı M, Başaran N. Importance of antibiotic residues in animal food. *Food Chem Toxicol.* 2019;125:462–466

Barua, H., Biswas, P.K., Talukder, K.A., Olsen, K.E. and Christensen, J.P., 2014. Poultry as a possible source of non-typhoidal *Salmonella enterica* serovars in humans in Bangladesh. *Veterinary microbiology*, 168(2-4), pp.372-380.

Chaisatit, C., Tribuddharat, C., Pulsrikarn, C. and Dejsirilert, S., 2011. Prevalence and Molecular Characterization of Antibiotic Resistant Bacteria Isolated from Chicken Meats Sold in Supermarkets in Bangkok. *Journal of Infectious Diseases and Antimicrobial Agents*, 28(3), p.252.

Cheng, J., Hu, H., Kang, Y. *et al.* Identification of pathogens in culture-negative infective endocarditis cases by metagenomic analysis. *Ann Clin Microbiol Antimicrob* 17, 43 (2018)

Courrol, L.C., Vallim, M.A. Spectroscopic Analysis of Chicken Meat Contaminated with *E. coli*, *Salmonella*, and *Campylobacter*. *Food Anal. Methods* 14, 512–524 (2021)

Ćwiek, K., Woźniak-Biel, A., Karwańska, M., Siedlecka, M., Lammens, C., Rebelo, A.R., Hendriksen, R.S., Kuczkowski, M., Chmielewska-Władyka, M. and Wieliczko, A., 2021. Phenotypic and genotypic characterization of *mcr-1*-positive multidrug-resistant *Escherichia coli* ST93, ST117, ST156, ST10, and ST744 isolated from poultry in Poland. *Brazilian Journal of Microbiology*, 52(3), pp.1597-1609.

FAO (2020) Meat market review, overview of global meat market developments in 2019

FAO (2023). Gateway to Poultry Production and Products. Available online: <https://www.fao.org/poultry-production-products/production/en/>

Fuentes-Castillo, D., Esposito, F., Cardoso, B., Dalazen, G., Moura, Q., Fuga, B., Fontana, H., Cerdeira, L., Dropa, M., Rottmann, J. and González-Acuña, D., 2020. Genomic data reveal international lineages of critical priority *Escherichia coli* harbouring wide resistome in Andean condors (*Vultur gryphus* Linnaeus, 1758). *Molecular Ecology*, 29(10), pp.1919-1935.

Jung, Dongyun, Beverly J. Morrison, and Joseph E. Rubin. "A review of antimicrobial resistance in imported foods." *Canadian Journal of Microbiology* 68, no. 1 (2022): 1-15.

Le Hello, S., Hendriksen, R.S., Doublet, B., Fisher, I., Nielsen, E.M., Whichard, J.M., Bouchrif, B., Fashae, K., Granier, S.A., Jourdan-Da Silva, N. and Cloeckaert, A., 2011. International spread of an epidemic population of *Salmonella enterica* serotype Kentucky ST198 resistant to ciprofloxacin. *Journal of Infectious Diseases*, 204(5), pp.675-684.

Menkem ZE, Ngangom BL, Tamunjoh SSA, Boyom FF. Antibiotic residues in food animals: public health concern. *Acta Ecol Sin.* 2019;39:411–415

Muinde, Patrick, John Maina, Kelvin Momanyi, Victor Yamo, John Mwaniki, and John Kiiru. "Antimicrobial Resistant Pathogens Detected in Raw Pork and Poultry Meat in Retailing Outlets in Kenya." *Antibiotics* 12, no. 3 (2023): 613.

Murray, Christopher JL, Kevin Shunji Ikuta, Fablina Sharara, Lucien Swetschinski, Gisela Robles Aguilar, Authia Gray, Chieh Han et al. "Global burden of bacterial antimicrobial resistance in 2019: a systematic analysis." *The Lancet* 399, no. 10325 (2022): 629-655.

Nisha AR. Antibiotic residues - a global health hazard. *Vet World*. 2008;1:375–377

OECD/FAO (2020) Meat. In: OECD-FAO agricultural outlook 2020-2029. OECD Publishing

OEC (2023) Poultry Meat in China. Available on: <https://oec.world/en/profile/bilateral-product/poultry-meat/reporter/chn>

Panisello, Pedro J., Roisin Rooney, Peter C. Quantick, and Rosalind Stanwell-Smith. "Application of foodborne disease outbreak data in the development and maintenance of HACCP systems." *International Journal of Food Microbiology* 59, no. 3 (2000): 221-234.

Patchanee, P., Eiamsam-Ang, T., Vanaseang, J., Boonhot, P. and Tadee, P., 2017. Determination of regional relationships among Salmonella spp. isolated from retail pork circulating in the Chiang Mai municipality area using a WGS data approach. *International journal of food microbiology*, 254, pp.18-24.

Patel, Jay, Anne Harant, Genevieve Fernandes, Ambele Judith Mwamelo, Wolfgang Hein, Denise Dekker, and Devi Sridhar. "Measuring the global response to antimicrobial resistance, 2020–21: a systematic governance analysis of 114 countries." *The Lancet Infectious Diseases* 23, no. 6 (2023): 706-718.

Rasmussen, M.M., Opintan, J.A., Frimodt-Møller, N. and Styris have, B., 2015. Beta-lactamase producing Escherichia coli isolates in imported and locally produced chicken meat from Ghana. *Plos one*, 10(10), p.e0139706.

Sajid A, Kashif N, Kifayat N, Ahmad S. Detection of antibiotic residues in poultry meat. *Pak J Pharm Sci*. 2016;29:1691–1694

Shi, Q., Ye, Y., Lan, P., Han, X., Quan, J., Zhou, M., Yu, Y. and Jiang, Y., 2021. Prevalence and Characteristics of Ceftriaxone-Resistant Salmonella in Children's Hospital in Hangzhou, China. *Frontiers in Microbiology*, 12, p.764787.

Shi, J., Zhu, H., Liu, C., Xie, H., Li, C., Cao, X. and Shen, H., 2023. Epidemiological and genomic characteristics of global mcr-positive Escherichia coli isolates. *Frontiers in Microbiology*, 13, p.1105401.

Sun, R.Y., Guo, W.Y., Zhang, J.X., Wang, M.G., Wang, L.L., Lian, X.L., Ke, B.X., Sun, J., Ke, C.W., Liu, Y.H. and Liao, X.P., 2022. Phylogenomic analysis of Salmonella Indiana ST17, an emerging MDR clonal group in China. *Journal of Antimicrobial Chemotherapy*, 77(11), pp.2937-2945.

Tavares, R.D., Tacão, M., Ramalheira, E., Ferreira, S. and Henriques, I., 2022. Report and Comparative Genomics of an NDM-5-Producing Escherichia coli in a Portuguese Hospital: Complex Class 1 Integrons as Important Players in bla NDM Spread. *Microorganisms*, 10(11), p.2243.

Teuber, M. "Spread of antibiotic resistance with food-borne pathogens." *Cellular and Molecular Life Sciences CMLS* 56 (1999): 755-763.

Reviewer #4 – Question #2

Are the larger clusters noted for *S. enterica* due to the smaller sample size (i.e. capturing less diversity than *E. coli*) or do you think this is a true representation of the greater population structure for *S. enterica*?

Because it is not feasible to produce more *S. enterica* isolates to analyse if the variation is due to smaller sample size, we have decided to select 143 *E. coli* isolates from the same farms from which the *S. enterica* isolates were cultured. Out of the 143 *S. enterica* isolates there were 113 isolates with a corresponding *E. coli* isolate that was cultured from the same sample. Therefore, we randomly selected only 30 samples keeping the same distribution of the farms as the 30 *S. enterica* isolates without a corresponding *E. coli* isolate. First, using the standard association index (I^S_A) to measure for clonality in the selected 143 *E. coli* isolates population (Haubold et al. 2000, Peng et al. 2022), we found the I^S_A to be 0.2833 (p -value < 0.0001) at whole cohort level and 0.1584 (p -value < 0.0001) at ST type level. These values are similar to the ones found using all the 518 *E. coli* isolates in our cohort (0.2126 (p -value < 0.0001) at whole cohort level and 0.1313 (p -value < 0.0001) at ST type level). Nonetheless, those values are much lower than the ones found for the 143 *S. enterica* isolates with an I^S_A of 0.9077 (p -value < 0.0001) at whole cohort level and 0.3883 (p -value < 0.0001) at ST type level indicating a stronger clonality compared to the *E. coli* isolates. Next, we created a network based on clusters of related isolates with less than 15 SNPs using the selected 143 *E. coli* isolates (Figure 10 below). We observed that this subset of *E. coli* isolates still does not contain large clusters as the *S. enterica* analysis. Moreover, only 52 isolates of the 143 isolates had a pairwise connection with 15 or less SNPs. Therefore, we hypothesize that the larger clusters noted for *S. enterica* are a representation of the greater population structure found in these isolates.

Figure 10. Network diagram showing pairwise connections between chicken (circle) and environmental (star) isolates with less than 15 pairwise SNP differences for 143 *E. coli* isolates. The panels in each row show the same network with the nodes colour-coded according to time point (left), source type (centre) or serotype/phylogroup (right). The lines between pairs of isolates are colour-coded by SNP number.

Updates to the submission

The following text was added to Supplementary Note 2:

Supplementary Material lines 32-33: “The larger clusters shown in Fig. S14 are likely due to the larger population structure for *S. Enterica* compared to *E. coli*.”

References

Haubold B, Hudson RR. LIAN 3.0: detecting linkage disequilibrium in multilocus data. *Linkage Analysis*. *Bioinformatics* 16, 847-848 (2000).

Peng Z, et al. Whole-genome sequencing and gene sharing network analysis powered by machine learning identifies antibiotic resistance sharing between animals, humans and environment in livestock farming. *PLoS Comput Biol* 18, e1010018 (2022).

Reviewer #4 – Question #3

It would be important to note in the discussion that the *in silico* mutations made for metabolic genes should be tested in an *in vitro* model (e.g. natural host derived and human macrophages and/or epithelial cells) in order to validate the physiological relevance of the metabolic pathway findings.

We agree that this is an important point, thanks. We have added the following text to the Discussion in the main manuscript, in a new paragraph where we comment on the limitations of the work – Discussion lines 783-790 and below:

Discussion lines 783-790: “A limitation of this study is that the analysis of association of AMR resistance to genetic features was primarily *in silico*, with experimental gene validation restricted to a small number of gene and a knockout approach (Supplementary Note 5). To fully validate the predictions made in this study experimental validation of the specific genetic mutations would be required. It is important to note that the *in silico* mutations made for metabolic genes should be tested in an *in vitro* model (e.g. natural host derived and human macrophages and/or epithelial cells) in order to validate the physiological relevance of the metabolic pathway findings.”

Other questions

Minor comments

Line 77, ARGs not yet defined

Corrected on introduction lines 74-76 and below.

Introduction lines 74-76: “A previous study showed that plasmids carrying antibiotic resistance genes (ARGs), can be transferred from *S. enterica* to *E. coli*”

Fig 1. The phylogroup colours in the figure legend for *E. coli* don't match the shading on the tree very well, whereas for Salmonella it looks very accurate. For example, phylogroup F appears as a pink shade in the legend and red on the tree, and phylogroup A appears a lighter blue than on the tree. As the figure is already very complex, this just creates a bit of confusion.

We thank the reviewer for this comment. This figure has been redrawn to simplify it as suggested by Reviewer 3. In addition we have changed the colours to pastels to improve the visual appearance. Whilst doing so we have ensured that the colours now match the legend. Please see Figure 1 in the manuscript and below.

Figure 1:

Line 220, change 'potential' to 'potentially'

Corrected on results lines 240-242 and below.

Results lines 240-242: "Together, these results show a diverse, non-clonal and evolutionarily distantly related cohort of *E. coli* isolates potentially indicative of large circulation populations of commensal *E. coli* in these farms, with many samples carrying multidrug resistance"

Line 248, delete the word 'many'

Corrected on results lines 270-272 and below.

Results lines 270-272: "When comparing plasmids type and content in the 113 *E. coli* and *S. enterica* strains isolated from the same samples we found that 70.6% of isolate pairs cultures from the same sample carried the same plasmid types"

Line 251, change 'chicken' to 'chickens' and italicise the p for the p value indicated.

Corrected on results lines 272-275 and below.

Results lines 272-275: "In the plasmid types that were shared in isolate pairs, we observed more sharing than would be expected by chance compared to typical plasmid prevalence in isolates collected from chickens in China ($p < 0.0001$, chi-squared test)"

Fig S9A, one of the arrows needs correction.

Corrected.

Line 308, is the word data repeated accidentally?

Here the repetition of word "data" is correct.

Line 327, Fig 8B should read Fig S8B

Corrected on results lines 389-391 and below.

Results lines 389-391: "Of these 21 models, 17 achieved high performance with an AUC greater than 0.9, and 13 of those achieved an AUC greater than 0.95, with the doxycycline performing best (AUC = 0.985), **Fig. S8b**"

Line 328, Fig 8C-G should read Fig S8C-G

Corrected on results lines 391-392 and below.

Results lines 391-392: "Similarly, all other performance metrics were good for these 17 models, **Fig. S8c-g**."

Lines 332, 338, 339, should all read Fig S9 as they are supp figs, not main figs

Corrected on results line 398 and 403-405 and below.

Results line 398: "For *S. enterica* a different ML pipeline was used as shown in **Fig. S9a**"

Results lines 403-405: "Of these thirteen models, all had good AUC performance, greater than 0.9, with ten achieving an AUC greater than 0.95, **Fig. S9b**. Performance across all metrics was high, **Fig. S9b-g**, and gentamicin (GEN) performed best across all metrics."

Line 352, should this read 'resistance phenotype'?

Corrected on results lines 420-422 and below.

Results lines 420-422: "**Figs 4 and 5** use a bee swarm plot to show the correlation of each of the top ten most important genes for each model for *E. coli* and *S. enterica* respectively, to predict the resistance phenotype."

Check grammar in Discussion, e.g., between lines 579 – 583, words like 'a' and 'the' are missing.

Corrected on discussion lines 704-708 and below.

Discussion lines 704-708: "In this work, by using a data-mining approach powered by machine learning, Bayesian divergence analysis and genome-scale metabolic modelling, we showed that at a larger scale, differences in the phylogeny and in the evolution of *S. enterica* and *E. coli* were observed, but, at a finer scale, most isolates of each species inhabiting the same host and environment, possessed same the plasmids and MGEs carrying clinically relevant ARGs."

Line 625, double comma needs to be corrected.

Corrected on results lines 756-758 and below.

Results lines 756-758: "Given the constantly evolving nature of AMR, it is crucial to advance techniques capable of detecting novel and emerging genetic determinants that underlie complex resistant phenotypes."

REVIEWER COMMENTS

Reviewer #1 (Remarks to the Author):

I compliment the authors for systematically and rigorously addressing all my previous comments. The revised manuscript is in a considerably improved shape and their responses are exemplary.

Reviewer #3 (Remarks to the Author):

I would like to thank the authors for considering my comments. I have outlined my remaining major and minor concerns below:

Major concerns:

1. Unfortunately, the authors have not addressed my concerns regarding the correlation between AMR genotypes & phenotypes (noted as reviewer #3 question #2). Specifically, the authors have not presented data that shows the correlation between laboratory derived AMR phenotypes and bioinformatically inferred AMR determinants/genotypes from commonly used databases such as CARD. Therefore, it is unclear at present if AMR was adequately explained using these known determinants and if the ML approach was necessary.
2. The authors have not provided adequate evidence of temporal signal to support dating analyses in the revised version of this manuscript as highlighted as a major comment in my previous review (detailed below). No response was provided to this comment in the 'response to reviewer comments' as it was omitted. I note that line 200 mentions that a temporal signal was found for *E. coli* serotypes O83:H42 and O8:H16 but there is no evidence or description of the analyses performed provided. Please include date-randomisation testing plots showing that a temporal signal is present and describe how the underlying analyses were carried out in the methods section.

Original comment:

"3. While the molecular dating analyses presented in the manuscript appeared to have used appropriate model selection processes, there was no validation of the temporal signal. The current field standards for validation of temporal signal are either date-randomisation testing (e.g. PMID: 28348834), or BETS analysis (e.g. PMID: 32895707). This is particularly problematic for the data presented as they were collected over a relatively short time frame of 2.5 years, with small sample sizes for each serovar/ST analysed, which can make it exceptionally difficult to infer accurate MRCA estimates. [...] Without these additional analyses and reported data the robustness of the estimates presented cannot be assessed."

Minor concerns:

1. The authors state they revised the yellow colour in Figure 4 (now Figure 3) to improve ease of reading, however, this does not appear to have been altered in the revised version of the manuscript.
2. Regarding the substitution rates in Table S10, it is difficult to compare these to those from past studies as they are not presented in the form of genome-wide rates (substitutions/genome/year). Please consider making this alteration. I also note that there is an order of magnitude difference between some of the *E. coli* rates which may indicate an issue with the underlying analysis, or a biologically interesting trend, though these rates are difficult to interpret as they are currently presented as substitutions per variable site per year.

Reviewer #4 (Remarks to the Author):

The authors have clearly made a very comprehensive and careful response to all 4 reviewers' comments. The attention to detail is much appreciated.

As for my specific comments, I am satisfied that the authors have addressed these in a satisfactory manner. I was most interested in the broader implications/applications of the novel machine learning processes and outcomes in a global context. Here, the authors fairly outlined that using China is a model for future studies and analyses and lays critical groundwork for similar projects to be undertaken in other parts of the world. Especially in those countries where livestock for human consumption is a major export and of critical economic value.

Point-by-point response to reviewers

We thank the Reviewers #1 and #4 for their supportive remarks and reviewer #3 for their supportive feedback and for additional insightful comments over two rounds of revisions. In the following, a point-by-point response to all the questions and comments posed by Reviewer #3 is provided. The original questions are in blue, our replies in black.

Detailed information of which figures and supplementary material were changed is provided below and in the individual responses to the reviewers' comments.

Items included in this submission:

- i. Cover letter
- ii. Point-by-point Response to reviewers (ResponsetoReviewers.pdf);
- iii. Revised manuscript marked-up copy (Manuscript_markedup.pdf)
- iv. Revised manuscript clean copy (Manuscript.pdf)
- v. Revised supplementary material (SupplementaryMaterial.pdf and supplementary table files)

Reviewer #1

I compliment the authors for systematically and rigorously addressing all my previous comments. The revised manuscript is in a considerably improved shape and their responses are exemplary.

Many thanks for this assessment, we appreciate you taking the time to review our manuscript.

Reviewer #3:

I would like to thank the authors for considering my comments. I have outlined my remaining major and minor concerns below:

We thank you for taking the time to critique our manuscript and have responded to each of your remaining points below.

Major concerns:

1. Unfortunately, the authors have not addressed my concerns regarding the correlation between AMR genotypes & phenotypes (noted as reviewer #3 question #2). Specifically, the authors have not presented data that shows the correlation between laboratory derived AMR phenotypes and bioinformatically inferred AMR determinants/genotypes from commonly used databases such as CARD. Therefore, it is unclear at present if AMR was adequately explained using these known determinants and if the ML approach was necessary.

We apologize that the response we gave in the previous revision did not fully satisfy your concerns regarding this point. In that revision we focused on the novelty of our results, highlighting that only 1% of our results were known AMR genes that can be currently found in the databases. In particular, we mentioned that in our specific case, the ML analysis identified a large number of accessory genes and mutations correlated to AMR phenotypes, many of which were found to be novel. In *E. coli*, of the 4119 genes selected by ML, only 1% were known AMR genes (as found in public databases, CARD (Alcock *et al*, 2020), ARG-Annot (Gupta *et al*, 2014), Resfinder

(Bortolaia *et al.* 2020) and AMRfinder (Feldgardeni *et al.* 2019)) whilst 99% were novel AMR associations. For *S. enterica*, of the 3501 genes selected, only 1.1% were known molecular determinants of AMR as found in public databases, whilst 98.9% were novel associations. However, the point you raise here is important, and we agree that we need to assess whether the existing known AMR genes in public databases can sufficiently explain the laboratory-derived phenotypes we observe. One way to test this is to consider the intersection of resistant isolates with the presence of known AMR genes (CARD) as previously done by us and others (Salnikova *et al.*, 2019; Abdelrazik *et al.*, 2021, Peng *et al.*, 2022). We have in fact previously done this analysis on a subset of our Chinese *E. coli* data (n=154) which was published as a pilot study (Peng *et al.*, 2022). In particular, we used a centred Jaccard/Tanimoto coefficient of similarity to test in a pairwise manner whether the observed AMR phenotypes could be explained by known AMR-associated genes (from (CARD [38], ResFinder [39], ARG Annot [40], NCBI AMRfinder [41])). Values spanned the range -0.13 to 0.53, where 1.0 would indicate a perfect positive association and -1 would indicate a perfect negative association. These results indicated that no single known AMR-associated gene could be found that explained the resistance phenotype observed for each antibiotic (Peng *et al.*, 2022), for this reason in that study we had to revert to ML to deduce the relationship between genome content and AMR profiles of each isolate to each one of the antimicrobials.

To address the reviewer comment for this study, we have now applied a Jaccard/Tanimoto analysis following the methodology of Salnikova *et al.*, 2019 to test whether the antibiotic resistance phenotypes of the *E. coli* and *S. enterica* could be fully explained by known AMR-associated genes (CARD) present in the cohort. Jaccard/Tanimoto similarity coefficients were calculated in a pairwise manner between the presence of resistant AST profiles for each antibiotic and the presence of known AMR genes as found in CARD. The Jaccard coefficient was calculated as $|A \cap B| / (|A| + |B| - |A \cap B|)$, where A is a resistant phenotype and B is the presence of the AMR gene. The significance of the Jaccard coefficient was tested statistically using an MCA approach (Chung, Miasojedow *et al.* 2019) and Jaccard coefficients with an FDR adjusted *p* value < 0.05 were used (Salnikova, Chernyshova *et al.* 2019). For *E. coli*, the maximum statistically significant Jaccard value (see Fig. 1 below) was 0.62 with many antibiotics achieving a maximum Jaccard association of much less than this (range of maximum Jaccard coefficient per antibiotic 0.006-0.62, mean 0.32). Similarly for *S. enterica* the maximum statistically significant Jaccard similarity coefficient was 0.71 (range of maximum Jaccard coefficient per antibiotic 0.01-0.71, mean 0.51), see Fig. 2 below.

Fig. 1. Pairwise Jaccard similarity coefficients for *E. coli*, testing the association between known AMR genes (CARD) and laboratory derived resistance phenotypes.

Fig. 2. Pairwise Jaccard similarity coefficients for *S. enterica*, testing the association between known AMR genes (CARD) and laboratory derived resistance phenotypes.

Whilst the above analysis used all the known AMR genes found in CARD and tested association with all laboratory-derived AMR phenotypes, we also checked the subset of association between known AMR genes selected in CARD based on the class of antibiotic they are resistant to and the specific laboratory-derived AMR phenotype to which resistance is conferred, to ensure we had not introduced noise by including all genes. Using the same methodology as above, this gave for *E. coli*, a maximum statistically significant Jaccard value of 0.62 as before and for *S. enterica* the maximum statistically significant Jaccard similarity coefficient marginally reduced to 0.69, see Fig. 3 below.

Given the low values of the Jaccard coefficients which leave at least 30% of the laboratory derived resistance phenotypes unexplained, we conclude that AMR we observe in our isolates cannot be adequately explained by correlation between the presence of known AMR genes (as found in the CARD database) and AMR phenotypes in either the *E. coli* or *S. enterica* isolates in our study. Hence other methodology, such as the machine learning we apply in our manuscript are needed.

We thank the reviewer for suggesting this additional analysis and have now added a summary of this into the manuscript in the Results lines 339-350, and Methods lines 921-928, and Supplementary Table 6.

Results lines 339-350: “We assessed whether the resistance/susceptibility profiles of either *E. coli* or *S. enterica* could be fully explained by the presence of known AMR genes (as found in CARD (Alcock *et al*, 2020)). As previously done by us and others (Salnikova *et al*, 2019, Abdelrazik *et al*, 2021, Peng *et al*, 2022), we used Jaccard/Tanimoto similarity

coefficients (Jaccard 1912, Tanimoto 1958) between the AMR phenotypes and the known AMR genes in a pairwise manner and found for *E. coli*, the maximum statistically significant Jaccard value was 0.62 with many antibiotics achieving a maximum Jaccard association of much less than this (range of maximum Jaccard coefficient per antibiotic 0.006-0.62, mean 0.32). Similarly for *S. enterica*, the maximum statistically significant Jaccard similarity coefficient was 0.71 (range of maximum Jaccard coefficient per antibiotic 0.01-0.71, mean 0.51). Given the low values of the Jaccard coefficients, we conclude that the presence of known AMR genes (as found in the CARD database) alone was not able to adequately explain the AMR phenotypes in either the *E. coli* or *S. enterica* isolates in our study, necessitating an alternative approach."

Methods lines 921-928: "Jaccard/Tanimoto similarity coefficients were calculated in a pairwise manner between the presence of resistant AST profiles for each antibiotic and the presence of known AMR genes as found in CARD. The Jaccard coefficient was calculated as $|A \cap B| / (|A| + |B| - |A \cap B|)$, where A is a resistant phenotype and B is the presence of the AMR gene. A Jaccard value of 1 represents perfect intersection and 0 represents no intersection. The significance of the Jaccard coefficient was tested statistically using an MCA approach (Chung, Miasojedow et al. 2019) and Jaccard coefficients with an FDR adjusted p value < 0.05 were used (Salnikova, Chernyshova et al. 2019)"

Chung, N. C., B. Miasojedow, M. Startek and A. Gambin (2019). "Jaccard/Tanimoto similarity test and estimation methods for biological presence-absence data." *BMC Bioinformatics* 20(15): 644.

Salnikova, L. E., E. V. Chernyshova, L. A. Anastasevich and S. S. Larin (2019). "Gene- and Disease-Based Expansion of the Knowledge on Inborn Errors of Immunity." *Front Immunol* 10: 2475.

Alcock, B. P., A. R. Raphenya, T. T. Y. Lau, K. K. Tsang, M. Bouchard, A. Edalatmand, W. Huynh, A. V. Nguyen, A. A. Cheng, S. Liu, S. Y. Min, A. Miroshnichenko, H. K. Tran, R. E. Werfalli, J. A. Nasir, M. Oloni, D. J. Speicher, A. Florescu, B. Singh, M. Faltyn, A. Hernandez-Koutoucheva, A. N. Sharma, E. Bordeleau, A. C. Pawlowski, H. L. Zubyk, D. Dooley, E. Griffiths, F. Maguire, G. L. Winsor, R. G. Beiko, F. S. L. Brinkman, W. W. L. Hsiao, G. V. Domselaar and A. G. McArthur (2020). "CARD 2020: antibiotic resistance surveillance with the comprehensive antibiotic resistance database." *Nucleic Acids Res* 48(D1): D517-d525.

Abdelrazik, E., M. Oweda and M. El-Hadidi (2021). Benchmarking of Antimicrobial Resistance Gene Detection Tools in Assembled Bacterial Whole Genomes. 2021 3rd Novel Intelligent and Leading Emerging Sciences Conference (NILES), IEEE.

Jaccard, P. (1912). "The distribution of the flora in the alpine zone.1." *New Phytologist* 11(2): 37-50.

Tanimoto, T. T. (1958). "Elementary mathematical theory of classification and prediction."

Peng, Z., A. Maciel-Guerra, M. Baker, X. Zhang, Y. Hu, W. Wang, J. Rong, J. Zhang, N. Xue, P. Barrow, D. Renney, D. Stekel, P. Williams, L. Liu, J. Chen, F. Li and T. Dottorini (2022). "Whole-genome sequencing and gene sharing network analysis powered by machine learning identifies antibiotic resistance sharing between animals, humans and environment in livestock farming." *PLOS Computational Biology* 18(3): e1010018.

2. The authors have not provided adequate evidence of temporal signal to support dating analyses in the revised version of this manuscript as highlighted as a major comment in my previous review (detailed below). No response was provided to this comment in the 'response to reviewer comments' as it was omitted. I note that line 200 mentions that a temporal signal was found for *E. coli* serotypes O83:H42 and O8:H16 but there is no evidence or description of the analyses performed provided. Please include date-randomisation testing plots showing that a

temporal signal is present and describe how the underlying analyses were carried out in the methods section.

Original comment:

“3. While the molecular dating analyses presented in the manuscript appeared to have used appropriate model selection processes, there was no validation of the temporal signal. The current field standards for validation of temporal signal are either date-randomisation testing (e.g. PMID: 28348834), or BETS analysis (e.g. PMID: 32895707). This is particularly problematic for the data presented as they were collected over a relatively short time frame of 2.5 years, with small sample sizes for each serovar/ST analysed, which can make it exceptionally difficult to infer accurate MRCA estimates. [...] Without these additional analyses and reported data the robustness of the estimates presented cannot be assessed.”

We sincerely apologise that this response was missed from the previous reviewer's response file. We did produce a response to this very valid question and unfortunately accidentally removed it from the final version.

We thank the reviewer for this comment and agree that the inclusion of the validation of temporal signal was a useful addition to the manuscript. Originally our analyses were assessed for temporal signal using TempEst, however we have also conducted a BETS analysis as suggested by the reviewer and included these results as Supplementary Table 2 (also shown below). In summary the BETS analysis showed support for a temporal analysis in all the *S. enterica* serogroups and all *E. coli* clades except Clades B and D, (now noted in the Results Lines 192-198 and below). As a result, the Clade B and D results have been removed from the Supplementary figure S4, and this has been mentioned in the methods (Lines 981-986 and below). In relation to the lack of substitution rates and 95% HPD values these have all now been added as a supplementary table, Table S3. The substitution rates for *E. coli* and *S. enterica* in our cohort predict *E. coli* mutating slightly faster than *S. enterica* but the difference is less than an order of magnitude. Compared to published substitution rates (Reeves, Liu et al. 2011, Hawkey, Le Hello et al. 2019), our substitution rates are two orders of magnitude greater. One reason for this difference may be that bacterial stress induced by the usage of antibiotics on the farms in our cohort could be leading to increased mutation rates as has been discussed in (Mao, Lane et al. 1997, Martinez and Baquero 2000, Andersson and Hughes 2012, Long, Miller et al. 2016). This has now been added to the Results lines 202-208 and below.

Supplementary Table S2:

Species	Group	Log Bayes Factor
S. enterica	Enteritidis	1.35
S. enterica	Indiana	2.33
S. enterica	Kentucky	8.30
S. enterica	Kedougou	21.91
S. enterica	Havana	107.63
E. coli	Clade A	2759.45
E. coli	Clade B1	-919.95
E. coli	Clade D	-302.27
E. coli	Clade E	41.98
E. coli	Clade F	770.73

E. coli	O8H16	-39.406
E. coli	O83H42	-3.801

Supplementary Fig. S4:

Fig. S4. Bayesian divergence analysis of *E. coli* isolates in five phylogroups (A) Clade A; (B) Clade E and (C) Clade F. Sample source and the farm the isolates were taken from and the timepoint for each sample are displayed as coloured rings.

Methods Lines 981-986: “BETS analysis(Duchene, Lemey et al. 2020) was used to check for a temporal signal in each serotype/phylogroup and found temporal signal in five *S. enterica* serotypes (Enteritidis, Indiana, Kentucky, Kedougou, Havana) and three *E. coli* phylogroups (A, E and F), which were taken further for analysis in BEAST, Table S2. The BETS analysis also showed that there was insufficient temporal signal in *E. coli* Clades B1 and D, so these clades were not subjected to BEAST analysis.”

Results Lines 192-198: “Finally, to better understand the phylogenetic and SNP distance differences observed between *E. coli* and *S. enterica* and to explain the evolution of these two species, we performed a Bayesian inference of phylogeny for the major phylogroups and

serotypes in our cohort. Bayesian estimation of Temporal Signal (BETS) analysis showed eight of the major phylogroups and serotypes contained a temporal signal suitable for analysis (*S. enterica*: Enteritidis, Indiana, Kentucky, Kedougou, Havana; *E. coli*: A, E and F), Table S2."

Results Lines 202-208: "Root heights varied also with *S. enterica* serotype averages between 5-29 years and *E. coli* phylogroup root height averages between 151 and 400 years, Table S3. Predicted nucleotide substitution rates for both species were relatively consistent across phylogroups and serotypes and were higher (two orders of magnitude) than though typically found in literature (Reeves, Liu et al. 2011, Hawkey, Le Hello et al. 2019). These higher than expected rates may have been caused bacterial stress induced by the usage of antibiotics on the farms in our cohort, which could be leading to increased mutation rates (Mao, Lane et al. 1997, Martinez and Baquero 2000, Andersson and Hughes 2012, Long, Miller et al. 2016)."

Andersson, D. I. and D. Hughes (2012). "Evolution of antibiotic resistance at non-lethal drug concentrations." *Drug Resistance Updates* 15(3): 162-172.

Duchene, S., P. Lemey, T. Stadler, S. Y. W. Ho, D. A. Duchene, V. Dhanasekaran and G. Baele (2020). "Bayesian Evaluation of Temporal Signal in Measurably Evolving Populations." *Mol Biol Evol* 37(11): 3363-3379.

Hawkey, J., S. Le Hello, B. Doublet, S. A. Granier, R. S. Hendriksen, W. F. Fricke, P. J. Ceysens, C. Gomart, H. Billman-Jacobe, K. E. Holt and F. X. Weill (2019). "Global phylogenomics of multidrug-resistant *Salmonella enterica* serotype Kentucky ST198." *Microb Genom* 5(7).

Long, H., S. F. Miller, C. Strauss, C. Zhao, L. Cheng, Z. Ye, K. Griffin, R. Te, H. Lee, C. C. Chen and M. Lynch (2016). "Antibiotic treatment enhances the genome-wide mutation rate of target cells." *Proc Natl Acad Sci U S A* 113(18): E2498-2505.

Mao, E. F., L. Lane, J. Lee and J. H. Miller (1997). "Proliferation of mutators in A cell population." *J Bacteriol* 179(2): 417-422.

Martinez, J. L. and F. Baquero (2000). "Mutation Frequencies and Antibiotic Resistance." *Antimicrobial Agents and Chemotherapy* 44(7): 1771-1777.

Reeves, P. R., B. Liu, Z. Zhou, D. Li, D. Guo, Y. Ren, C. Clabots, R. Lan, J. R. Johnson and L. Wang (2011). "Rates of Mutation and Host Transmission for an *Escherichia coli* Clone over 3 Years." *PLOS ONE* 6(10): e26907.

Minor concerns:

1. The authors state they revised the yellow colour in Figure 4 (now Figure 3) to improve ease of reading, however, this does not appear to have been altered in the revised version of the manuscript.

Apologies, the revised version of this file was omitted from the re-submission, this has not been corrected, see below:

Figure S3:

2. Regarding the substitution rates in Table S10, it is difficult to compare these to those from past studies as they are not presented in the form of genome-wide rates (substitutions/genome/year). Please consider making this alteration. I also note that there is an order of magnitude difference between some of the *E. coli* rates which may indicate an issue with the underlying analysis, or a biologically interesting trend, though these rates are difficult to interpret as they are currently presented as substitutions per variable site per year.

We thank the reviewer for this comment. Rates in units of substitutions/site/year were included to make comparison easy between other studies with similar analyses, however, we agree with the reviewer that adding genome-wide rates would aid comparison with past papers. Hence, we have now included these in Table S3 (previously Table S10). When considering genome-wide substitution rates in *E. coli*, clade E is predicted to mutate more slowly than clades A and F. Others have observed that phylogroup E tends to have a larger genome (as also seen in our data) resulting in lower replication rates (Clermont, Condamine et al. 2021) and one could speculate that this lower replication rate could result in the lower mutation rate we predict. This is a pattern we also see in our *S. enterica* isolates, with the largest genomes predicted to have the slowest mutation rates. We have added this to the results lines 208-214.

Results lines 208-214: “Of note, when considering genome-wide substitution rates in *E. coli*, clade E is predicted to mutate more slowly than clades A and F. Others have observed that phylogroup E tends to have a larger genome (as also seen in our data) resulting in lower replication rates (Clermont, Condamine et al. 2021) and one could speculate that this lower replication rate could result in the lower mutation rate we predict. This is a pattern we also see in our *S. enterica* isolates, with the largest genomes predicted to have the slowest mutation rates.”

Clermont, O., B. Condamine, S. Dion, D. M. Gordon and E. Denamur (2021). “The E phylogroup of *Escherichia coli* is highly diverse and mimics the whole *E. coli* species population structure.” *Environmental Microbiology* 23(11): 7139-7151.

Reviewer #4

The authors have clearly made a very comprehensive and careful response to all 4 reviewers' comments. The attention to detail is much appreciated.

As for my specific comments, I am satisfied that the authors have addressed these in a satisfactory manner. I was most interested in the broader implications/applications of the novel machine learning processes and outcomes in a global context. Here, the authors fairly outlined that using China is a model for future studies and analyses and lays critical groundwork for similar projects to be undertaken in other parts of the world. Especially in those countries where livestock for human consumption is a major export and of critical economic value.

Many thanks for this assessment, we appreciate you taking the time to review our manuscript.

REVIEWERS' COMMENTS

Reviewer #3 (Remarks to the Author):

I would like to thank the authors for considering my comments on the manuscript. I am satisfied that my concerns have been appropriately and thoughtfully addressed.

Reviewer #4 (Remarks to the Author):

I am satisfied that the authors have addressed my comments and appreciate the attention to detail for reviewer 3's additional concerns.